# Convergent organization of aberrant MYB complex controls oncogenic gene expression in acute myeloid leukemia

Sumiko Takao[1,2†], Lauren Forbes[1,2,3†], Masahiro Uni[1,2†], Shuyuan Cheng[1,2,3], Jose Mario Bello Pineda[4,5,6,7], Yusuke Tarumoto[8,9], Paolo Cifani[1], Gerard Minuesa[1], Celine Chen[1], Michael G Kharas[1,3], Robert K Bradley[4,6,7], Christopher R Vakoc[8], Richard P Koche[10], Alex Kentsis[1,2,3]*

[1]Molecular Pharmacology Program, Sloan Kettering Institute, New York, United States; [2]Tow Center for Developmental Oncology, Department of Pediatrics, Memorial Sloan Kettering Cancer Center, New York, United States; [3]Departments of Pharmacology and Physiology & Biophysics, Weill Cornell Graduate School of Medical Sciences, Cornell University, New York, United States; [4]Computational Biology Program, Public Health Sciences Division, Fred Hutchinson Cancer Research Center, Seattle, United States; [5]Medical Scientist Training Program, University of Washington, Seattle, United States; [6]Basic Sciences Division, Fred Hutchinson Cancer Research Center, Seattle, United States; [7]Department of Genome Sciences, University of Washington, Seattle, United States; [8]Cold Spring Harbor Laboratory, Cold Spring Harbor, United States; [9]Institute for Frontier Life and Medical Sciences, Kyoto University, Kyoto, Japan; [10]Center for Epigenetics Research, Sloan Kettering Institute, New York, United States

*For correspondence: kentsisresearchgroup@gmail.com

†These authors contributed equally to this work

**Abstract** Dysregulated gene expression contributes to most prevalent features in human cancers. Here, we show that most subtypes of acute myeloid leukemia (AML) depend on the aberrant assembly of MYB transcriptional co-activator complex. By rapid and selective peptidomimetic interference with the binding of CBP/P300 to MYB, but not CREB or MLL1, we find that the leukemic functions of MYB are mediated by CBP/P300 co-activation of a distinct set of transcription factor complexes. These MYB complexes assemble aberrantly with LYL1, E2A, C/EBP family members, LMO2, and SATB1. They are organized convergently in genetically diverse subtypes of AML and are at least in part associated with inappropriate transcription factor co-expression. Peptidomimetic remodeling of oncogenic MYB complexes is accompanied by specific proteolysis and dynamic redistribution of CBP/P300 with alternative transcription factors such as RUNX1 to induce myeloid differentiation and apoptosis. Thus, aberrant assembly and sequestration of MYB:CBP/P300 complexes provide a unifying mechanism of oncogenic gene expression in AML. This work establishes a compelling strategy for their pharmacologic reprogramming and therapeutic targeting for diverse leukemias and possibly other human cancers caused by dysregulated gene control.

## Introduction

Gene dysregulation is one of the most prevalent features in human cancers (*Bradner et al., 2017*). In many tumors, this is due to the pathogenic mutations of promoters, enhancers, and genes encoding either transcription factors or factors that regulate chromatin and gene expression. In blood cancers, and acute myeloid leukemias (AML) in particular, aberrant gene expression is thought to

contribute to most important properties of leukemia cells, including self-renewal, growth, and resistance to therapy. For example, numerous pathogenic chromosomal translocations in AML, such as those involving *AML1* (*RUNX1*) and *MLL1* (*KMT2A*) produce chimeric transcription or chromatin remodeling factors that cause disease (*Look, 1997*). Consequently, therapeutic strategies aimed at restoring normal gene expression are compelling because of their ability to target the causal molecular processes and induce leukemia cell differentiation and elimination, leading in principle to durable disease control.

While specific molecular dependencies have been identified for some genetic subtypes of AML, such as DOT1L or Menin inhibition for *MLL*-rearranged leukemias (*Krivtsov et al., 2019*), and CARM1 inhibition for *AML1*-rearranged leukemias (*Greenblatt et al., 2019*), distinct pathogenetic mechanisms of diverse AML subtypes also appear to converge on shared molecular pathways. For example, approximately 25% of adult and childhood AMLs, including both *MLL*-rearranged and non-rearranged cases, require aberrant activation of the transcription factor MEF2C, conferring susceptibility to MARK and SIK inhibitors, which are currently being explored for clinical trials for patients (*Brown et al., 2018*; *Tarumoto et al., 2018*; *Vakoc and Kentsis, 2018*). Similarly, nearly 50% of examined AML specimens exhibit aberrant activation of HGF/MET/FGFR signaling (*Kentsis et al., 2012*) and are being currently targeted therapeutically in the ongoing clinical trial of combined MET and FGFR inhibitors in patients with relapsed or refractory AML (ClinicalTrials.gov Identifier NCT03125239). Even for therapies targeting leukemogenic proteins directly, such as inhibitors of IDH1/2, FLT3, KIT, SYK, as well as epigenetic and apoptotic therapies such as decitabine and venetoclax, their therapeutic efficacy and resistance depend on the underlying gene expression phenotypic states of AML cells (*Tyner et al., 2018*). Thus, there is intense interest in defining shared molecular dependencies controlling leukemogenic gene expression in AML that can provide effective therapeutic options for patients.

Recently, MYB has emerged as a therapeutic target in AML, as transient suppression of *Myb* nearly completely eliminates leukemia development in mouse models in vivo while sparing normal hematopoietic cells (*Zuber et al., 2011*). Indeed, pioneering studies have implicated Myb as a key mediator of leukemias (*Klempnauer and Bishop, 1984*; *Luger et al., 2002*). MYB is the cellular homologue of the viral *v-Myb* oncogene that can cause avian leukemias and function as a pioneer transcription factor in mammalian cells (*Biedenkapp et al., 1988*). MYB functions as a master regulator of gene expression in diverse cell types, including hematopoietic cells where it controls cell proliferation and differentiation (*Ramsay and Gonda, 2008*). Both mutations and translocations of *MYB* have causal roles in various human malignancies, including leukemias. For example, aberrant expression of *TAL1* in T-cell acute lymphoid leukemia (T-ALL) is induced by pathogenic somatic mutations that create neomorphic MYB-binding sites (*Mansour et al., 2014*). Likewise, *MYB* is recurrently rearranged in distinct subtype of blastic plasmacytoid dendritic cell neoplasms (BPDCN), a highly refractory hematologic malignancy (*Suzuki et al., 2017*).

Notably, the *Booreana* strain of mice that impairs the binding of Myb by its co-activator CBP/P300 (Crebbp/Ep300) due to the mutation of Myb E308G in its transcriptional activation domain is resistant to leukemogenesis induced by the otherwise fully penetrant *MLL-AF9* and *AML1-ETO* oncogenes but has largely normal hematopoiesis (*Pattabiraman et al., 2014*). Altogether, these findings indicate that MYB and its co-factor CBP/P300 are fundamentally dysregulated in AML, presumably through disordered gene expression that characterizes most forms of this disease. However, the specific details of this mechanism remain poorly understood, largely due to the lack of suitable tools.

Recently, we developed a peptidomimetic inhibitor of MYB:CBP/P300 (*Ramaswamy et al., 2018*). Here, we report its second-generation version that has significantly increased potency, and consequently suppresses leukemic MYB functions in most AML subtypes tested, while relatively sparing normal hematopoietic progenitor cells. By rapid and selective peptidomimetic interference with the binding of CBP/P300 to MYB, but not CREB or MLL1, we find that the leukemic functions of MYB are mediated by CBP/P300-mediated co-activation of a distinct set of transcriptional factor complexes that are aberrantly assembled with MYB in AML cells, which is associated at least in part with their inappropriate expression. This therapeutic remodeling is accompanied by dynamic redistribution of CBP/P300 complexes to genes that control cellular differentiation and growth. These findings provide a unifying mechanism of oncogenic gene control, involving aberrant assembly of transcription factor complexes and sequestration of CBP/P300 to promote oncogenic gene

expression and block cellular differentiation. This paradigm should apply to other human cancers caused by dysregulated gene control, elucidate specific molecular determinants of leukemia pathogenesis, and enable the development of definitive therapies for patients.

## Results

### Genome-wide CRISPR screen identifies CBP requirement for the susceptibility of AML cells to peptidomimetic blockade of MYB:CBP/P300

In prior work, we used genetic and structural evidence to design a peptidomimetic inhibitor of the MYB:CBP/P300 transcription coactivation complex, termed MYBMIM (*Ramaswamy et al., 2018*). To elucidate its mechanisms of action in an unbiased manner, we carried out a genome-wide CRISPR knockout screen to identify genes whose depletion affects the susceptibility of AML cells to MYB-MIM. We used MOLM13 cells stably expressing Cas9 and transduced them with lentiviruses encoding the genome-wide GeCKOv2 library at a multiplicity of infection of 0.3 (*Figure 1A*). Following selection of transduced cells, we treated them with MYBMIM or PBS control in independent biological replicates, and quantified the enrichment and depletion of cell clones expressing specific sgRNAs by DNA sequencing of lentiviral barcodes (*Figure 1A*). This screen revealed a variety of genes whose depletion confers relative resistance and susceptibility to MYBMIM treatment, consistent with the presence of diverse cellular pathways that regulate oncogenic gene expression (*Supplementary file 2a*). The most significantly affected gene whose depletion was required to confer resistance to MYB-MIM was *CBP* (*Figure 1B*). In contrast, CBP depletion exhibited no enrichment upon PBS treatment (*Figure 1—figure supplement 1*). Thus, MYBMIM is a specific inhibitor of MYB:CBP, and emphasizes the exquisite specificity of peptidomimetic inhibitors as pharmacologic modulators of protein interactions.

### CRYBMIM is a peptidomimetic chimera that specifically binds CBP/P300 KIX domain

Of all the functional genetic dependencies examined to date, the transcription factor *MYB* demonstrates the broadest dependency across diverse AML subtypes, as compared to other non-hematopoietic cancers (*Tarumoto et al., 2018*). To generalize this analysis, we queried 688 human cancer cell lines tested as part of the DepMap Cancer Dependency Map to identify genes that are selectively required for the growth and survival of leukemia as compared to other cancer types. We found that *MYB* is the most significantly required human gene in 37 leukemia cell lines, including 20 AML cell lines, of diverse molecular subtypes (p=1.1e-15; *Figure 2A*).

MYB target gene activation requires its specific interaction with CREB-binding protein (CBP)/P300 for co-activation (*Dai et al., 1996*). The helical MYB transactivation domain comprising residues 293–310 binds to the KIX domain of CBP/P300 (*Zor et al., 2004*). Using molecular mechanics simulations, we previously developed a peptidomimetic inhibitor of this interaction, termed MYBMIM (*Ramaswamy et al., 2018*). MYBMIM uses stereoselective substitution of D-amino acids to confer proteolytic stability, and the cationic TAT domain for cell penetration. As a result, MYBMIM can specifically inhibit MYB:CBP/P300 binding in cells, but its activity is less pronounced in non-MLL-rearranged AML cells (*Ramaswamy et al., 2018*). Given that MYBMIM bound to recombinant CBP KIX domain with the dissociation constant of 21.3 ± 2.9 μM, as compared to the native MYB peptide of 4.2 ± 0.5 μM (*Ramaswamy et al., 2018*), we reasoned that a peptidomimetic inhibitor with higher affinity to CBP/P300 would be more effective.

Consequently, we used molecular modeling to extend MYBMIM into the adjoining binding site that binds CREB (*Radhakrishnan et al., 1997*; *Cheng et al., 2008*). We appended CREB residues 124–147 to MYBMIM, while replacing the EKIRK motif to maintain favorable backbone geometry, as confirmed by molecular energy minimization calculations in implicit solvent (*Figure 2B and C* and *Figure 2—figure supplement 1*). Termed CRYBMIM, this design preserves key MYB residues implicated in leukemic transformation, including E308 which forms a salt bridge with CBP, while including the pS133-containing portion of CREB that is responsible for its high-affinity binding to CBP (*Zor et al., 2004*; *Cheng et al., 2008*; *Radhakrishnan et al., 1997*). Similarly, we also designed two additional peptidomimetic inhibitors targeting the distinct CREB and MLL1-binding sites that are

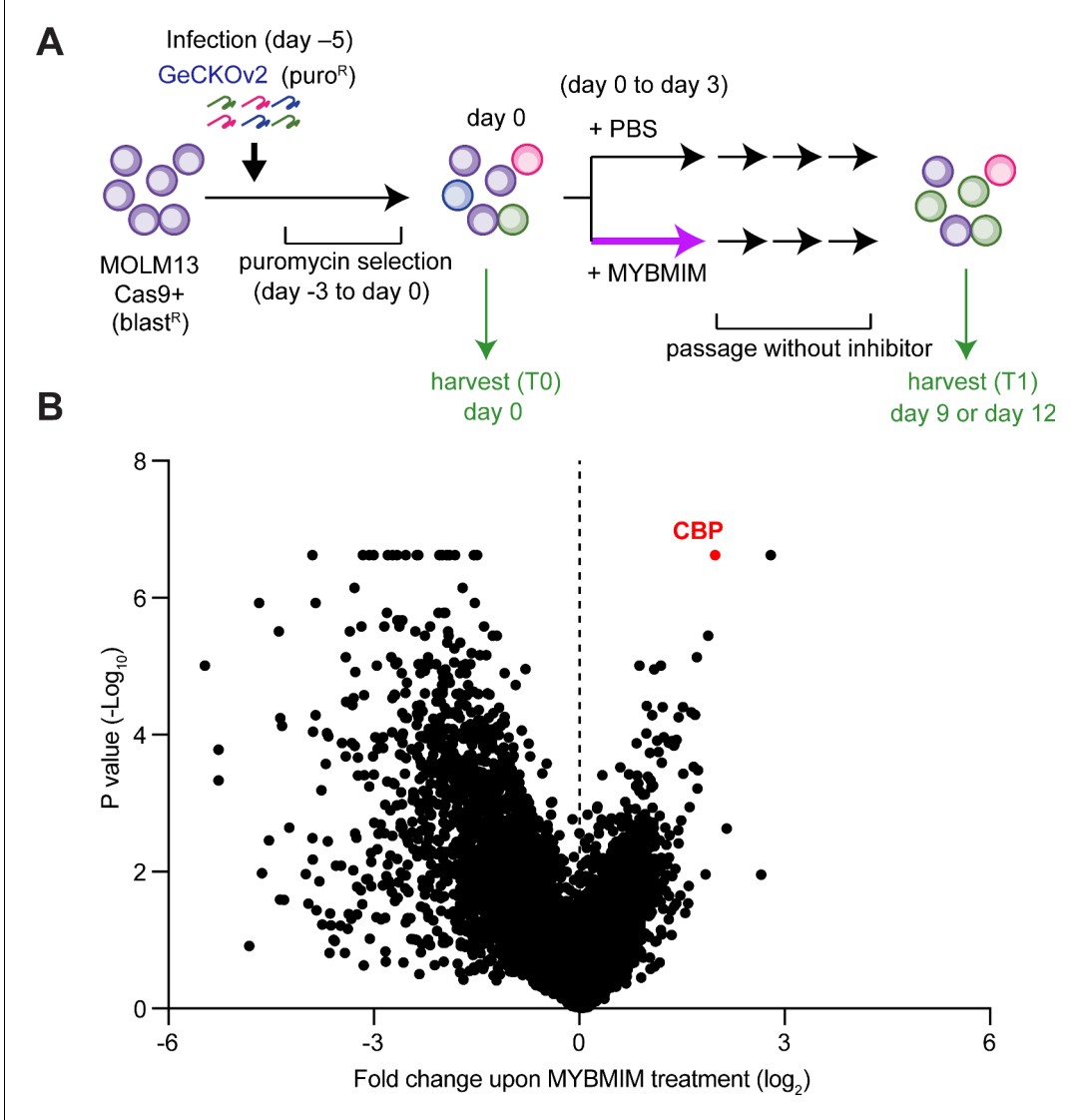

**Figure 1.** CBP depletion is required to confer resistance to MYBMIM in AML cells. (**A**) Schematic of the genome-wide CRISPR screen to identify genes whose loss modifies MYBMIM effects in MOLM13 cells expressing Cas9. Cells were transduced with the GeCKOv2 library expressing single sgRNAs at low multiplicity of infection, followed by 3-day treatment with 10 μM MYBMIM versus PBS control, with the sgRNA representation assessed by DNA sequencing. (**B**) Volcano plot showing the relative abundance of cell clones expressing sgRNAs targeting specific genes (fold change of MYBMIM-treated cells at T1 versus T0) and their statistical significance from biological replicates. Dashed line represents no enrichment, with positive values representing genes whose depletion confers relative resistance to MYBMIM. CBP is marked in red.

The online version of this article includes the following source data and figure supplement(s) for figure 1:

**Source data 1.** Genome-wide CRISPR knockout screen (GeCKO) gene summary.

**Figure supplement 1.** CBP depletion is dispensable in AML cells.

proximal to but non-overlapping with the MYB-binding site, termed CREBMIM and MLLMIM, respectively (*Figure 2B and C* and *Figure 2—figure supplement 1A–C*).

As expected, both CRYBMIM and CREBMIM bind to the purified recombinant CBP KIX domain with significantly improved (three– to six-fold) affinities as compared to MYBMIM and measured by microscale thermophoresis ($K_d$ of 5.7 ± 0.2 μM, 2.9 ± 0.7 μM, and 17.3 ± 1.6 μM, p=1e-15; *Figure 2D*). To confirm these peptides can bind the CBP/P300 complex from cells, we immobilized biotinylated CRYBMIM and CREBMIM peptides on streptavidin beads and used them to affinity purify CBP/P300 from non-denatured nuclear extracts of MV411 AML cells. Consistently, we observed efficient binding of nuclear CBP/P300 to peptide-conjugated but not control streptavidin

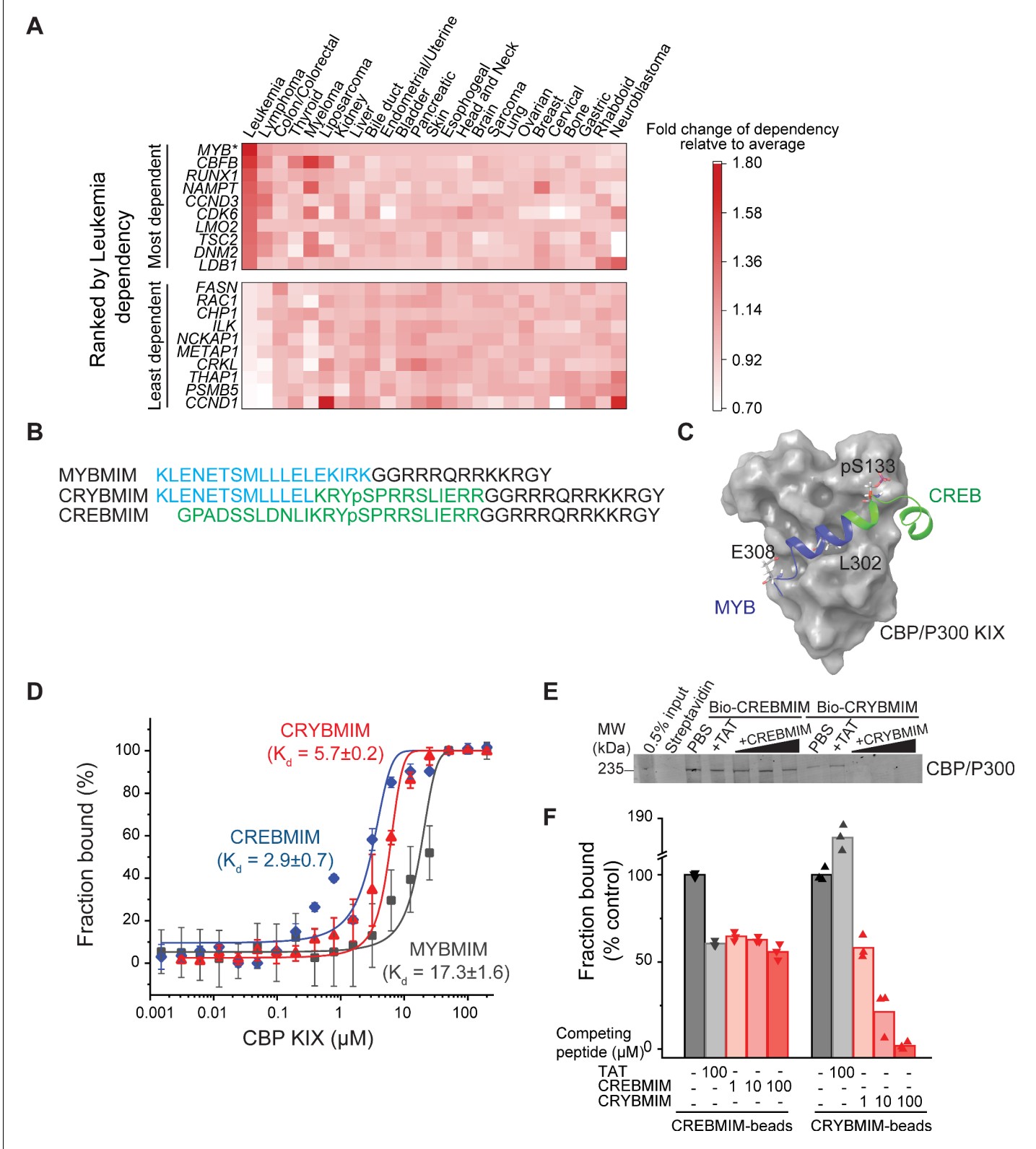

**Figure 2.** CRYBMIM is an improved peptidomimetic chimera that specifically binds the KIX domain of CBP/P300 in vitro and in cells. (**A**) Heatmap of the top 10 and bottom 10 gene dependencies for survival and proliferation of 652 cancer cell lines in the DepMap Cancer Dependency Map Project, ranked by the greatest dependency for 37 leukemia lines, 20 of which are AML cell lines, as indicated by the red color gradient; *p=1.1e-15, t-test of leukemia versus other tumor types. (**B**) Retro-inverso amino acid sequences of MYBMIM, CREBMIM and CRYBMIM, with amino acids derived from MYB,

*Figure 2 continued on next page*

*Figure 2 continued*

CREB, and TAT marked in blue, green, and black, respectively. (C) Molecular model of the CRYBMIM:KIX complex. Residues making contact with KIX are labeled, with portions derived from MYB and CREB marked in blue and green, respectively. (D) Binding of FITC-conjugated CREBMIM (blue), CRYBMIM (red), and MYBMIM (black), as measured using microscale thermophoresis; $K_d$ = 2.9 ± 0.7, 5.7 ± 0.2, and 17.3 ± 1.6, respectively. Error bars represent standard deviations of three biological replicates; p<1e-15, ANOVA, for CRYBMIM versus MYBMIM. (E) Western blot showing binding of nuclear CBP/P300 isolated from AML cells to biotinylated CRYBMIM or CREBMIM, specifically competed by the excess free peptides as indicated. (F) Quantification of CBP/P300 binding to CRYBMIM and CREBMIM by fluorescence densitometry, with black, gray, and red denoting PBS control, TAT control, and peptide competition, respectively.

The online version of this article includes the following figure supplement(s) for figure 2:

**Figure supplement 1.** Molecular models of CREB and MLL structures in complex with CBP KIX, and CRYBMIM selectively binds to KIX domain in CBP/ P300.

beads (*Figure 2E*). We found that increasing concentrations of free CRYBMIM could compete with the binding of CBP/P300 to immobilized CRYBMIM, with the apparent $K_i$ of approximately 4.7 µM based on quantitative fluorescence densitometry measurements (*Figure 2F*). We confirmed the specificity of this binding by incubating bound complexes with 100-fold excess of TAT control peptides, which demonstrated no measurable displacement as compared to PBS control (*Figure 2E–F*). In contrast, excess CREBMIM was significantly less effective at displacing CBP/P300 from immobilized CREBMIM (apparent $K_i$ >100 µM, *Figure 2E–F*), consistent with the much higher nM affinity and allosteric effects that characterize the CREB:CBP/P300 interaction (*Goto et al., 2002*; *Radhakrishnan et al., 1997*). This is also consistent with the presence of distinct CBP transcription factor complexes in AML cells, some of which (MYB) are susceptible to peptidomimetic blockade, and others (CREB) exhibiting more stable interactions. Importantly, CRYBMIM binds CBP/P300 specifically, as exposure of AML cell extracts to streptavidin-immobilized biotinylated CRYBMIM leads to efficient binding to CBP/P300, but not MED15 which is highly expressed in MV411 AML cells and contains a known KIX domain with the closest sequence similarity to CBP/P300 (38% identity; *Figure 2—figure supplement 1D–E*). Thus, CRYBMIM is a specific high-affinity inhibitor of MYB:CBP/ P300 binding.

## Potent and broad-spectrum activity of CRYBMIM against diverse subtypes of AML

To confirm that CRYBMIM maintains effective cell penetration and nuclear accumulation in AML cells, we studied its fluorescein isothiocyanate (FITC)-conjugated derivative using live cell confocal fluorescence microscopy (*Figure 3A*). Consistently, we observed that both FITC-CRYBMIM and FITC-CREBMIM efficiently localized to the nuclei of MV411 AML cells within 1 hr of peptide treatment (*Figure 3A*). We confirmed specific nuclear accumulation of CRYBMIM as opposed to non-specific membrane binding that can affect TAT-containing peptides by confocal subcellular imaging of cells co-stained with specific mitochondrial and nuclear dyes (*Figure 3A*).

Previously, we observed that MYBMIM blocks the binding of MYB to CBP/P300 in AML cells, requiring relatively high 20 µM concentrations for 3 hr to achieve this effect (*Ramaswamy et al., 2018*). To ascertain whether CRYBMIM can achieve more potent interference with the binding of MYB to the CBP/P300 complex in cells due to its improved affinity as compared to MYBMIM (*Figure 2D*), we treated MV411 cells with 10 µM peptides for 1 hr and immunoprecipitated CBP/P300 using specific antibodies (*Figure 3B*). Under these more stringent conditions, we found that CRYBMIM is indeed more potent compared to MYBMIM, as evidenced by the substantial depletion of MYB from the immunoprecipitated CBP/P300 complex by CRYBMIM but not MYBMIM under these conditions (*Figure 3B*).

This effect was specific because the inactive analogue of CRYBMIM, termed CG3, in which three key residues have been replaced with glycines (*Supplementary file 1a*), was unable to compete with MYB:CBP/P300 binding in cells, an effect observed with CBP/P300-specific but not control isotype non-specific antibodies (*Figure 3B*). Neither CREBMIM nor CRYBMIM treatment interfered with the binding of CREB to the cellular CBP/P300 complex (*Figure 3B*), in agreement with the affinities of their direct binding to the recombinant CBP KIX domain (*Figure 2D–F*), the nM affinity of CREB: CBP/P300 interaction (*Radhakrishnan et al., 1997*), and the specific molecular features required for MYB but not CREB or MLL1 binding (*Figure 2C* and *Figure 2—figure supplement 1*).

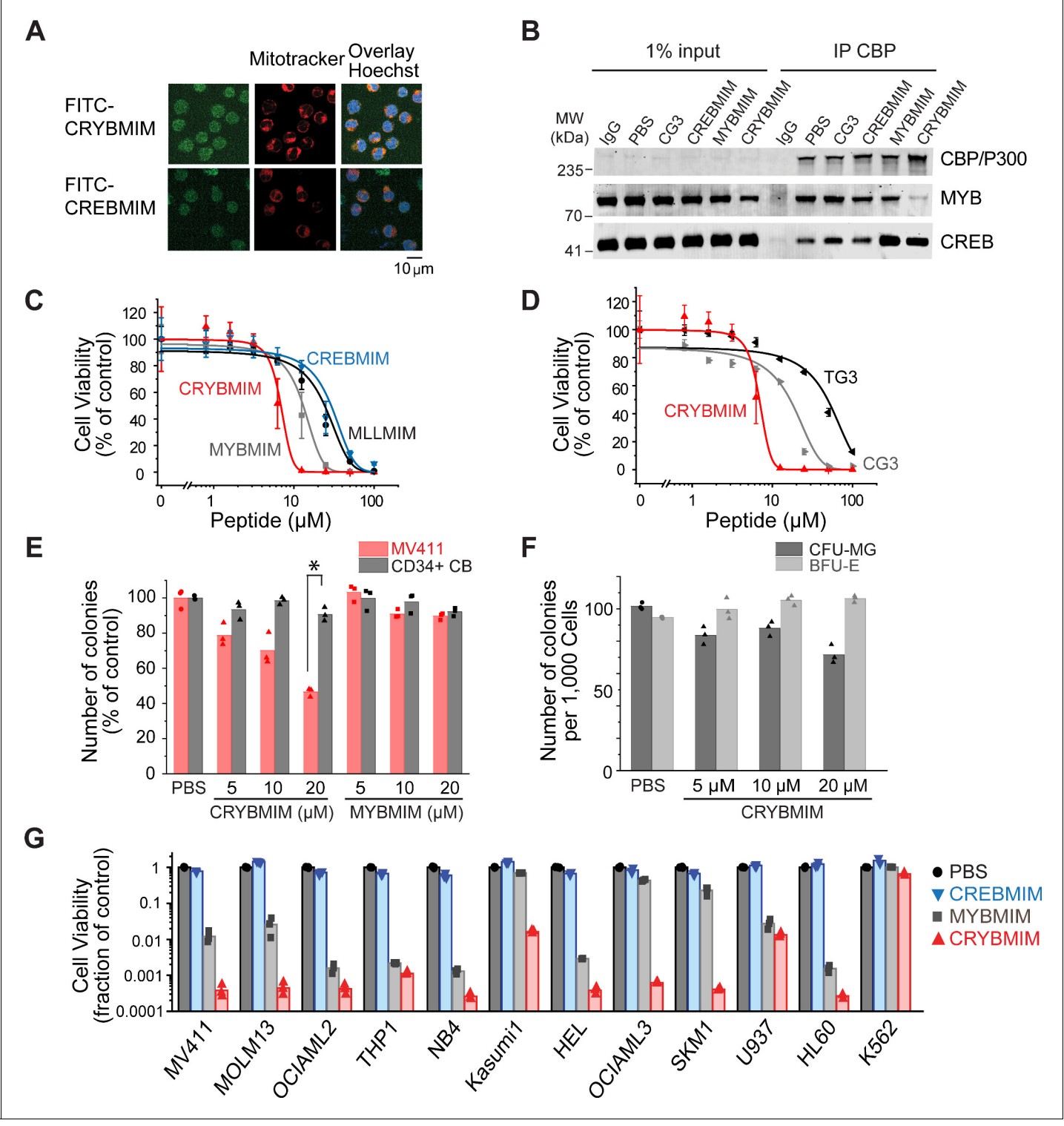

**Figure 3.** Potent and broad-spectrum activity of CRYBMIM against diverse AML cell lines but relatively sparing of normal hematopoietic progenitor cells. (**A**) Representative live cell confocal microscopy images of MV411 cells treated with 100 nM FITC-conjugated peptides as indicated for 1 hr, counterstained with Mitotracker (red) and Hoechst 33342 (blue). Scale bar indicates 20 µm, with z-stack of 1.5 µm. (**B**) Western blots showing immunoprecipitated nuclear CBP/P300 co-purified with MYB and CREB from MV411 cells treated with 10 µM peptides as indicated for 1 hr. (**C–D**) Cell viability of MV411 cells as a function of increasing concentrations of 48 hr peptide treatment, comparing (**C**) CRYBMIM to MYBMIM, CREBMIM and MLLMIM ($IC_{50}$ = 6.88 ± 3.39, 13.1 ± 3.3 29.15 ± 3.79, and 24.22 ± 2.00, p<1e-15, ANOVA); (**D**) TG3 and CG3 ($IC_{50}$ = 16.65 ± 1.00 and 48.91 ± 2.55, p<1e-15, ANOVA). Error bars represent standard deviation of three biological replicates. (**E**) Colony forming ability of CD34+ cells isolated from human
*Figure 3 continued on next page*

*Figure 3 continued*

umbilical cord blood (CD34+ CB, gray) and MV411 AML (red) cells following CRYBMIM or MYBMIM treatment. Data represent three biological replicates; *p=7.4e-5, t-test of normal CD34+ CB versus MV411 AML cells. (F) Preservation of clonogenic capacity of CD34+ CB cells in differentiating into erythroid blast forming units (BFU-E, light gray) and granulocyte macrophage colony forming units (CFU-GM, gray) as a function of CRYBMIM treatment. (G) Cell viability of AML cell lines treated with control PBS (black), 20 µM CREBMIM (blue), MYBMIM (gray), or CRYBMIM (red) as indicated for 6 days with media replacement every 48 hr in three biological replicates (p=8.6e-3, t-test for CRYBMIM versus control).

The online version of this article includes the following figure supplement(s) for figure 3:

**Figure supplement 1.** CRYBMIM relatively spares normal hematopoietic progenitor cells in vitro.
**Figure supplement 2.** Activity of CRYBMIM on non-hematopoietic cells.

To test the prediction that higher affinity binding of CRYBMIM to CBP/P300 would translate into improved anti-leukemia potency, we assessed its effects on the viability of cultured human leukemia and normal hematopoietic cells. Consistent with this prediction, we observed that CRYBMIM exhibited significantly higher potency against MV411 AML cells, as compared to MYBMIM as well as CREBMIM and MLLMIM ($IC_{50}$ = 6.9 ± 3.4 µM, 13 ± 3.3, 29 ± 3.8 µM, and 24 ± 2.0 µM, p=1e-15; *Figure 3C*). We confirmed CRYBMIM's specificity by analyzing its inactive analogue CG3 and MYB-MIM's inactive analogue TG3, in which three residues forming key electrostatic and hydrophobic interactions with the KIX domain were replaced by glycines; both exhibited significantly reduced activity ($IC_{50}$ of 17 ± 1.0 µM and 49 ± 2.6 µM, p=2.75e-4 and <1e-15, respectively; *Figure 3D*). To assess the effects of CRYBMIM on normal hematopoietic progenitor cells, we used primary human CD34+ umbilical cord progenitor cells, cultured in serum-free medium in methylcellulose, as well as in liquid culture (*Figure 3—figure supplement 1D*). We observed minimal effects on the clonogenic growth of normal progenitor cells, as compared to that of MV411 AML cells which was significantly suppressed (91 ± 3.8% versus 47 ± 2.4% colonies, p=1e-4; *Figure 3E* and *Figure 3—figure supplement 1A*). Extending the duration of treatment led to small reduction of myeloid/granulocyte progenitors, and an increase in erythroid progenitor colony-forming units (70.5 ± 5.6% and 112.3 ± 2.2%, respectively; *Figure 3F*). In contrast, doxorubicin, which is commonly used to treat AML, caused significant and substantial impairments in the clonogenic capacity of all hematopoietic progenitor cells (*Figure 3—figure supplement 1B–C*). Thus, CRYBMIM exhibits improved anti-leukemia activity, while relatively sparing normal blood cells.

Importantly, CRYBMIM achieved significantly improved, logarithmic suppression of growth and survival of most AML cell lines tested, as compared to MYBMIM and CREBMIM (*Figure 3G*). For example, whereas MYBMIM induced nearly 100-fold suppression of growth of MV411 cells after 6 days of treatment in agreement with prior studies (*Ramaswamy et al., 2018*), CRYBMIM achieved more than 1000-fold suppression compared to control (p=8.6e-3; *Figure 3G*), consistent with its improved biochemical affinity (*Figure 2D*). This improved activity of CRYBMIM spanned diverse AML subtypes, including *MLL*-rearranged, *AML1-ETO* translocated, *PML-RARA*-translocated, *DNMT3A*-mutant, *NPM1c*-mutant, *TP53*-mutant, *MYC*-amplified, and *WT1*-mutant cell lines, with the exception of erythroblastic *BCR-ABL1*-translocated K562 cells (10 of 11 cell lines tested; *Figure 3G* and *Supplementary file 1b*). In contrast, under these conditions, CRYBMIM had no significant effects on the growth and differentiation of normal human umbilical cord blood progenitor cells in vitro (*Figure 3E and F* and *Figure 3—figure supplement 1A*). Interestingly, CRYBMIM also exhibited anti-tumor effects on various solid tumor cell lines, including some medulloblastoma, neuroblastoma, and breast carcinoma cells, at least some of which also exhibit high levels of *MYB* expression and genetic dependence (*Figure 3—figure supplement 2A–B*). In all, CRYBMIM has broad-spectrum activity against diverse subtypes of AML, while relatively sparing normal hematopoietic cells.

## CBP is specifically required for susceptibility of AML cells to peptidomimetic MYB blockade

CBP and EP300 are closely related transcriptional coactivators with distinct activities in cells. To precisely define their contributions, we used CRISPR interference to elucidate their requirements for susceptibility to CRYBMIM (*Figure 4A*). We used MV411, MOLM13, OCIAML3, and K562 AML cells that stably express Cas9 and transduced them with mCherry or GFP-expressing lentiviruses encoding specific and independent sgRNAs targeting *CBP* and *EP300*, as compared to the *AAVS1* safe

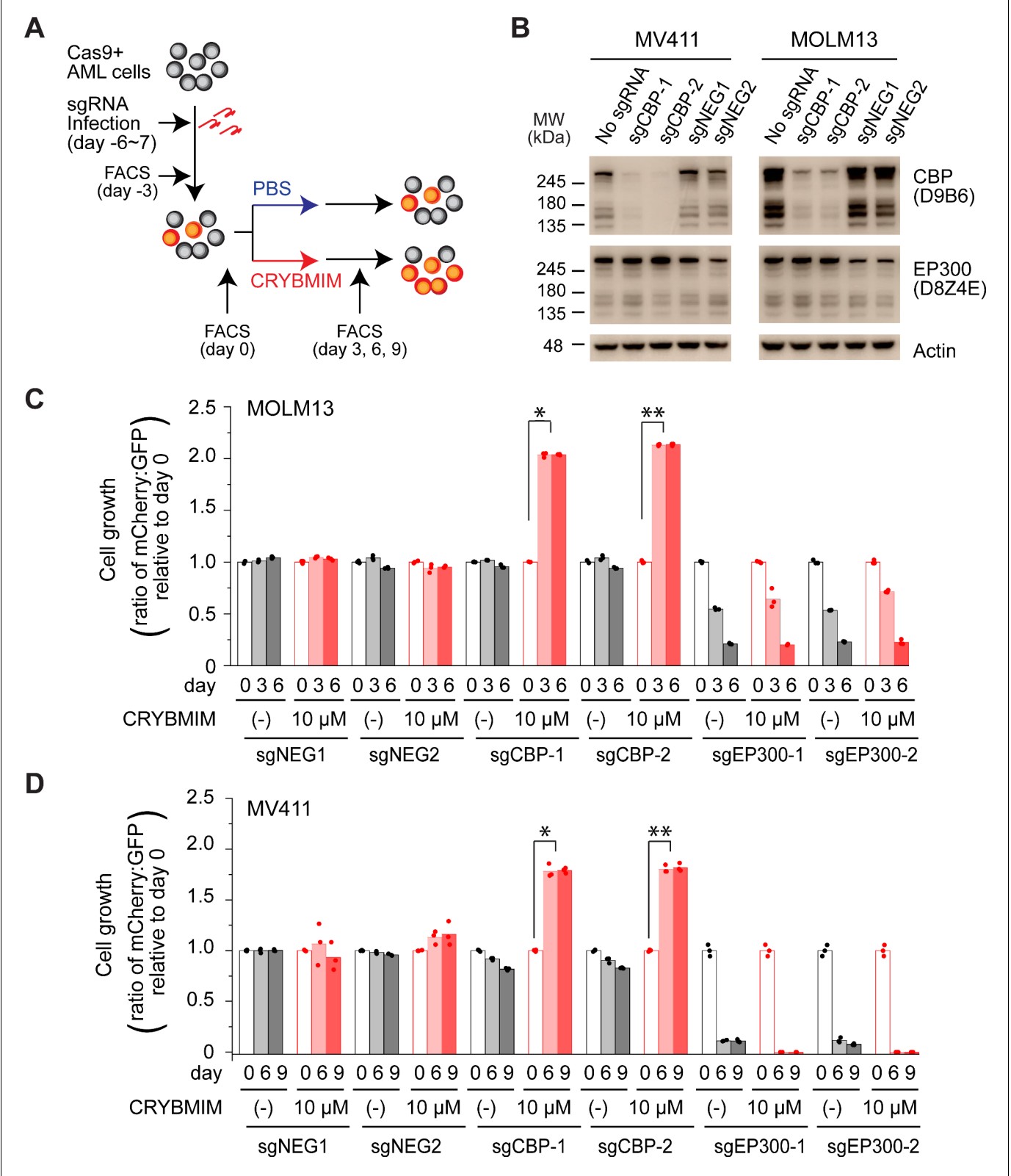

**Figure 4.** CBP but not P300 is dispensable for the growth and survival of AML cells, and is required for the susceptibility to peptidomimetic MYB blockade by CRYBMIM. (**A**) Schematic of the competitive assays to define specific genetic dependencies. AML cells expressing Cas9 and GFP are transduced with sgRNAs targeting specific genes and expressing mCherry, followed by quantitation of cell abundance by fluorescence activated cell scanning (FACS) of GFP and mCherry-expressing cells. (**B**) Western blots demonstrating specific depletion of CBP in MV411 (left) and MOLM13 (right)

*Figure 4 continued on next page*

*Figure 4 continued*

cells expressing sgCBP-1 and sgCBP-2, as compared to control sgNEG1 and sgNEG2 targeting the *AAVS1* safe harbor locus. EP300 is shown for specificity, and Actin serves as the loading control. (**C**) Relative growth of GFP-expressing MOLM13 cells expressing mCherry and unique sgRNAs targeting *AAVS1* control (sgNEG1 and sgNEG2), *CBP* (sgCBP-1 and sgCBP-2), and *P300* (sgEP300-1 and sgEP300-2) and quantified by FACS on day 0, 3, and 6 after 2-day treatment with 10 µM CRYBMIM or PBS. Data represent biological triplicates of at least 10,000 cells per condition; *p=6.0e-10, **p=4.1e-8, t-test for day 6 versus day 0 of CRYBMIM treatment of CBP-depleted cells. (**D**) Analogous experiment as (**C**) using MV411 cells; *p=1.1e-6, **p=3.7e-6, t-test for day 6 versus day 0 of CRYBMIM treatment of CBP-depleted cells.

The online version of this article includes the following figure supplement(s) for figure 4:

**Figure supplement 1.** Genetic dependencies of CRYBMIM susceptibility in AML cells.

harbor locus and *CDK1* that is required for cell survival, as negative and positive controls, respectively. We confirmed specific depletion of the majority of CBP protein by Western immunoblotting in cells expressing sgCBP-1 and sgCBP-2 with intact EP300 expression, but not those not transduced with sgRNA lentiviruses or those expressing sgNEG-1 and sgNEG-2 (*Figure 4B*). We found that depletion of *EP300* caused gradual decrease in cell proliferation over 3–6 days (*Figure 4—figure supplement 1A–C*), whereas depletion of *CDK1* caused acute cell cycle arrest and apoptosis (*Figure 4—figure supplement 1A–C*). In contrast, depletion of *CBP* had no significant effects on steady-state cell fitness, but caused significant resistance to CRYBMIM, where CBP-deficient cells outcompeted their non-transduced counterparts when treated with CRYBMIM (p=6.0e-10 and 4.0e-8 for sgCBP-1 and sgCBP-2 in MOLM13 cells, respectively, and 1.1e-6 and 3.7e-6 for MV411 cells; *Figure 4C–D*). In contrast, K562 cells that are mostly resistant to CRYBMIM exhibited selective fitness upon depletion of EP300 (*Figure 4—figure supplement 1D*). This suggests that CBP and EP300 contribute to distinct gene expression programs, presumably as part of specific transcription factor complexes, as required for their susceptibility to peptidomimetic blockade.

## CRYBMIM blocks oncogenic MYB gene expression and restores normal myeloid cell differentiation

The assembly of MYB with CBP/P300 controls gene expression in part due to its transcriptional co-activation at specific enhancers and promoters (*Kasper et al., 2010*; *Zhao et al., 2011*). Previously, we found that MYBMIM can suppress MYB:CBP/P300-dependent gene expression, leading to AML cell apoptosis that required MYB-mediated suppression of *BCL2* (*Ramaswamy et al., 2018*). However, because MYBMIM's suppression of gene expression was accompanied mostly by apoptosis, we were unable to discern the molecular mechanisms that directly dysregulate the activity of the CBP/P300 transcription factor complex in AML cells.

Given that CRYBMIM has increased affinity for CBP/P300 similar to that of native MYB (5.7 ± 0.2 µM and 4.2 ± 0.5 µM, respectively), we reasoned that its improved activity would now permit detailed kinetic studies to define the specific gene expression programs that are aberrantly activated in AML cells. Consistent with this prediction, comparison of the effects of CRYBMIM on gene expression and consequent apoptosis of MV411 cells revealed that 1 and 4 hr exposures led to significant changes in gene expression with minimal induction of apoptosis (*Figure 5—figure supplement 1*). Thus, we used RNA-sequencing (RNA-seq) to define the changes on gene expression genome-wide upon 1 and 4 hr exposures to CRYBMIM as compared to PBS control (*Figure 5A–B*). We found that after 4-hr duration of treatment, CRYBMIM causes significant downregulation of 2869 genes, including known MYB target genes *MYC*, *IKZF1*, *GATA2*, and *KIT* (*Figure 5A–C*, *Supplementary file 2b-e*). Similar to MYBMIM, CRYBMIM also caused significant upregulation of distinct genes, an effect that was substantially more pronounced upon 4 hr of treatment (4099 genes; 5A-B, *Supplementary file 2b-e*). Interestingly, in addition to the expected suppression of MYB target genes (*Figure 5C*), gene set enrichment analysis also revealed significant induction of myeloid and monocyte differentiation programs (*Figure 5D–E* and *Supplementary file 2f*). These effects were specific because in contrast to CRYBMIM, CREBMIM treatment exhibited minimal changes in gene expression (*Figure 5—figure supplement 1C*), consistent with its inability to disrupt the cellular CBP/P300 complex (*Figure 3B*). For example, CRYBMIM induced significant increases in the AP-1 family transcription factors *FOS* and *JUN*, as well as *IL6* and *CSF1* that control myeloid differentiation (*Figure 5B*; *Gonda et al., 1993*; *Selvakumaran et al., 1992*). Nearly 40% of the genes induced by

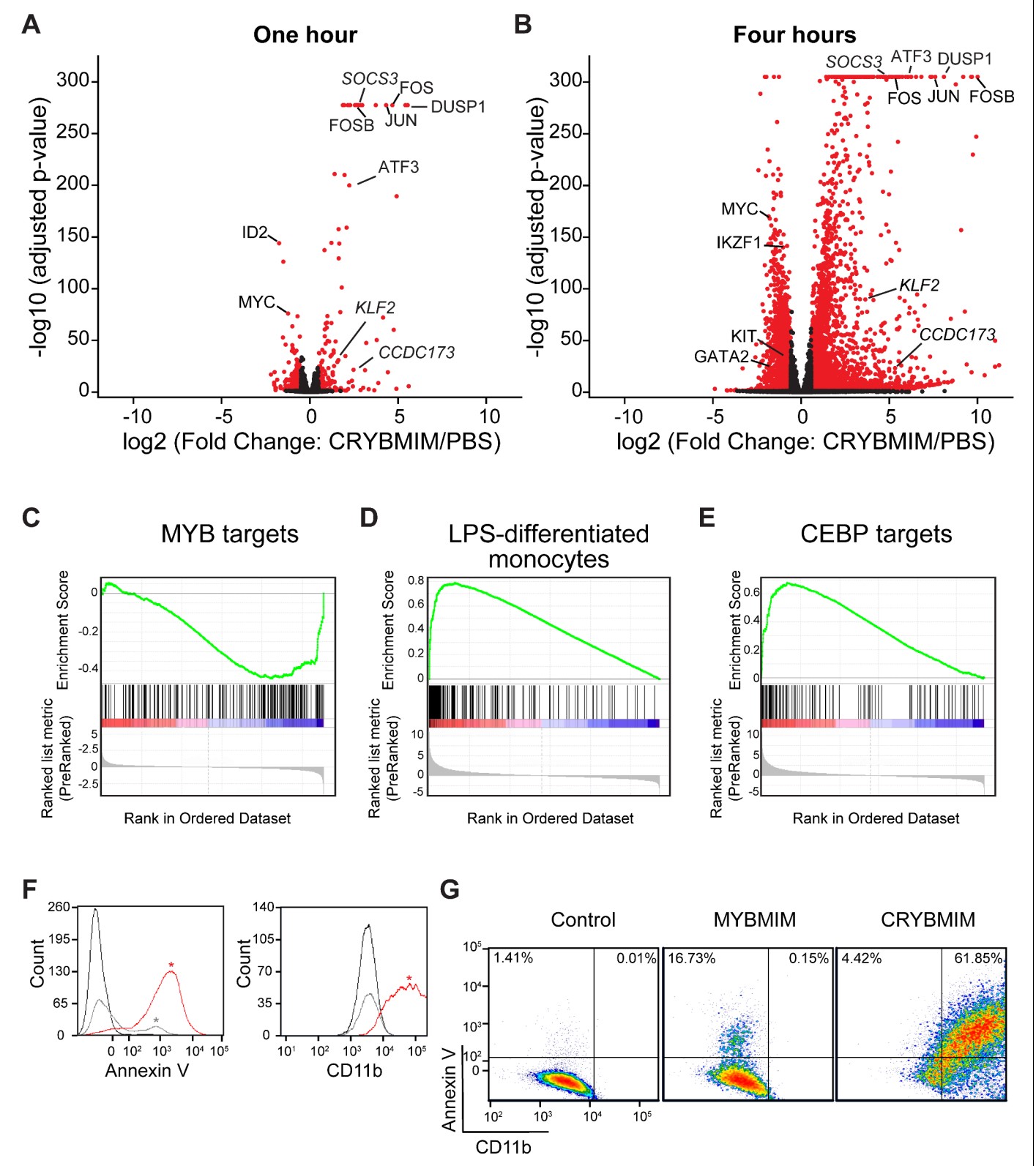

**Figure 5.** CRYBMIM blocks oncogenic MYB gene expression and restores normal myeloid cell differentiation. (**A–B**) Volcano plots of normalized gene expression of MV411 cells upon one (**A**) and four hour (**B**) treatment with CRYBMIM, as compared to PBS control, with select genes labeled; p-values denote statistical significance of three biological replicates. (**C–E**) Gene set enrichment analysis of up- and downregulated gene sets: (**C**) MYB_Q6, (**D**) GSE9988_LPS_VS_VEHICLE_TREATED_MONOCYTES_UP, and (**E**) GERY_CEBP_TARGETS_377. (**F**) Histogram of Annexin V- or CD11b-stained MV411

*Figure 5 continued on next page*

Figure 5 continued

cell fluorescence, treated with CRYBMIM (red), MYBMIM (gray), or control PBS (black); *p<1e-15, Kruskal-Wallis test. (G) Scatter plots comparing Annexin V- versus CD11b-stained MV411 cell fluorescence, treated with control PBS, MYBMIM or CRYBMIM; blue to red color indicates increasing cell density.

The online version of this article includes the following source data and figure supplement(s) for figure 5:

**Source data 1.** Differential gene expression analysis by RNA-seq in CRYBMIM, CREBMIM vs PBS treated MV411 cells (1 hr and 4 hr).

**Figure supplement 1.** CRYBMIM but not CREBMIM causes significant changes in gene expression in AML cells upon short duration exposure.

deletion of RUVBL2 were also found to be induced by CRYBMIM treatment, including *JUN*, *FOS*, and *FOSB* (*Armenteros-Monterroso et al., 2019*). In agreement with these findings, CRYBMIM treatment induced significant phenotypic differentiation of MV411 cells, as evidenced by the induction of monocytic CD11b expression, as measured by flow cytometry (*Figure 5F–G*). While MYBMIM primarily induces apoptosis of MV411 cells, CRYBMIM effects include both apoptosis and differentiation, as evident from flow cytometry analyses (*Figure 5G*). This suggests that differentiation blockade is directly linked to oncogenic MYB-dependent gene expression in AML.

## Sequestration of CBP/P300 contributes to MYB-dependent leukemogenic gene expression

The direct link between leukemogenic gene expression and differentiation blockade suggests that CRYBMIM not only blocks the assembly of the MYB:CBP/P300 complex, but also induces its remodeling to promote AML differentiation. To test this hypothesis, we analyzed the occupancy of MYB and CBP/P300 genome-wide using chromatin immunoprecipitation followed by sequencing (ChIP-seq). Consistently, we found that CRYBMIM treatment leads to significant redistribution of both MYB and CBP/P300 on chromatin (*Figure 6A–D*). For example, after 1 hr of CRYBMIM treatment, both MYB and CBP/P300 are evicted from loci enriched in the MYB-associated DNA-binding motifs (*Figure 6A and C*, lost peaks). By 4 hours of CRYBMIM treatment, MYB was significantly depleted from 2587 promoters and enhancers, primarily at sites associated with the MYB DNA-binding motifs, as well as motifs corresponding to NFY, ETS family, AP-1 family, SP1, and C/EBP family member transcription factors (*Figure 6B and D*). Noticeably, CBP/P300 was not only depleted from 7579 genes, but also significantly redistributed to 11,324 new loci (*Figure 6D*), consistent with the remodeling of its complex upon CRYBMIM treatment. Loci depleted of CBP/P300 were significantly enriched in MYB-associated DNA-binding motifs, as well as those corresponding to the SPI1/ETS, MAF, C/EBP family member, and SP2 transcription factors (*Figure 6D*). Likewise, loci that gained CBP/P300 upon CRYBMIM treatment were enriched in DNA-binding motifs for the AP-1, RUNX1, EGR, and C/EBP family member transcription factors (*Figure 6D*). In all, peptidomimetic blockade of MYB:CBP/P300 assembly causes remodeling of transcription factor complexes at loci controlling leukemic gene expression, associated with genome-wide redistribution of CBP/P300 transcriptional co-activation complexes. Thus, sequestration of CBP/P300 from genes controlling hematopoietic differentiation contributes to the leukemogenic MYB-dependent gene expression.

## MYB assembles aberrant transcription factor complexes in AML cells

Dynamic redistribution of CBP/P300 transcription factor complexes associated with distinct transcription factor activities upon blockade of MYB:CBP/P300 binding suggests that MYB organizes an aberrant transcriptional co-activator complex in AML cells. To define this complex, we used specific antibodies to immunoprecipitate MYB from non-denatured nuclear extracts of MV411 AML cells, and identify co-purifying proteins using high-accuracy quantitative mass spectrometry. To control for abundant proteins and other contaminants that may co-purify non-specifically, we used non-specific isotype control antibodies and precursor ion quantitation to establish stringent statistical parameters that led to the identification of 724 unique proteins that are specifically associated with MYB in MV411 AML cells (*Figure 7A*). This included CBP itself, as confirmed by the high-confidence identification of unique peptide spectra that distinguish CBP from P300, with four additional peptides shared by both CBP and P300 (*Figure 7—figure supplement 1*), as well as other known MYB interactors such as CEBPB (*Supplementary file 3a*).

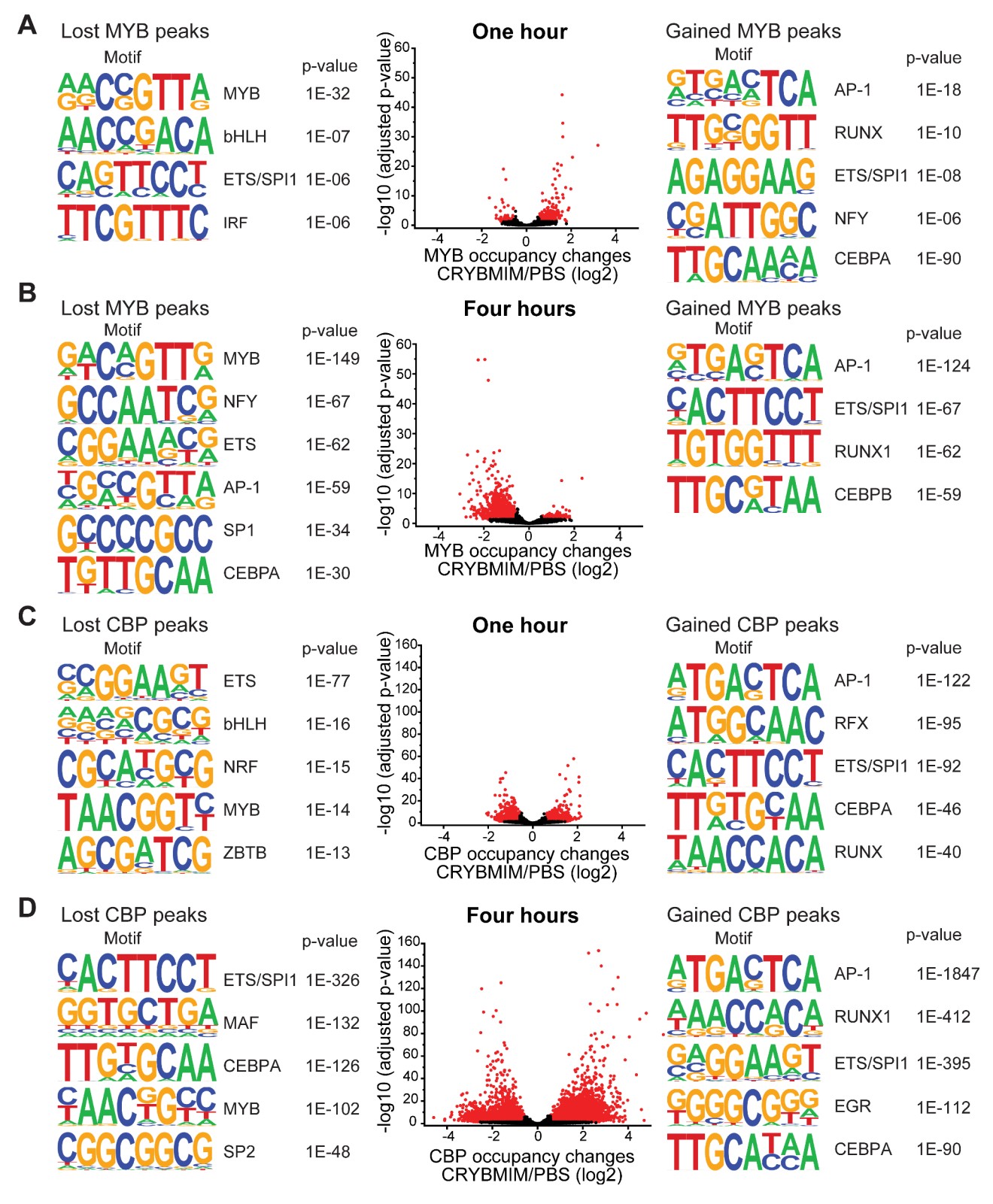

**Figure 6.** CRYBMIM remodels MYB and CBP/P300 chromatin complexes in AML cells. (**A–B**) Volcano plots of relative MYB chromatin occupancy in MV411 cells changes after 1 hr (**A**) and 4 hr (**B**) of 20 μM CRYBMIM treatment compared to PBS control, as analyzed by ChIP-seq. Sequence motifs found in CRYBMIM-induced MYB-depleted (left) and MYB-enriched loci (right) are shown. p-Values denote statistical significance of three technical replicates. (**C–D**) Volcano plots of CBP/P300 chromatin occupancy changes after 1 hr (**C**) and 4 hr (**D**) of 20 μM CRYBMIM treatment as compared to

*Figure 6 continued on next page*

*Figure 6 continued*

control, as analyzed by ChIP-seq. Sequence motifs found in CRYBMIM-induced CBP/P300-depleted (left) and CBP/P300-enriched loci (right) are shown. p-Values denote statistical significance of three technical replicates.

The online version of this article includes the following source data and figure supplement(s) for figure 6:

**Source data 1.** Differentially occupied chromatin loci measured by MYB and CBP/P300 ChIP-seq in CRYBMIM vs PBS treated MV411 cells (1 hr and 4 hr).
**Figure supplement 1.** CRYBMIM remodels MYB chromatin occupancies in AML cells.

Because MYB is required for the growth and survival of diverse AML subtypes, we reasoned that its essential non-redundant co-factors can be identified from the analysis of their functional dependencies, as assessed by genetic CRISPR interference (*Tsherniak et al., 2017*). We assigned the CRISPR dependency score of CBP itself in MV411 cells as the threshold to identify functionally non-redundant MYB:CBP co-factors (*Figure 7B*). We found that these genes encode factors with diverse molecular functions, including a group of 59 chromatin-associated proteins (*Supplementary file 3b*). By using currently annotated protein-protein interactions (*Oughtred et al., 2019*), we constructed their interaction network, based on interactions detected by affinity purifications coupled with either western immunoblotting or mass spectrometry (*Figure 7C*). Analysis of their expression in normal human hematopoietic progenitor as compared to AML cells (*Cancer Genome Atlas Research Network et al., 2013*), led to the identification of candidate co-factors with apparently aberrant expression in AML cells (*Figure 7C*, dark red). Indeed, 10 such factors were selectively required for 37 leukemia but not 615 other non-hematopoietic cancer cells lines (*Figure 7D*). The physical association of MYB, CBFB, ZEB2, C/EBP family members, LYL1, SPI1, RUNX1, LMO2, and GFI1 and their non-redundant functional dependencies in AML cells (*Figure 7A–D*) are in agreement with the chromatin dynamics involving distinct MYB, C/EBP family members, LYL1, SPI1, and RUNX1 DNA-binding motifs observed in CRYBMIM-treated cells (*Figure 6A–B*), associated with the apparent redistribution and remodeling of their CBP/P300 co-activator complexes (*Figure 6C–D*). Indeed, these factors directly associate with the MYB regulatory complex, and their DNA-binding motifs are enriched at loci affected by CRYBMIM treatment (*Figure 6*).

To define the composition of the MYB transcription factor complex across biologically and genetically distinct subtypes of AML, we used co-immunoprecipitation to measure the physical association of MYB and its cofactors in a panel of AML cell lines, spanning representative cell types with relatively high (MV411, HL60, OCIAML2, OCIAML3) and low (U937, Kasumi-1, K562) susceptibility to CRYBMIM (*Figure 8*). We found that in all seven cell lines tested, MYB was physically associated with LYL1 and E2A transcription factors (*Figure 8A*). In contrast, LMO2 was physically associated with MYB in all cell lines except OCIAML2, CEBPA was co-assembled with MYB in OCIAML2 and U937 cells, and SATB1 was co-assembled with MYB in MV411 and HL60 cells (*Figure 8A*). These findings are all in complete agreement with the chromatin dynamics of MYB, as observed using ChIP-seq (*Figure 6*). In addition, we corroborated these findings by examining the association of specific transcription factors with CBP/P300 (*Figure 8B*). The only exception is SPI1/PU.1, which does not appear to be physically associated with MYB or CBP/P300 in examined cell lines, suggesting that the apparent SPI1/PU.1 sequence motifs observed in MYB-associated loci are due to other ETS family factors (*Figure 6*). In all, MYB assembles a convergently organized transcription factor complex in genetically diverse AML cells.

## MYB transcription complexes are associated with aberrant cofactor expression and assembly in AML cells

Insofar as various forms of AML exhibit blockade of normal hematopoietic differentiation induced by distinct leukemia oncogenes, and at least some of the MYB-assembled cofactors have reduced gene expression in normal CD34+ hematopoietic progenitor cells (*Figure 7C*), we reasoned that their assembly in leukemia cells may be due to their aberrant co-expression. To test this, we measured their protein expression in human AML cells using quantitative fluorescent immunoblotting, as compared to normal human CD34+ umbilical cord blood progenitor, adult peripheral blood B- and T-lymphocytes, and monocytes (*Figure 9A*). We found that most transcription factors that are assembled with MYB in diverse AML cell lines could be detected in one or more normal human

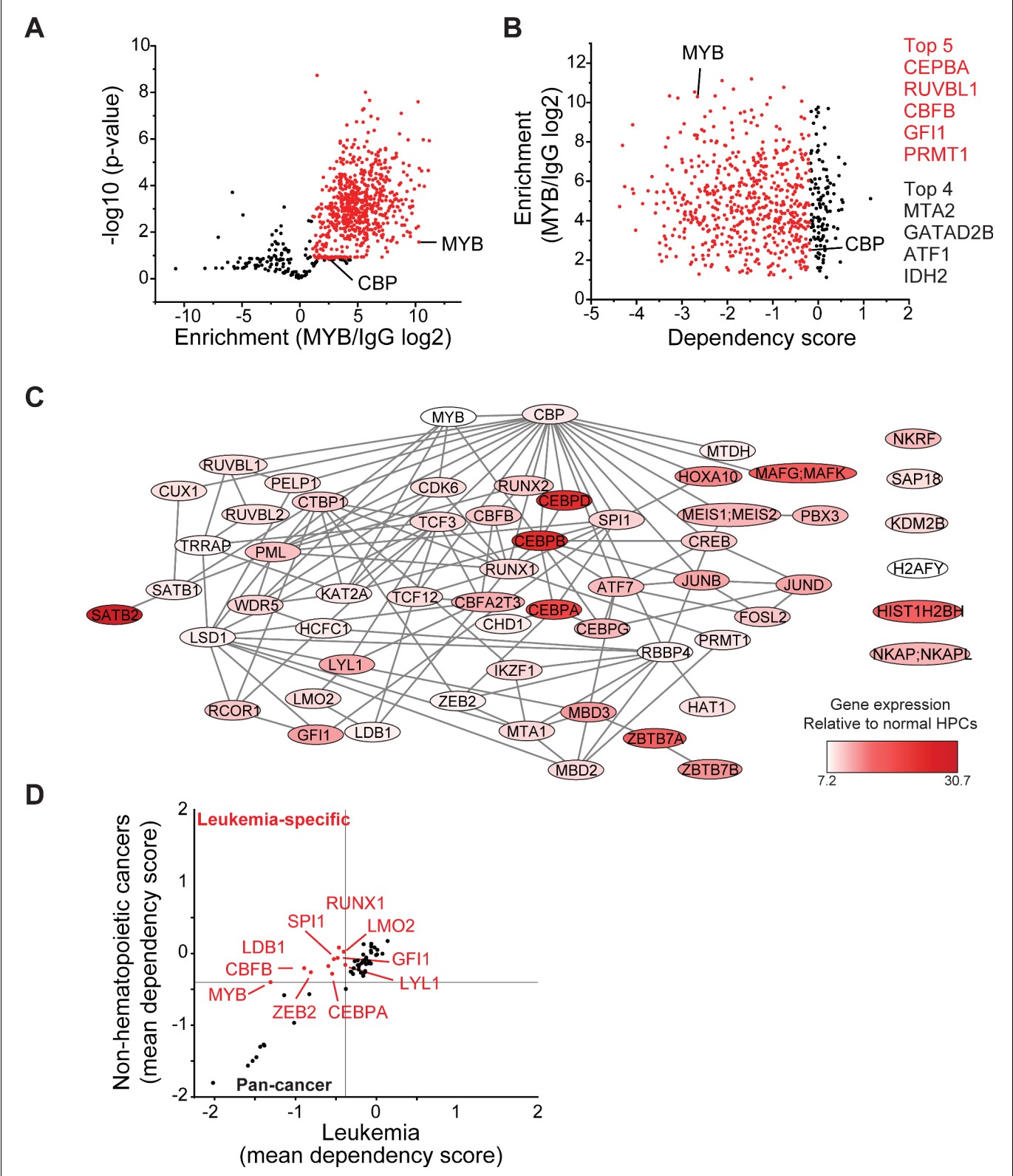

**Figure 7.** MYB assembles aberrant nuclear transcription factor complexes in AML cells. (**A**) Volcano plot of nuclear MYB-associated proteins compared to IgG control, as analyzed by affinity purification-mass spectrometry of MV411 cells. Red symbols denote specifically MYB-associated proteins, as defined by association with CBP (MYB/IgG log$_2$ >1). p-Values denote statistical significance of three biological replicates. (**B**) Enrichment of MYB-associated proteins (red) as a function of their corresponding CRISPR DepMap dependency scores in MV411 cells. Red symbols denote functionally

*Figure 7 continued on next page*

*Figure 7 continued*

required proteins, as defined by the genetic dependency of CBP (score <−0.18). (**C**) Network of BioGRID protein interactions for MYB-associated nuclear AML proteins as a function of their respective hematopoietic expression aberrancy scores, based on their relative gene expression in AML cells as compared to normal bone human bone marrow progenitor cells (white to red color gradient indicates increasingly aberrant gene expression). (**D**) Comparison of the genetic dependency scores in leukemia cell lines as compared to all other non-hematopoietic cancer cell lines for MYB-associated nuclear AML proteins, with red symbols denoting proteins that are required in leukemia as compared to non-hematopoietic cancers.

The online version of this article includes the following source data and figure supplement(s) for figure 7:

**Source data 1.** Lists of proteins identified by IP-MS of IgG control and MYB or CBP/P300 complex purifications from MV411 cell nuclei.
**Figure supplement 1.** CBP is the primary binding partner of MYB in AML cells.

blood cells, albeit with variable abundance, with the exception of CEBPA and SATB1 that were measurably expressed exclusively in AML cells (*Figure 9A*).

To determine whether specific combinations of MYB-assembled transcription factors are associated with the leukemic activity of the MYB complex, we clustered the protein abundance values of distinct groups of transcription factors using principal component analysis (*Figure 9B–D*). We found that this approach exhibited excellent separation between CRYBMIM-sensitive (MV411, OCIAML2, OCIAML3, HL60) and less sensitive (Kasumi1, K562) cell lines, with the first (PC1, 62%) and second (PC2, 19%) eigenvectors explaining more than 80% of the variability in CRYBMIM susceptibility (*Figure 9B*). In particular, the protein abundance levels of CEBPA and LYL1 exhibited the greatest contribution to the observed clustering (*Figure 9C*), consistent with their observed activity in MYB-assembled chromatin dynamics (*Figure 6*).

In addition to aberrant co-expression of various MYB-assembled cofactors, their aberrant assembly in leukemia cells may also contribute to their oncogenic functions. To test this hypothesis, we purified MYB complexes from normal human cord blood progenitor cells using immunoprecipitation, and determined the abundance of specific cofactors as compared to human AML cells using western immunoblotting (*Figure 9E*). While MYB, CBP/P300 and LYL1 were physically associated in normal umbilical cord blood mononuclear cells, we did not observe their physical association with E2A, SATB1 and LMO2. In contrast, MYB and CBP/P300 complexes were highly enriched in LYL1, SATB1, E2A, and LMO2 in MV411 and HL60 AML cells (*Figure 9E*). Thus, oncogenic MYB transcription factor complexes are aberrantly organized in AML cells, associated at least in part with inappropriate transcription factor expression.

If convergent assembly of MYB with common transcription factors in biologically diverse subtypes of AML is responsible for the induction of oncogenic gene expression and blockade of normal hematopoietic differentiation, then pharmacologic blockade of this process would suppress shared gene expression programs associated with AML growth and survival, and promote gene expression programs associated with hematopoietic differentiation. To test this prediction, we carried out comparative gene expression analyses using RNA-sequencing of CRYBMIM effects in *AML1-ETO*-translocated Kasumi1, *DNMTA3A;NPM1*-mutant OCIAML3, *MLL*-rearranged MV411, *NRAS*-mutant;*MYC*-amplified HL60, and *CALM10*-rearranged U937 AML cell lines (*Figure 5A–B* and *Figure 9—figure supplement 1*). Unsupervised clustering of differentially expressed genes exhibited excellent separation between CRYBMIM- and PBS-treated cells. In agreement with the prediction, we observed a shared set of genes that was suppressed in expression upon CRYBMIM treatment of all AML cell lines, such as *MYC* for example (*Figure 5A* and *Figure 9—figure supplement 1A–B*). Similarly, we observed a shared set of genes that was induced by CRYBMIM treatment, including numerous genes associated with hematopoietic differentiation such as *FOS*, *JUN*, and *ATF3*, as well as *SERPINE*, *KLF6*, *DDIT3*, and *NFKBIZ* (*Figure 5A–B* and *Figure 9—figure supplement 1B*). Thus, oncogenic gene expression in biologically diverse subtypes of AML involves convergent and aberrant assembly of MYB transcription factor complexes that induce genes that promote leukemogenesis and repress genes that control cellular differentiation.

## Aberrant organization of the MYB transcription factor complex is regulated by proteolysis

We noted that MYB:CBP/P300 binding in AML cells was reduced by several orders of magnitude upon CRYBMIM treatment (*Figure 6*). This contrasts to the nearly equal binding affinities of

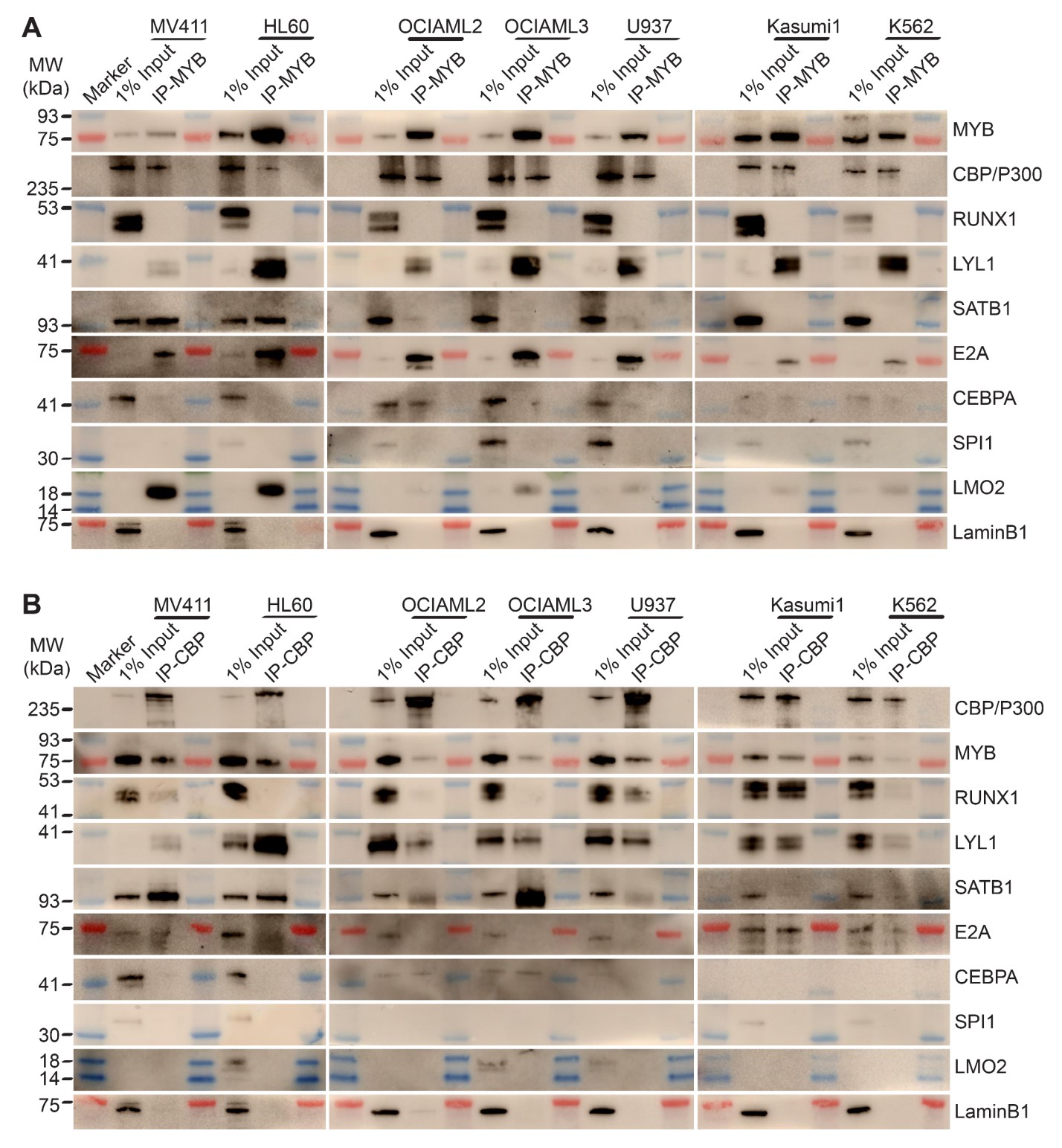

**Figure 8.** MYB and CBP/P300 assemble convergently organized nuclear transcription factor complexes in genetically diverse AML cells. (**A–B**) Western blots of specific transcription factors in immunoprecipitated MYB (**A**) and CBP/P300 (**B**) nuclear complexes in seven genetically diverse AML cell lines, as indicated. Blue and red bands indicate molecular weight markers. LaminB1 serves as the loading control.

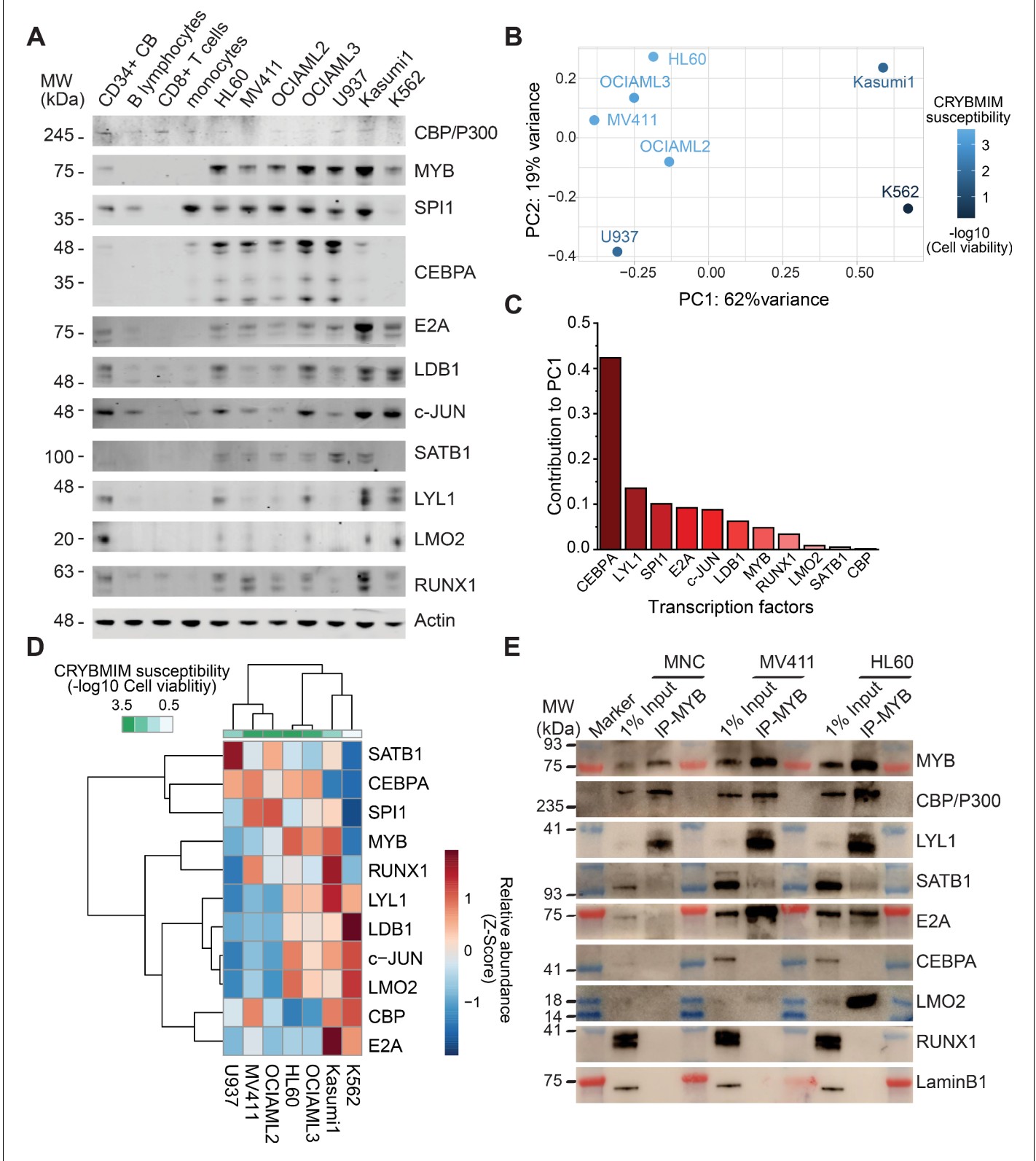

**Figure 9.** Specific MYB complex factors are aberrantly expressed and assembled in AML as compared to normal human blood cells. (**A**) Western blots of specific MYB complex transcription factors in normal human blood cells and genetically diverse leukemia cells, as indicated. Actin serves as the loading control. (**B**) Principal component analysis of MYB complex transcription factor abundance, as quantified by image densitometry, as a function of susceptibility of various AML cell lines to CRYBMIM (blue color index). (**C**) Contribution of individual MYB complex transcription factor abundance to the

*Figure 9 continued on next page*

*Figure 9 continued*

top PCA eigenvector. (D) Heatmap of hierarchical clustering of MYB complex individual transcription factor abundance and CRYBMIM susceptibility. (E) Western blots of specific transcription factors in specific MYB nuclear complexes immunoprecipitated from normal human umbilical cord mononuclear cells (MNC), as compared to MV411 and HL60 AML cells. Blue and red bands indicate molecular weight markers. LaminB1 serves as loading control. The online version of this article includes the following source data and figure supplement(s) for figure 9:

**Source data 1.** All coding gene expression changes measured by RNA-seq in 1 hr CRYBMIM vs PBS treated 5 AML cell lines.
**Figure supplement 1.** MYB transcription complexes induce shared and repress distinct gene expression programs in genetically diverse AML cells, as remodeled by peptidomimetic CRYBMIM inhibition.

CRYBMIM and native MYB to CBP/P300 KIX domain (*Figure 2D*), suggesting that cellular processes must contribute to the biologic effects of CRYBMIM. Thus, we examined cellular MYB protein levels using quantitative immunoblotting (*Figure 10A*, *Figure 10—figure supplement 1*). We found that CRYBMIM treatment induced nearly 10-fold reductions in cellular MYB protein levels with exponential kinetics on the time-scale of 1–4 hr in CRYBMIM-sensitive MV411, HL60, OCIAML2, and OCIAML3 cells. In contrast, CRYBMIM treatment induced less pronounced depletion of MYB in U937, Kasumi1, and K562 cells that are less sensitive to CRYBMIM. Susceptibility of diverse AML cell lines to CRYBMIM, as measured by cell viability (*Figure 3G*), was significantly correlated with the apparent kinetics of MYB protein decay (Pearson $r = 0.94$; *Figure 10B*).

Rapid reduction of MYB protein levels by CRYBMIM is consistent with proteolysis. To investigate this directly, we quantified CRYBMIM-induced reduction of MYB protein levels in MV411 cells upon co-treatment with the proteosomal/protease inhibitor MG132 (*Figure 10C*). Consistent with the proteolytic depletion of MYB upon CRYBMIM treatment, MG132 co-treatment led to near complete rescue of this effect (*Figure 10C*). This effect was specific because overexpression of *BCL2*, which blocks MYBMIM-induced apoptosis (*Ramaswamy et al., 2018*), and rescued the depletion of cellular CREB, presumably due to non-specific proteolysis that accompanies apoptosis, was unable to rescue CRYBMIM-induced proteolysis of MYB (*Figure 10C*). Since CBP is required for the anti-leukemic effects of CRYBMIM, we queried whether MYB protein levels are affected by CBP depletion. We observed no measurable differences in MYB protein levels between wild-type, CBP-deficient or control *AAVS1*-CRISPR targeted MV-411 cells (*Figure 10—figure supplement 1C*). This suggests that either P300 can compensate for CBP-mediated functions, and/or MYB proteolysis is not regulated solely by its interaction with CBP/P300, but requires specific activities of CBP/P300 induced by CRYBMIM. In all, MYB transcription complexes are regulated by specific factor proteolysis in AML cells and can be induced by its peptidomimetic blockade.

## Release and redistribution of MYB-sequestered transcription factors restores normal myeloid differentiation

We reasoned that the remodeling of MYB regulatory complexes and their associated chromatin factors such as CBP/P300 are responsible for the anti-leukemia effects of CRYBMIM, at least in part via reactivation of cellular differentiation of MV411 AML cells (*Figure 5*). To test this, we prioritized CEBPA, LYL1, SPI1, and RUNX1 as MYB-associated co-factors based on their physical interactions and functional dependencies in AML cells (*Figure 7*), and analyzed their chromatin dynamics in response to CRYBMIM treatment using ChIP-seq analysis. Consistent with the release and redistribution mechanism, we observed both coherent and factor-specific dynamics of MYB co-factors on chromatin upon CRYBMIM treatment (*Figure 11A*). Clustering of observed dynamics revealed nine classes of apparent chromatin responses (*Figure 11B*). Approximately one-third of the affected genes lost both MYB and CBP/P300 in response to CRYBMIM, as well as RUNX1, LYL1, and/or CEBPA (*Figure 11B*; yellow clusters 1, 3, and 6). Genes in the MYB and CBP/P300-depleted clusters 1, 3, and 6 were enriched in those associated with the development of hematopoietic progenitor cells, as well as MYC and HOXA9/MEIS1 targets (*Supplementary file 3c*), consistent with current and past gene expression profiling studies (*Figure 5* and *Figure 9—figure supplement 1*). In addition, these genes were enriched in pathways involving chromatin repression, consistent with the enrichment of DNA-binding sequence motifs of transcriptional repressors YY1 and REST/NRSF (*Figure 11B*). This suggests a potential mechanism for the long-hypothesized repressive functions of MYB. While we found no apparent changes in JUN occupancy, motif analysis at loci that both lost

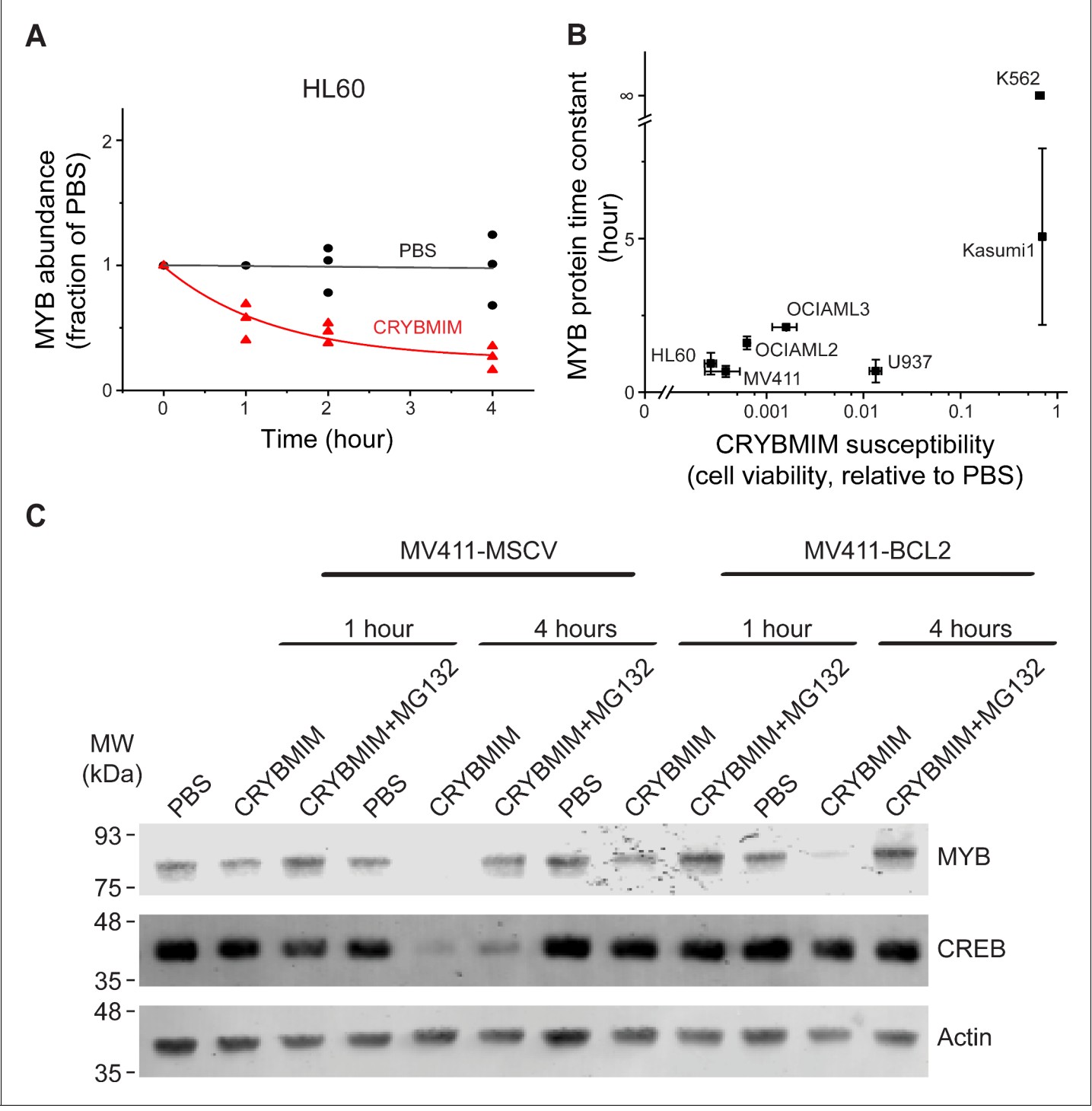

**Figure 10.** Peptidomimetic remodeling of MYB transcriptional complexes leads to rapid MYB proteolysis. (**A**) Quantification of MYB abundance in HL60 cells as a function of duration of 20 µM CRYBMIM treatment (red) as compared to PBS control (black) using Western blot image densitometry. Lines represent single exponential decay fits. Western blots and fits for all cell lines studied are shown in *Figure 10—figure supplement 1*. Symbols represent biological triplicates. (**B**) MYB protein half-life, as estimated by exponential decay kinetics, in genetically diverse AML cell lines as a function of CRYBMIM susceptibility (Pearson *r* = 0.94, excluding resistant K562). Horizontal bars represent standard deviation of CRYBMIM susceptibility. Vertical bars represent standard deviation of time constants. (**C**) Western blots of MYB and CREB in MV411 AML cells transduced with MSCV retroviruses encoding *GFP* control (MV411-MSCV) or *BCL2* (MV411-BCL2), treated with 20 µM CRYBMIM or PBS control with or without 10 µM of MG132 for 1 or 4 hr, as indicated. Actin serves as loading control.

The online version of this article includes the following figure supplement(s) for figure 10:

*Figure 10 continued on next page*

*Figure 10 continued*

**Figure supplement 1.** CRYBMIM causes MYB proteolysis with differential degradation rates in AML cells.

and gained MYB and CBP/P300 revealed enrichment of AP-1 sequence elements, consistent with the presence of other AP-1 family member(s) in MYB regulatory complexes. Minor apparent contribution of MYB:CBP/P300-independent chromatin dynamics involved genes enriched in MLL targets (*Figure 11B*, *Supplementary file 3c*; blue clusters 5, 8), consistent with the activity of MLL fusion proteins in MV411 cells.

Notably, the two chromatin dynamics clusters 4 and 9 that gained both MYB and CBP/P300 in response to CRYBMIM, associated with the accumulation of CEBPA, RUNX1, and/or CREB, were enriched in genes controlling myeloid differentiation programs (*Figure 11B*; orange, and *Supplementary file 3c*). This is consistent with the CRYBMIM-induced gene expression differentiation programs and accompanying morphologic features of myeloid differentiation (*Figure 5*). While CRYBMIM induces MYB proteolysis, residual MYB can remain bound to chromatin, as evident by its accumulation in specific loci upon CRYBMIM treatment. MYB-binding loci lost upon CRYBMIM treatment showed significant enrichment for known MYB binding motifs, while CRYBMIM-induced MYB peaks did not (p=1e-149 vs 1e-3, 56% vs 5.7% of target sites, respectively). This raises the possibility that DNA-binding affinity of MYB could be regulated by CBP/P300, either by direct effects such as MYB deacetylation upon CRYBMIM treatment, or indirectly via binding with other transcription factors. It is possible that other transcription factors may contribute to oncogenic gene expression in AML cells, such as CREB for example, as evident from their contribution to MYB-independent chromatin dynamics (*Figure 11B*; pink cluster 2). However, this is likely a minor effect, given the relatively modest reprogramming of gene expression by CREBMIM that targets the CREB:CBP/P300 complexes in AML cells (*Figure 5* and *Figure 5—figure supplement 1*).

Globally, the most pronounced feature of MYB complex remodeling is the release of CBP/P300 from genes that are associated with AML cell growth and survival to those that are associated with hematopoietic differentiation. To examine this, we compared relative gene expression as a function of the relative occupancy of CBP/P300 and MYB upon 4 hr of CRYBMIM treatment (*Figure 11—figure supplement 1A–B*). In contrast to the model in which blockade of MYB:CBP/P300 induces loss of gene expression and loss of transcription factor and CBP/P300 chromatin occupancy, we also observed a large number of genes with increased expression and gain of CBP/P300 occupancy (*Figure 11—figure supplement 1A–B*). This includes numerous genes that control hematopoietic differentiation, such as *FOS*, *JUN*, and *ATF3*. In the case of *FOS*, we observed that CRYBMIM-induced accumulation of CBP/P300 was associated with increased binding of RUNX1, and eviction of CEBPA and LYL1 (*Figure 11—figure supplement 1C*).

To confirm directly that peptidomimetic blockade of MYB:CBP/P300 releases CBP/P300 and promotes its association with alternative transcription factors, we specifically immunoprecipitated MYB and CBP/P300 from MV411 AML cells using respective antibodies, and determined their composition by western immunoblotting (*Figure 12A–B*). In agreement with biochemical studies, we observed substantial depletion of CBP/P300 from immunoprecipitated MYB complexes upon CRYBMIM treatment (*Figure 12A*), and of MYB from immunoprecipitated CBP/300 complexes (*Figure 12B*). Similarly, we observed reduced binding of LYL1, SATB1, E2A, LMO2, and CEBPA. In contrast, whereas no detectable RUNX1 was found co-associated with MYB either at baseline or upon CRYBMIM treatment (*Figure 12A*), assembly of RUNX1 with CBP/P300 was increased by more than fourfold upon CRYBMIM treatment, as measured by image densitometry (p=3.4e-4; *Figure 12B–C*).

In all, these results support the model in which the core regulatory circuitry of AML cells is organized aberrantly by MYB and its associated co-factors including LYL1, C/EBP family members, E2A, SATB1 and LMO2, which co-operate in the induction and maintenance of oncogenic gene expression, as presumably co-opted by distinct oncogenes in biologically diverse subtypes of AML (*Figure 13*). This involves apparent sequestration of CBP/P300 from genes controlling myeloid cell differentiation. Thus, oncogenic gene expression is associated with the assembly of aberrantly organized MYB transcriptional co-activator complexes, and their dynamic remodeling by selective

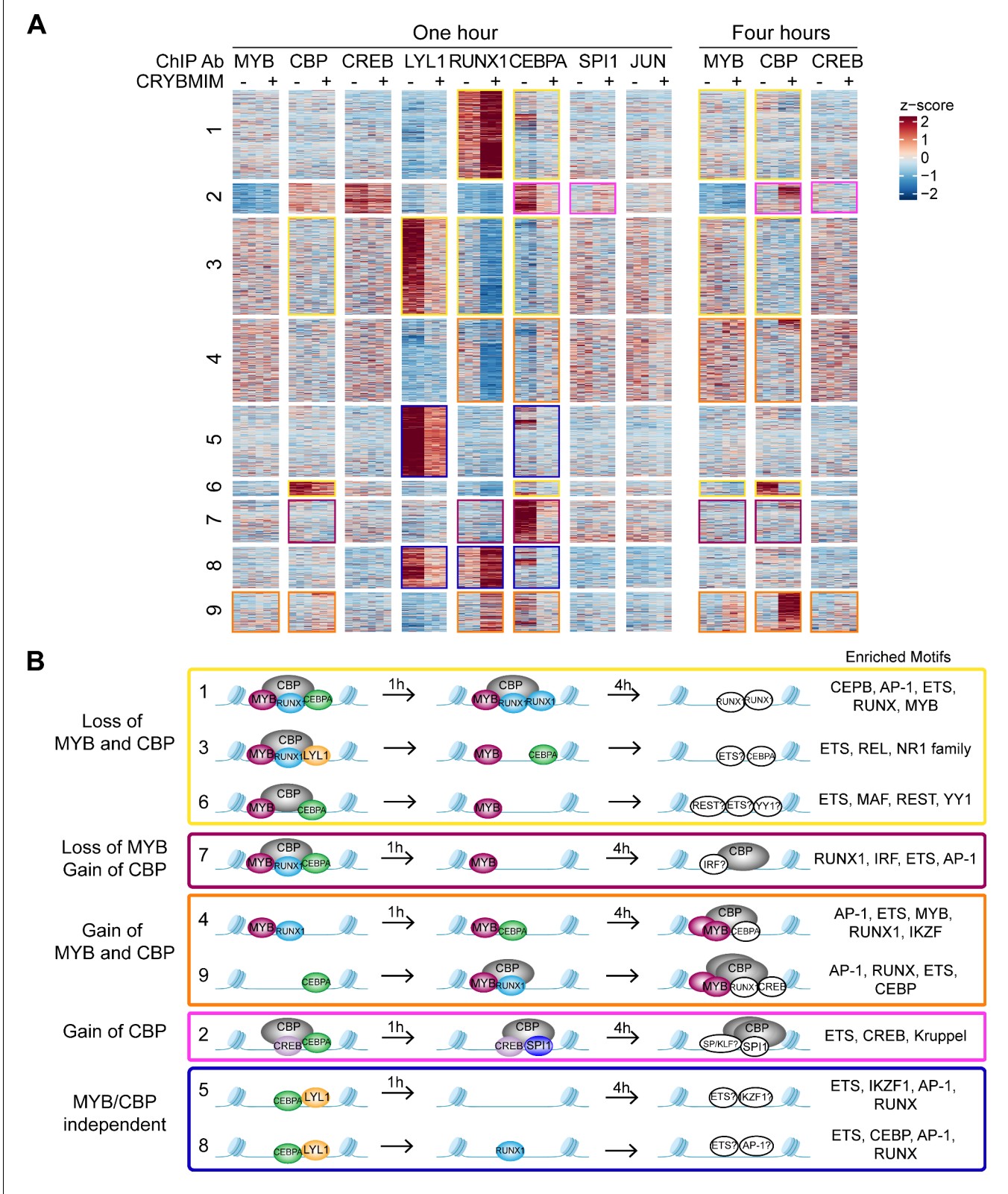

**Figure 11.** Chromatin dynamics of CRYBMIM remodeled MYB transcription factor complexes in AML cells. (**A**) Heatmap of transcription factor chromatin occupancy in MV411 cells as a function of time of control PBS or CRYBMIM treatment. Nine clusters identified using k-means clustering are marked by yellow (loss of MYB and CBP), purple (loss of CBP and gain of MYB), orange (gain of MYB and CBP), pink (gain of CBP), and blue (no apparent changes of MYB and CBP) boxes, based on the similarity of their z-scores, with red and blue representing enrichment or depletion of factors,
*Figure 11 continued on next page*

*Figure 11 continued*

respectively. (B) Groups of clusters comprising similar responses to CRYBMIM treatment based on MYB and CBP/P300 dynamics. Sequence motifs enriched at specific loci for each cluster are listed. Factors in white denote factors presumed to be enriched based on sequence motif enrichment.

The online version of this article includes the following source data and figure supplement(s) for figure 11:

**Source data 1.** Differentially increased or decreased peaks measured by multiple TF ChIP-seq in 1 hr CRYBMIM vs PBS-treated MV411 cells.

**Figure supplement 1.** Oncogenic MYB transcription complexes sequester CBP/P300 to control AML gene expression.

blockade of protein interactions can be leveraged therapeutically to induce AML cell differentiation and apoptosis.

## Discussion

Dysregulated gene expression is a near universal feature of all human cancers. This is particularly relevant for leukemias which are frequently caused by mutations of genes encoding transcription and chromatin remodeling factors. Among all types of leukemias examined to date, the transcription factor MYB ranks as the most selectively required functional genetic dependency. This nominates MYB both as a compelling therapeutic target, and a focus of mechanistic studies to define fundamental mechanisms of dysregulated gene expression in leukemias.

By rapid and selective peptidomimetic interference with the binding of CBP/P300 to MYB, but not CREB or MLL, we find that the leukemic functions of MYB are mediated by CBP/P300-mediated co-activation of a distinct set of transcriptional factor complexes that are aberrantly assembled with MYB in AML cells. The second-generation, cell-penetrant peptidomimetic MYB inhibitor, termed CRYBMIM, has potent and broad-spectrum activity against diverse subtypes of AML, while relatively sparing normal hematopoietic progenitor cells. Consequently, its improved activity enables high-resolution, genome-wide studies of chromatin and gene expression dynamics that control MYB-dependent leukemic expression in AML cells. We find that CRYBMIM blocks oncogenic MYB gene expression and restores myeloid cell differentiation. This effect involves aberrantly organized MYB regulatory complexes, stably composed of additional transcription factors including LYL1, C/EBP family members, E2A, LMO2 and SATB1, that are reminiscent of core regulatory circuits observed in MYB-dependent T-cell lymphoblastic leukemias and other cancers (*Mansour et al., 2014*; *Sanda et al., 2012*). Unexpectedly, we find that MYB-dependent leukemogenic gene expression also involves apparent sequestration of CBP/P300. In turn, peptidomimetic MYB:CBP/P300 blockade releases and redistributes CBP/P300 and other sequestered transcription factors to induce cell differentiation. In all, these findings establish a compelling strategy for pharmacologic reprogramming of oncogenic gene expression that supports its targeting for leukemias and other human cancers caused by dysregulated gene control.

What is the origin of aberrant MYB transcriptional complexes and functions in leukemia cells? MYB is not known to be mutated in most cases of AML, and this study points to its aberrant assembly as the convergent mechanism by which it is pathogenically dysregulated. Indeed, previous studies have found cell-type-specific features of MYB gene activation, suggesting the presence of other factors that influence MYB activity (*Lei et al., 2004*). Furthermore, MYB alone is not sufficient for leukemic cell transformation, indicating the need for specific co-factors in its leukemogenic activity (*Gonda et al., 1989*; *Hu et al., 1991*).

By integrating functional genomics and proteomics, combined with gene expression and chromatin dynamics analyses, we identified a set of factors in complex with MYB that appear to be aberrantly and stably co-assembled, including C/EBP family members, LYL1, E2A, LMO2, and SATB1. Their physical interactions and chromatin co-localization with MYB are associated with oncogenic gene expression and blockade of cell differentiation in AML cells. Interestingly, we found no CRYBMIM-induced remodeling of MYB regulatory complexes independent of CBP/P300, indicating that KIX-dependent interaction between MYB and CBP/P300 is required for most of MYB transcriptional activity in AML cells.

It is possible that somatic mutations of regulatory DNA elements, such as those physically associated with MYB regulatory complexes, contribute to the aberrant assembly of these complexes on chromatin, as observed for the oncogenic *TAL1* enhancer mutations in cases of T-ALL

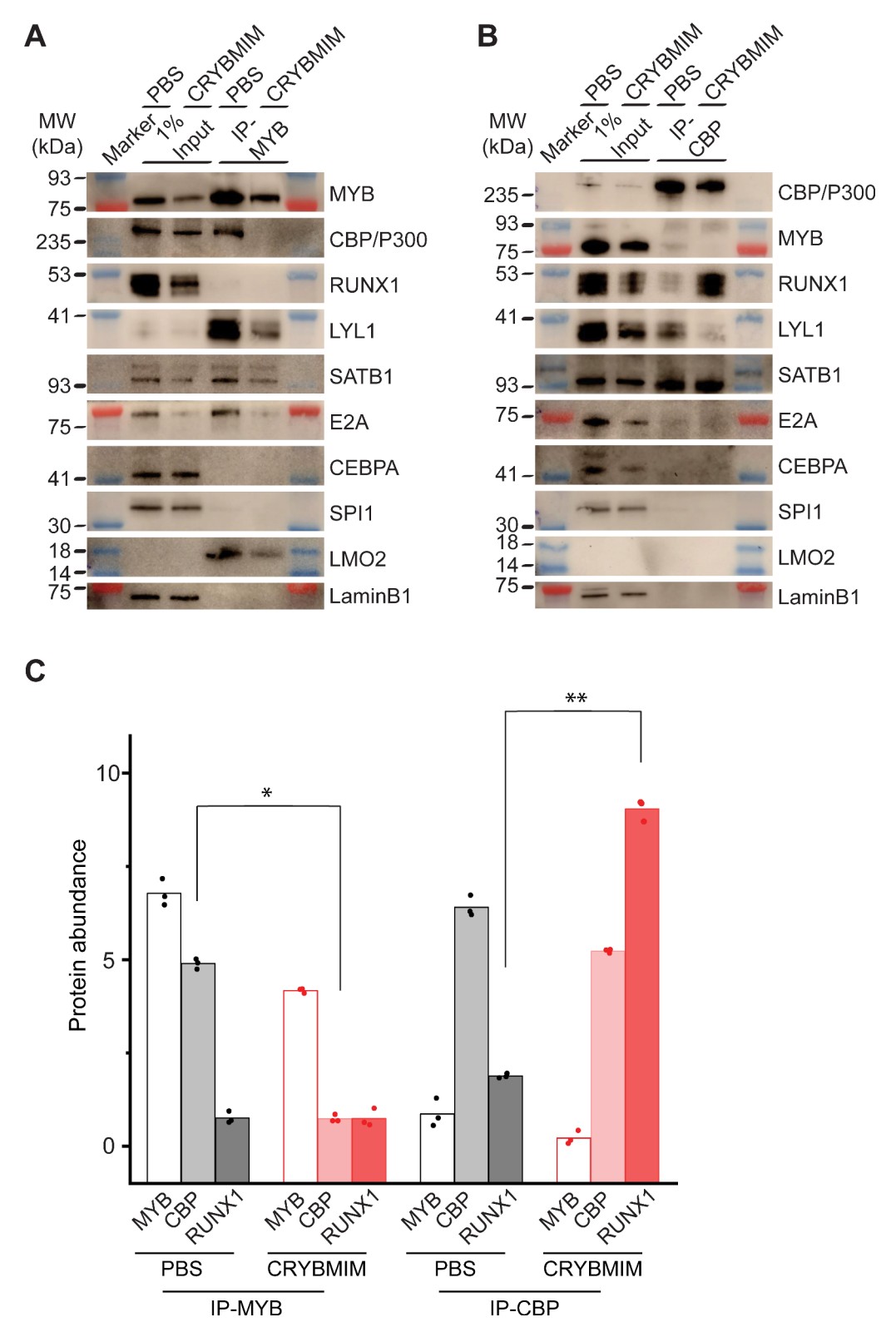

**Figure 12.** Peptidomimetic blockade of MYB:CBP/P300 by CRYBMIM releases CBP/P300 that recruits RUNX1 to chromatin. (A–B) Western blots of specific transcription factors in nuclear complexes with immunoprecipitated MYB (A) and CBP/P300 (B) in MV411 cells upon treatment with 10 μM CRYBMIM or PBS control for 1 hr. Blue and red bands indicate molecular weight markers. LaminB1 serves as loading control. (C) Abundance of MYB, CBP/P300 and RUNX1, as measured by western blot image densitometry, in immunoprecipitated MYB and CBP/300 nuclear complexes in MV411 cells

*Figure 12 continued on next page*

*Figure 12 continued*

treated with CRYBMIM or PBS control. Symbols represent biological triplicates; *p=4.7e-6, t-test for CBP in MYB complex upon CRYBMIM treatment, **p=3.4e-6, t-test for RUNX1 in CBP/P300 complex CRYBMIM treatment.

(*Mansour et al., 2014*), and recently suggested for other leukemias (*He et al., 2019*). It is also possible that leukemic gene expression by MYB involves additional transcriptional co-activators, such as TAF12, as part of the recently described TFIID-SAGA complex interaction (*Xu et al., 2018*), or other TAFs which have also been implicated in transcriptional co-regulation in leukemias (*Jian et al., 2017*; *Xu et al., 2019*). Lastly, it is also possible that distinct subtypes of AML diverge from the aberrant regulatory complex assembly model presented here. For example, LYL1, an oncogene that is aberrantly expressed in diverse subtypes of AML (*Meng et al., 2005*), assembles with MYB in leukemia cells examined in our study, similar to its functional homologue TAL1 in cases of T-ALL (*Mansour et al., 2014*; *Sanda et al., 2012*). Alternative bHLH transcription factors may cooperate with MYB in some leukemia subtypes, including potential differences in their cells of origin (*Jones, 2004*). It will be important to determine how aberrant co-expression of such oncogenic regulatory complex co-factors is induced in leukemia cells by diverse oncogenes, such as for example by kinase-dependent dysregulation of transcription factor assembly recently described for MEF2C and LYL1 (*Brown et al., 2018*; *Tarumoto et al., 2018*; *Vakoc and Kentsis, 2018*).

The switch-like response of AML cells to peptidomimetic disassembly of the MYB:CBP/P300 chromatin complex suggests that cellular CBP/P300 exists in a dynamic equilibrium under limiting conditions. Such a model is supported by the Rubinstein-Taybi syndrome due to heterozygous deletion mutations that reduce *CBP* gene dosage, leading to human developmental defects. This model also explains the distinct requirements of CBP and P300 in normal hematopoiesis and leukemia cell development (*Cheng et al., 2017*; *Wang et al., 2011*), as also observed in our studies in genetically diverse AML cell types, as well as the functional requirement for P300 in CBP-deficient cancers (*Ogiwara et al., 2016*). Definition of the mechanisms of this molecular switch regulating discrete gene expression programs is expected to reveal distinct mechanisms of dysregulated gene control in AML and other transcription-dysregulated cancers.

Current small molecule inhibitors of MYB lack sufficient selectivity (*Pattabiraman and Gonda, 2013*; *Uttarkar et al., 2015*; *Uttarkar et al., 2016*). Our peptidomimetic strategy suggests that structure-based design of effective pharmacologic MYB inhibitors is not only possible, but also desirable given its favorable therapeutic index. First, the functional requirement for peptidomimetic blockade of MYB but not CREB or MLL1 binding in supporting oncogenic gene expression and cell survival suggests that ligands to these permissive binding sites may be used to gain binding affinity of pharmacologic MYB inhibitors. Indeed, such a fragment-based design strategy was successfully used to develop effective BH3 mimetics, including venetoclax that has recently been approved for leukemia therapy.

Quantitative improvement in binding affinity from MYBMIM to CRYBMIM is associated with qualitative improvement in biological potency, due to the combination of enhanced MYB:CBP/P300 binding competition and proteolytic remodeling of its complex. Such event-driven pharmacology has recently been used to develop a variety of pharmacologic modulators of protein interactions, such as PROTACs. The dual mechanism of action observed for peptidomimetic MYB:CBP/P300 inhibitors, involving suppression of oncogenic MYB activity and remodeling of CBP/P300, provides a pharmacologic strategy for both precise chemical probes and improved therapeutics. Indeed, ligation of the KIX domain of CBP/P300 by CRYBMIM may allosterically modulate the activity of its complexes, similar to the allosteric regulation of its acetyltransferase activity by auto-acetylation and intramolecular bromodomain binding.

Functional proteomic maps of MYB regulatory complexes provided by our study should be useful in identifying key protein-protein interactions and post-translational enzymatic modifications that are aberrantly induced in AML cells, as targets for improved therapies. For example, this may involve transcription factor acetylation by CBP/P300 (*Roe et al., 2015*; *Wang et al., 2011*), warranting the investigation of recently developed selective CBP/P300 acetyltransferase inhibitors, which have shown particular activity in hematopoietic cancers (*Lasko et al., 2017*).

Finally, recent studies have found that stable non-genetic resistance is a common feature of relapsed AML, and this resistance at least in part is due to the use of alternative enhancers to sustain

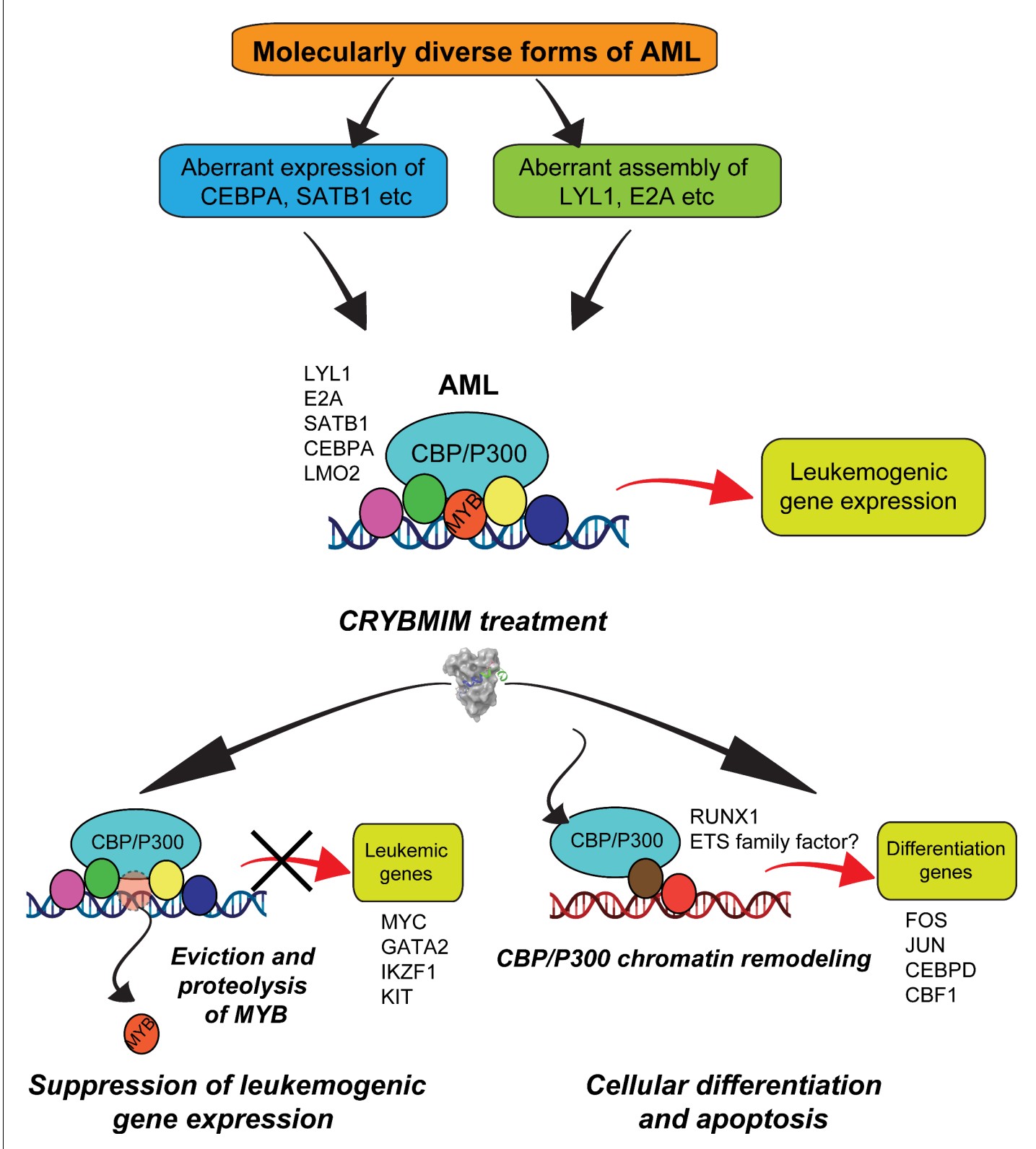

**Figure 13.** Convergent organization of aberrant MYB complexes controls oncogenic gene expression in acute myeloid leukemia. Schematic of the molecular organization of MYB transcription factor complexes, induced convergently in genetically diverse subtypes of AML, leading to oncogenic gene expression that requires MYB:CBP/P300 interaction and causes susceptibility to its peptidomimetic remodeling, leading to MYB chromatin eviction and proteolysis, and CBP/P300 release to induce cellular differentiation and apoptosis.

aberrant gene expression (*Bell et al., 2019*). Therapeutic remodeling of complexes of master regulators such as MYB may constitute an effective strategy to reprogram oncogenic gene expression that may prevent or overcome such resistance, providing a platform for therapy of regulatory complex-mediated gene dysregulation in human cancers. Development and investigation of clinical-grade MYB inhibitors, including improved derivatives of MYBMIM and CRYBMIM, are important directions of future work for patients with MYB-dependent acute myeloid and lymphoblastic leukemias, blastic plasmacytoid dendritic cell neoplasms, gliomas, breast, colon, and adenoid cystic carcinomas.

## Materials and methods

### Reagents
Synthetic peptides were produced by solid phase synthesis, purified by liquid chromatography, and confirmed by mass spectrometry (Tufts University Core Facility). Peptides were dissolved in phosphate buffered saline (PBS) at a concentration of 1 mM, as measured using optical absorbance measurements at 280 nm. Cell culture media was obtained from Corning. All cell lysis buffers were supplemented with protease inhibitors comprised of AEBSF (0.5 mM, Santa Cruz, SC-202041B), bestatin (0.01 mM, Fisher/Alfa Aesar, J61106-MD), leupeptin (0.1 mM, Santa Cruz, SC-295358B), and pepstatin (0.001 mM, Santa Cruz, SC-45036A), and cOmplete, EDTA-free Protease Inhibitor Cocktail (Roche) as required. MG132 was obtained from Cell Signaling Technologies.

### Plasmids
MSCV-IRES-GFP retroviral vector encoding human *BCL2* was a gift from Takaomi Sanda (*Sanda et al., 2013*; *Ramaswamy et al., 2018*). Plasmids used for genome-wide CRISPR screen and CRISPR-competitive assays are described below.

### Cell culture
The cell lines MV411, MOLM13, OCIAML2, THP1, NB4, Kasumi1, HEL, OCIAML3, SKM1, U937, HL60, and K562 were obtained from the American Type Culture Collection (ATCC, Manassas, Virginia, USA) or DSMZ (Leibniz Institute, Braunschweig, Germany). The identity of all cell lines was verified by STR analysis (Integrated Genomics Operation Core Facility, MSKCC) and absence of Mycoplasma contamination was determined using Lonza MycoAlert (Lonza Walkersville, Inc, Walkersville, MD, USA). Cell lines were cultured in 5% $CO_2$ in a humidified atmosphere at 37°C and were maintained in RPMI 1640 medium (Corning) supplemented with 10% fetal bovine serum (FBS) (VWR), antibiotics (100 U/ml penicillin and 100 µg/mL streptomycin), and L-glutamine (1%), referred to as complete media. Human umbilical cord blood was obtained from the New York Blood Center. Human B lymphocytes, CD8+ T cells, and monocytes were obtained from Cellero.

### Genome-wide CRISPR screen
Cas9-expressing MOLM13 cells were established by transduction with LentiV_Cas9_Blast (*Tarumoto et al., 2020*), and then the cells were infected with lentivirus of human CRISPR Knockout GeCKOv2 library (*Sanjana et al., 2014*) with MOI of approximately 0.3 for transduction of single guide RNA per cell. After selection of guide RNA-positive cells by 3 days of puromycin (1 µg/ml) treatment, the cells were divided into three populations, in which guide RNA representation was approximately 500 cells/guide. One of the populations was harvested as T0 sample. The other populations were treated with 10 µM MYBMIM or PBS for 3 days, followed by culture without MYBMIM or PBS for about a week to allow the survived cells to grow, and harvested as T1 sample. Genomic DNA was isolated using phenol/chloroform extraction. Illumina sequencing library was generated by two-step PCR where guide RNA regions were first amplified, followed by the amplification with primers containing sequencing adaptor/barcodes. Barcoded libraries were pooled and analyzed by single-end sequencing using NextSeq 500 (Illumina). MAGeCK tools (*Li et al., 2014*) were used to count guide RNA reads in each sample and to calculate log fold-change and p-value between T0 and T1 samples to identify the guide RNAs enriched in T1 population. Screens were performed in biologic duplicates.

## Molecular modeling

All structures were built using Maestro (Schrodinger). Structures of KIX with MYB, CREB, and MLL peptides were modeled based on PDB IDs 1SBO, 1KDX, and 2LSX, respectively. For CRYBMIM, PDB 1KDX was used as a starting model, replacing CREB amino acids 137–146 with MYB amino acids 298–310. Resulting models were energy minimized using MacroModel (Schrodinger) using the OPLS_2005 force field with implicit water at a dielectric constant of 80.

## Protein expression and purification

CBP KIX was purified as described previously (*Ramaswamy et al., 2018*). Briefly, BL21(DE3) cells (Invitrogen) transformed with pGEX-KIX plasmid were induced at 37° C with isopropyl beta-D-1-thio-galactopyranoside (IPTG) for 3 hr. Cells were harvested by centrifugation at 3000 x g for 15 min. Pellets were resuspended in lysis buffer of 50 mM Tris-HCl pH 7.3, 150 mM NaCl, 0.1% Tween-20, 1 mM DTT, 5 mM EDTA, supplemented with protease inhibitors at a ratio of 50 mL lysis buffer per 1 L of bacterial culture. Cells were lysed for ten minutes on ice (15 s on, 15 s off, 40% amplitude) using the Misonix probe sonicator (Qsonica, Newtown, CT). Lysates were cleared by centrifugation for 1 hr at 21,800 x g at 4° C. Cleared lysates were incubated with 4 mL glutathione agarose resin slurry (GoldBio) for 1 hr at 4°C to capture GST-KIX. Resin was washed four times with 50 mM Tris-HCl pH 7.4, 150 mM NaCl. KIX domain was cleaved from GST by incubation of resin-bound GST-KIX with 160 U thrombin (GE Healthcare) overnight at room temperature. Resin was centrifuged at 500 x g for 5 min. Supernatant containing cleaved KIX was collected and dialyzed at 4° C against 1 L of 50 mM MOPS pH 6.5, 50 mM NaCl, 10% glycerol, 1 µM tris-2-carboxyethylphosphine (TCEP) overnight at 4°C with mixing. Cleaved KIX was purified using a linear gradient of 50 mM to 1 M NaCl over ten column volumes by cation exchange chromatography using MonoS 5/50 GL column at a flow rate of 1 mL per minute (GE Healthcare). Fractions containing purified KIX were confirmed by SDS-PAGE and dialyzed against 1 L of 50 mM potassium phosphate pH 5.5, 150 mM NaCl, 10 µM TCEP, 30% glycerol overnight at 4°C with mixing. Purified protein was aliquoted and stored at −80°C.

## Microscale thermophoresis

Binding of purified recombinant KIX with fluorescein isothiocyanate (FITC)-conjugated peptides was measured by microscale thermophoresis using the Monolith NT.115 (NanoTemper Technologies). Assays were conducted in 50 mM sodium phosphate, 150 mM NaCl, 0.01% NP-40, pH 5.5. FITC-conjugated peptides (FITC-CREBMIM at 300 nM, FITC-CRYBMIM at 250 nM, and FITC-MYBMIM at 2000 nM, as based on the optical absorbance measurements of FITC at 495 nm) were mixed with increasing concentrations of purified KIX (0.0015 to 50 µM, 1:1 serial dilutions) and loaded into MST Premium Coated capillaries. MST measurements were recorded at room temperature for 10 s per capillary using fixed IR-laser power of 80% and LED excitation power of 40–50%.

## Streptavidin affinity chromatography

Streptavidin magnetic beads (Pierce) were washed with 30 mM Tris-HCl pH 7.4, 150 mM NaCl, 0.1% Tween-20 (TBST) twice prior to use. For each purification, 100 µL of streptavidin beads were used (1 mg beads, binding capacity 3500 pmol biotinylated fluorescein per mg). Biotinylated peptides were bound to streptavidin bead slurry by incubation at room temperature for 1 hr in TBST. Peptide-bound beads were washed twice in TBST and immediately used for purifications. Nuclear lysates were extracted as described below for co-immunoprecipitation. Twenty million cells were used per purification. Nuclear lysates were incubated with Bio-CRYBMIM or Bio-CREBMIM as indicated for 1 hr at room temperature with rotation. Beads were washed twice in lysis buffer and separated for subsequent competition. Per competition, beads were incubated in a total of 1 mL of lysis buffer supplemented with competing peptide for 1 hr at room temperature with rotation. Beads were washed twice in lysis buffer. Bound proteins were eluted by adding 40 µL of western blot sample buffer described below and incubated for 20 min at 95°C, with vortexing half-way through.

## Sequence analysis

Sequence alignment was performed and identity/similarity of each set of sequences were calculated by EMBOSS Needle using the Needleman-Wunsch algorithm from EMBL-EBI (https://www.ebi.ac.uk/Tools/psa/).

## Confocal fluorescence microscopy

Confocal imaging was performed using the Zeiss LSM880 confocal microscope and 63X objective with 1.5 micron z-stack images. Nunc Lab-tek II 8-chamber coverslips were prepared for cell adhesion by the addition of poly-L-Lysine and incubation at room temperature for 1 hr. Poly-L-lysine solution was removed from each chamber and chambers were allowed to air dry. A total of $2 \times 10^5$ cells in 200 µL of fresh media were added to each chamber and incubated for 1 hr at 37°C for attachment. FITC-conjugated peptides were added to cell suspensions at a concentration of 100 nM, mixed, and incubated for 1 hr at 37°C. Cells were counter-stained using Hoechst 33342 and Mitotracker Red CMX ROS (MProbes) for 10 min at a final dilution of 1:10,000 prior to imaging.

## Colony formation assays

Mononuclear cells were isolated from blood using Ficoll-Paque PLUS density centrifugation and enriched for $CD34^+$ cells using the CD34 MicroBead Kit UltraPure, according to the manufacturer's instructions (MACS Miltenyi Biotech, Bergisch Gladbach, German). Methocult H4034 Optimum (StemCell Technologies) semi-solid media, which contains recombinant human SCF, GM-CSF, G-CSF, IL-3, and erythropoietin, was used for the growth of hematopoietic progenitor cells in colony-forming units. $CD34^+$ cells or MV411 cells were resuspended in Iscove's Modified Dulbecco's Medium (IMDM, Corning) media supplemented with 2% FBS at a concentration of $2 \times 10^5$ cells/mL. The cell suspension, the indicated peptide solution at the appropriate concentration, and Methocult H4034 were mixed in a ratio of 0.5/0.5/10 (cell suspension/peptide solution/Methocult) for a final cell concentration of 1000 cells/well of six-well plates. Mixture was vortexed for 30 s and incubated at room temperature for 5 min. Using a blunt-end 16 G needle (StemCell Technologies), 1.1 mL of the solution was added to each well of six-well plates. Peptide treatment conditions were analyzed in biological triplicates. Plates were incubated at 37°C with 5% $CO_2$ for 14 days. The plates were imaged using a Zeiss AxioObserver.Z1 (Zeiss, Oberkochen, Germany) with a 10x/0.45NA objective. Entire well was scanned using transmitted light and the resulting images were stitched using ZEN Blue 2.3 Desk (Zeiss, Oberkochen, Germany) and then exported at half resolution to tif files.

## Cell viability studies

Cells were resuspended in fresh media and plated at a concentration of $2 \times 10^5$ cells in 200 µL per well in a 96-well plate. Peptides or PBS control were added at indicated concentrations. Cells were incubated at 37°C with 5% $CO_2$ for 6 days with media and peptide replacement every 48 hr. On day 6, cell viability was assessed using the CellTiter-Glo Luminescent Viability assay, according to the manufacturer's instructions (Promega). Luminescence was recorded using the Infinite M1000Pro plate reader with an integration time of 250 ms (Tecan). For dose-response assays, cells were resuspended and plated at a concentration of 2000 cells in 200 µL per well of a 96-well plate. MV411 cells were plated in complete RPMI media described above. CD34+ cells isolated from umbilical cord blood were plated IMDM media supplemented with 15% FBS, 100 U/mL penicillin and 100 µg/mL streptomycin, 1% L-glutamine, recombinant human cytokines (PeproTech: 100 ng/mL SCF, 100 ng/mL TPO, 100 ng/mL FLT3, 20 ng/mL IL-6, 20 ng/mL IL-3), and 50 µM β-mercaptoethanol. Increasing concentrations of peptides (0.78–100 µM; 1:1 serial dilution) were added as indicated. Cells were incubated for 48 hr at 37°C with 5% $CO_2$. Cell viability was assessed using Cell Titer Glo as described above.

## CRISPR competitive assays

Cas9-expressing AML cell lines, MOLM13, MV411, OCIAML3, Kasumi1, and K562, were established by lentiviral transduction with a Cas9 expression vector EFS-Cas9-P2A-Puro, as described (*Lu et al., 2018*; *Tarumoto et al., 2018*). Indicated gene-targeting sgRNAs were cloned into LRG2.1 (U6-sgRNA-GFP) or LRC2.1 (U6-sgRNA-mCherry), as described (*Lu et al., 2018*; *Tarumoto et al., 2018*). Specific sgRNA sequences are listed in *Supplementary file 1c*. Lentivirus for each sgRNA was produced by transfecting HEK293T cells with sgRNA-expression vector and helper plasmids, pMD2.G, and psPAX2, using TransIT-LT1 Transfection Reagent (Mirus Bio LLC, Madison, WI, USA) according to the manufacture's instruction. Virus supernatant was collected at 48 and 72 hr post-transfection, pooled, filtered, concentrated by centrifugation using the Amicon Ultra-15 Centrifugal Filter Units (EMD Millipore, Darmstadt, Germany) and stored at −80°C. Cas9-expressing AML cells were

transduced with sgRNA virus particles by spin infection (3500 rpm, 35°C, 90 min) and the subsequent overnight incubation (37°C with 5% $CO_2$) in the presence of 8 µg/ml polybrene. Fresh complete media was supplied to the cells on the next day and every 3 days subsequently. One week after infection, cells were treated with 10 µM or 20 µM of CRYBMIM or PBS for 2 days. After treatment completion, live cells were purified using EasySep Dead Cell Removal (Annexin V) Kit (StemCell Technologies). The remaining viable cells were further incubated in complete media without CRYB-MIM for 6–9 days with media replacement every 3 days. mCherry- and GFP-expressing cells were quantified using flow cytometry (BD Fortessa, Becton Dickinson, San Jose, CA).

## Western blot analysis
Cells were lysed in 30 mM Tris-HCl pH 6.8, 1% SDS, 2% beta-mercaptoethanol, 7% glycerol, 0.0002% Bromophenol Blue buffer or Laemmli sample buffer (BioRad) supplemented with protease inhibitors at a ratio of 100 or 200 µL sample buffer per 1 million cells. Cell suspensions were incubated at 95°C for 15 min, with vortexing every 5 min. Lysates were clarified by centrifugation for 5 min at 18,000 x g. Twenty µL of clarified lysates were resolved by sodium dodecyl sulfate-polyacrylamide gel electrophoresis (SDS-PAGE) using 4–12% or 10% polyacrylamide Bis-Tris gels (Invitrogen) and transferred onto Immobilon FL PVDF or P PVDF membranes (Millipore, Billerica, MA, USA) at 30V for 30–120 min, for fluorescent or chemiluminescent blotting, respectively. For fluorescent western blotting, membranes were blocked using Odyssey Blocking buffer (Li-Cor, Lincoln, Nebraska, USA). Blots were incubated in primary antibodies as indicated. Blotted membranes were visualized using secondary antibodies conjugated to IRDye 800CW or IRDye 680RD (goat anti-rabbit, 1:15,000, and goat anti-mouse, 1:15,000) and the Odyssey CLx fluorescence scanner, according to manufacturer's instructions (Li-Cor, Lincoln, Nebraska, USA). After visualization, bands of interest were selected and signal intensity was quantified using the Image Studio Lite. For some whole cell lysate analysis and all co-IP experiments, chemiluminescent western blotting was used. Blotted membranes were blocked using 5% non-fat milk in TBS, incubated in primary antibodies as indicated, and visualized using HRP-conjugated secondary antibodies (donkey anti-rabbit, 1:15,000, and sheep anti-mouse, 1:15,000, GE Healthcare) and ECL substrate (SuperSignal West Femto Maximum Sensitivity Substrate and Atto Ultimate Sensitivity Substrate, ThermoFisher) with Amersham ImageQuant 800 OD, according to manufacturer's instructions (Cytiva, Marlborough, MA, USA). After imaging, protein signal intensity was quantified using ImageJ. All antibodies are listed in Key Resource Table.

## Nuclear isolation from AML cells
Nuclear purifications were carried out per 100 million cells. Cells were collected by centrifugation at 400 x g for 5 min and washed in cold PBS. Washed cell pellets were resuspended in 15 mL hypotonic buffer (10 mM HEPES pH 7.9, 10 mM NaCl, 1 mM $MgCl_2$, 0.5 mM DTT, protease inhibitors) and incubated on ice for 1 hr. Cells were lysed using 40 strokes by Dounce homogenization. Suspensions were then centrifuged for 15 min at 3300 x g to pellet nuclei. Nuclear pellets were further purified by sucrose density gradient centrifugation. Pellets were resuspended in 2.5 mL of 0.025 M Sucrose, 10 mM $MgCl_2$ and layered on top of 2.5 mL 0.88 M sucrose, 0.05 mM $MgCl_2$ and centrifuged for 10 min at 1200 x g. The final nuclear pellet was resuspended in 4 mL lysis buffer (50 mM Tris-HCl pH 7.4, 150 mM NaCl, 0.25 mM EDTA, 1 mM DTT, 0.5% Triton X-100, 10% glycerol, protease inhibitors) and incubated on ice for 20 min. Nuclear suspensions were homogenized by 15 strokes in a Dounce homogenizer without frothing, and clarified by centrifugation for 15 min at 18,000 x g. Clarified lysates were immediately used for immunoprecipitations.

## Co-immunoprecipitation
For MYB immunoprecipitations, 7.5 µg of anti-MYB antibodies (EP769Y, Abcam) were conjugated to 1 mg of M-270 Epoxy-coated magnetic beads (Invitrogen) according to manufacturer's instructions. For CBP immunoprecipitations, 50 µL of each of Protein A and Protein G Dynabeads (Invitrogen) were combined and washed in 1 mL of PBS with 0.5% BSA. Fifteen µg of anti-CBP antibodies were added to Protein A/G beads in 1 mL PBS with 0.5% BSA and incubated for 1 hr at room temperature with rotation. Beads were then washed with 1 mL PBS with 0.5% BSA and beads were resuspended in a final volume of 100 µL of PBS with 0.5% BSA. One hundred million cells were used per immunoprecipitation. For displacement assays, cells were treated with 10 µM peptides as indicated for 1 hr

at 37°C in complete RPMI media prior to nuclear isolation. Immunoprecipitations were carried out using 100 µL of respective antibody-bead slurry per immunoprecipitation overnight at 4°C with rotation. Beads were washed three times with 1 mL of cold lysis buffer. Proteins were eluted in 40 µL of 0.2 M glycine pH 3 for 30 min on a ThermoMixer (Eppendorf) at 900 rpm at room temperature. Eluates were neutralized with 8 µL of 1 M Tris, pH 11. Samples were prepared for western blot by addition of western blot sample buffer described above together with 2.5 µL of 1M DTT and incubated at 95°C for 5 min.

## Quantitative RT-PCR

Cells were treated as described above for cell viability assays for indicated times points. RNA was isolated using RNeasy kit according to the manufacturer's instructions (Qiagen). Complementary DNA was synthesized using the SuperScript III First-Strand Synthesis system according to the manufacturer's instructions (Invitrogen). Quantitative real-time PCR was performed using the KAPA SYBR FAST PCR polymerase with 200 ng template and 200 nM primers, according to the manufacturer's instructions (Kapa Biosystems, Wilmington, MA, USA). Ct values were calculated by ROX normalization using the ViiA seven software (Applied Biosystems).

## Chromatin immunoprecipitation and sequencing

To prepare antibody-coupled beads, 30 µg of antibodies as indicated per ChIP were incubated with 1 mg of 1:1 Protein A/Protein G slurry in PBS with 0.5% BSA overnight at 4°C with rotation. Beads were washed twice in PBS with 0.5%. Fifty million cells were used per ChIP assay. Cells were fixed in 1% formaldehyde for 10 min at room temperature. Crosslinking was quenched by adding glycine to a final concentration of 125 mM and incubating at room temperature for 5 min. Cells were pelleted by centrifugation at 500 x g at 4°C for 5 min. Cells were washed twice by resuspending 5 mL of PBS with 1 mM PMSF and centrifuging at 500 x g at 4°C for 5 min. Wash step was repeated twice. Crosslinked pellets were resuspended in 2 mL of 50 mM Tris-HCl pH 8.1, 100 mM NaCl, 5 mM EDTA, 0.2% NaN$_3$, 0.5% SDS, and protease inhibitors. Nuclei were pelleted by centrifugation at 15,000 x g for 10 min at 4°C. Pellets were resuspended in 50 mM Tris-HCl, 100 mM NaCl, 5 mM EDTA, 0.2% NaN$_3$, 0.3% SDS, 1.5% Triton X-100, and protease inhibitors and sonicated using the Covaris S220 adaptive focused sonicator at 5% duty Factor, 140W peak incident power, 200 cycles per burst for 30 min at 4°C to obtain 100–500 base pair chromatin fragments (Covaris, Woburn, CA). Nuclear sheared lysates were clarified by centrifugation at 15,000 x g for 10 min at 4°C. Supernatants were incubated with antibody-coupled Protein A/G beads as indicated overnight at 4°C with rotation. Beads were washed three times in 1 mL of 50 mM HEPES-KOH pH 7.6, 500 mM LiCl, 1 mM EDTA, 1% NP-40, 0.7% Na deoxycholate. For the final wash, 1 mL of 50 mM Tris-HCl pH 8, 10 mM EDTA, 50 mM NaCl was added to the beads. Beads were centrifuged for 3 min at 960 x g at 4°C and supernatant was removed. To elute, 210 µL of 50 mM Tris-HCl pH 8, 10 mM EDTA, 1% SDS was added to the beads and incubated for 30 min at 65°C. Beads were centrifuged for 1 min at 16,000 x g at room temperature. Supernatant contains eluted samples. Crosslinks were reversed by incubation overnight at 65°C. Samples were diluted to 400 µL with 50 mM Tris-HCl pH 8, 10 mM EDTA, 50 mM NaCl, and 4 µL of 500 µg/ml RNase (Roche, Cat. No 11119915001) was added and incubated for 1 hr at 37° C to digest RNA. Proteins were then digested by addition of 2 µL 20 mg/mL Proteinase K (Roche) and incubation for 2 hr at 55° C. Samples were purified using PureLink PCR Purification Kit. Libraries were prepared using the KAPA HTP Library preparation kit (Roche), according to the manufacturer's instructions, and sequenced on the Illumina HiSeq 2500 instruments, with 20–30 millions of 50 bp paired-end reads for each sample.

For ChIP-seq analysis, reads were quality and adapter trimmed using 'trim_galore' before aligning to human genome assembly hg19 with bowtie2 using the default parameters. Aligned reads with the same start position and orientation were collapsed to a single read before subsequent analysis. Density profiles were created by extending each read to the average library fragment size and then computing density using the BEDTools suite. Enriched regions were identified using MACS 2.0 and scored against matched input libraries. Genomic 'blacklisted' regions were filtered (http://www.broadinstitute.org/~anshul/projects/encode/rawdata/blacklists/hg19-blacklist-README.pdf) and remaining peaks within 500 bp were merged. Read density normalized by sequencing depth was

then calculated for the union of peaks, and peak dynamics were assessed in DESeq2 using a fold change of 1.5 and an FDR-adjusted p-value of 0.05.

## RNA sequencing

Cells were treated with peptide or control in triplicate as described above for cell viability assays at 20 µM for 1 hr and 4 hr. Cells were collected and RNA was isolated using RNeasy kit (Qiagen) according to manufacturer's instructions. After RiboGreen quantification and quality control by Agilent BioAnalyzer, 500 ng of total RNA underwent polyA selection and TruSeq library preparation according to instructions provided by Illumina (TruSeq Stranded mRNA LT Kit), with 8 cycles of PCR. Samples were barcoded and run on a HiSeq 4000 as PE100, using the HiSeq 3000/4000 SBS Kit (Illumina). An average of 58 million paired reads was generated per sample. At most, ribosomal reads represented 2.9% of the total reads generated and the percent of mRNA bases averaged 74%. Reads were quality and adapter trimmed using 'trim_galore' before aligning to human assembly hg19 with STAR v2.5 using the default parameters. Coverage and post-alignment quality were assessed using the Picard tool CollectRNASeqMetrics (http://broadinstitute.github.io/picard/). Read count tables were created using HTSeq v0.6.1. Normalization and expression dynamics were evaluated with DESeq2 using the default parameters.

## Genome annotations and gene expression estimation

The UCSC knownGene (*Meyer et al., 2013*), Ensembl 71 (*Flicek et al., 2013*), and MISO v.2.0 (*Katz et al., 2010*) annotations were combined into a single genome annotation. Gene expression was estimated as previously described (*Dvinge et al., 2014*). RSEM v.1.2.4 (*Li and Dewey, 2011*) was modified to invoke Bowtie v.1.0.0 (*Langmead et al., 2009*) through the '-v 2' option in order to map all reads to the merged genome annotation and obtain gene expression estimates. With TopHat v.2.1.1 (*Trapnell et al., 2009*), the remaining unaligned reads were mapped to the human genome (GRCh37/hg19) and a database of splice junctions. All expression estimates were normalized via the trimmed mean of M values (TMM) method (*Robinson and Oshlack, 2010*).

## Gene expression analyses

A minimum absolute log2(expression fold-change) requirement was selected to identify differentially-expressed genes between CRYBMIM and PBS treatment, in the maximal number of AML cell lines tested. Feasible thresholds were restricted to expression differences of at least 1.5-fold and associated sample group comparison p-values<0.05. Unsupervised clustering on AML cell lines was based on 41 genes which best separated the CRYBMIM- and PBS-treated samples (*Figure 9—figure supplement 1A*). These genes were differentially expressed in at least four AML cell lines, with a minimum absolute log2(expression fold-change)=0.794. Clustered gene expression heatmaps were created with the pheatmap package (*Kolde, 2020*), using Ward's minimum variance method and a Euclidean distance for hierarchical clustering. For sample group comparisons of differential gene expression, a two-sided t-test was used to determine statistical significance. Statistical analyses were performed in the R programming environment with Bioconductor (*Huber et al., 2015*) and the dplyr (*Wickham et al., 2020*) package. The associated figures were created using the ggplot2 (*Wickham, 2016*) package.

## Flow cytometry

MV411 cells were resuspended to a concentration of $1 \times 10^6$ cells per mL and plated in 12-well tissue culture plates. To assess apoptosis, MYBMIM was used as a positive control and cells were treated as described (*Ramaswamy et al., 2018*). Cells were treated with 20 µM CRYBMIM for 1 hr. Following peptide treatment, cells were washed with PBS and resuspended with PBS with the addition of Annexin V-FITC (1:50) and CD11b (1:50). Cells were incubated on ice for 40 min followed by two washes with cold PBS. DAPI was added prior to flow cytometric analysis (1:10,000).

## Gene dependency analysis across cancer cell types

CRISPR knockout screen gene effect data from Project Achilles 19Q4 was downloaded from the DepMap Consortium (DepMap, Broad (2019): DepMap 19Q4 Public Dataset doi:10.6084/m9.figshare.11384241.v2). Dependency scores for each gene were averaged by tumor type. A differential

dependency score was then calculated as the relative difference between the average gene dependency score for a tumor type and the average gene dependency score across all cell types. All genes were then ranked by leukemia gene differential dependency scores, and the top 10 and bottom 10 leukemia gene dependencies were plotted as a heat map. p-Values were calculated comparing MYB dependency in leukemias versus all other tumor types using two-tailed t-test.

## Mass spectrometry proteomics

Immunoprecipitations were carried out as described above in biological triplicates. Eluates were resolved by SDS-PAGE 10% polyacrylamide Bis-Tris gels (Invitrogen) at 100 V for 5 min. To visualize proteins, gels were stained using Silver Stain for Mass Spectrometry kit (Pierce) according to manufacturer's instructions. Relevant gel portions were excised and destained using 50 µL of 30 mM $K_3[Fe(CN)_6]$ in 100 mM aqueous $Na_2S_2O_3$ by incubation at room temperature for 30 min, with gentle mixing halfway through. Following destaining, 500 µL of 25 mM aqueous $NH_4HCO_3$ were added to each tube and incubated for 5 min at room temperature on a ThermoMixer (Eppendorf) at 700 rpm. Solution was removed and gel pieces were washed by adding 500 µL of 50% acetonitrile in 25 mM aqueous $NH_4HCO_3$ and incubating for 10 min at room temperature on a ThermoMixer at 700 rpm. Wash step was repeated two more times. All solution was removed and 100 µL of acetonitrile was added to each tube and incubated for 5 min at room temperature on a ThermoMixer (Eppendorf) at 700 rpm. All solution was removed and destained gel fragments were vacuum centrifuged and stored at −20°C. For reduction of disulfide bonds, gel fragments were re-hydrated in 25 µL of 10 mM dithiothreitol in 100 mM aqueous $NH_4HCO_3$ for 1 hr at 56°C. Tubes were cooled to room temperature. For alkylation of thiols, 25 µL of 55 mM iodoacetamide in 100 mM aqueous $NH_4HCO_3$ was added to each tube and incubated for 30 min at room temperature in the dark. To quench the alkylation reaction, 5 µL of 100 mM dithiothreitol in 100 mM aqueous $NH_4HCO_3$ was added and incubated for 5 min at room temperature. Gel fragments were washed by adding 50 µL of acetonitrile and incubation for 5 min at room temperature followed by addition of 500 µL of 100 mM aqueous $NH_4HCO_3$ and incubation for 10 min at room temperature. All solution were removed and wash step was repeated. All solution were removed and 100 µL of acetonitrile was added and tubes were incubated for 10 min at room temperature. All solution were removed and gel fragments were vacuum centrifuged and stored at −20°C. For digestion of proteins into peptides, gel fragments were cooled on ice. Protease LysC was added at a ratio of 1:25 protease:protein, assuming 1 µg of protein per immunoprecipitation. Gel fragments were incubated for 5 min on ice. Samples were diluted to 50 µL, or to cover gel fragments using 50 mM aqueous $NH_4HCO_3$. Digestion was carried out for 4 hr at 37°C. Following LysC digestion, trypsin was added to each tube at the 1:25 protease:protein ratio in 50 mM aqueous $NH_4HCO_3$. Trypsin digestion was carried out for 16 hr at 37°C. To elute peptides from gel fragments, 50 µL of 1% formic acid in 70% acetonitrile was added to each tube and incubated for 30 min at room temperature on a ThermoMixer at 1400 rpm. Eluates were removed and saved, and elution step was repeated with fresh 1% formic acid in 70% acetonitrile. New eluates were pooled with the first elution for a total of 100 µL of eluate per sample. Samples were vacuum centrifuged to dryness. Prior to mass spectrometry analysis, samples were desalted and purified by solid phase extraction using C18 Micro SpinColumns (Nest Group) and eluted with 70% acetonitrile supplemented with 0.1% formic acid. Eluates were vacuum centrifuged to remove all solution. Samples were resuspended in 8 µL of 0.1% aqueous formic acid and sonicated for 5 min to ensure full resuspension. Three µL of each sample were used for mass spectrometry analysis.

The LC system used a vented trap-elute configuration (EasynLC1000, Thermo Fisher scientific) coupled to an Orbitrap Lumos mass spectrometer (Thermo Fisher Scientific, San Jose, CA) via a nano electro-spray DPV-565 PicoView ion source (New Objective). The trap column was fabricated with a 5 cm × 150 µm internal diameter silica capillary with a 2 mm silicate frit, and pressure loaded with Poros R2-C18 10 µm particles (Life Technologies). The analytical column consisted of a 25 cm × 75 µm internal diameter column with an integrated electrospray emitter (New Objective), packed with ReproSil-Pur C18-AQ 1.9 µm particles (Dr. Maisch). Peptides were resolved over 90 min using a 3–45% gradient of acetonitrile with 0.1% aqueous formic acid at 250 nL/min.

Precursor ions in the 375–3000 m/z range were isolated using the quadrupole and recorded every 3 s using the Orbitrap detector (60,000 resolution, with 445.1200 ions used as lockmass), with an automatic gain control target set at $10^6$ ions and a maximum injection time of 50 ms. Data-dependent precursor ion selection was enforced, limiting fragmentation to monoisotopic ions with charge

2–5 and precursor ion intensity greater than $5 \times 10^4$, and dynamically excluding already fragmented ions for 30 s (10 ppm tolerance). Selected ions were isolated (Q1 isolation window 0.7 Th) and fragmented using HCD (normalized collision energy 30) using the top speed algorithm. Product ion spectra were recorded in the Orbitrap at 30,000 resolution (AGC of $8 \times 10^4$ ions, maximum injection time 54 ms), in profile mode.

Mass spectra were analyzed using MaxQuant (version 1.6.0.16). For identification, spectra were matched against the human UniProt database (as of October 2017), supplemented with contaminant proteins from the cRAP database with FDR < 0.01. After m/z recalibration, mass tolerance was set at 4.5 and 20 ppm for precursor and fragment ions, respectively. Cysteine carbamidomethylation was set as fixed chemical modification, while methionine oxidation and protein N-terminus acetylation were set as variable. Protease specificity was set to trypsin/P, with up to two missed cleavages allowed. The match between runs feature was enabled (0.7 min tolerance, 20 min alignment). Quantification was performed using the LFQ algorithm.

Contaminating peptides such as keratin and non-human proteins were excluded and label-free quantification (LFQ) intensity values for protein groups were used for analysis. IgG and MYB immunoprecipitation samples were analyzed together (three replicates per condition; six samples total). Proteins in the following categories were excluded from analysis: (i) zero LFQ intensity across all replicates, (ii) zero LFQ intensity in five of six replicates, (iii) proteins with LFQ intensity recorded in only one replicate in both sets of samples. Enrichment scores were defined as the log2 ratio of MYB:IgG.

### Statistical analysis, principal component analysis, and hierarchical clustering

OriginPro 2018 (Origin Lab) was used for statistical analysis. Principal component analysis was performed using prcomp (R Studio). The hierarchical cluster analysis was carried out with R function pheatmap with Pearson's distance as the distance metrics and 'average' as the clustering method.

## Acknowledgements

We thank all our laboratory members, especially Benjamin Herzberg, Shray Khanna and Isaac Krasnopolsky, as well as Henrik Molina, Caitlin Steckler, and the MSK Integrated Genomics Operation core for technical assistance. We thank Alejandro Gutierrez, Marc Mansour, Kristian Helin and Yaniv Kazansky for critical comments on the manuscript. This research was supported by the NIH R01 CA204396, R01 DK103854, R01 HL128239, R01 HL151651, R01 CA251138, T32 GM073546, P30 CA008748, St. Baldrick's Foundation, Hyundai Hope on Wheels, Burroughs Wellcome Fund, Damon Runyon-Richard Lumsden Foundation, Rita Allen Foundation, Leukemia and Lymphoma Society, Blood Cancer Discoveries Grant program (Leukemia and Lymphoma Society 8023–20), the Starr Cancer Consortium, Mr. William H and Mrs. Alice Goodwin and the Commonwealth Foundation for Cancer Research the Center for Experimental Therapeutics at MSKCC, Edward P Evans Foundation, Mark Foundation for Cancer Research, and the Paul G Allen Frontiers Group.

## Additional information

#### Competing interests

Alex Kentsis: AK is a consultant for Novartis and Rgenta. A patent application related to this work has been submitted to the U.S. Patent and Trademark Office entitled 'Agents and methods for treating CREB binding protein-dependent cancers' (application PCT/US2017/059579). The other authors declare that no competing interests exist.

#### Funding

| Funder | Grant reference number | Author |
| --- | --- | --- |
| National Institutes of Health | R01 CA204396 | Alex Kentsis |
| National Institutes of Health | P30 CA008748 | Alex Kentsis |
| National Institutes of Health | T32 GM073546 | Lauren Forbes |

| | | |
|---|---|---|
| National Institutes of Health | R01 DK103854 | Robert K Bradley |
| National Institutes of Health | R01 HL128239 | Robert K Bradley |
| National Institutes of Health | R01 HL151651 | Robert K Bradley |
| National Institutes of Health | R01 CA251138 | Robert K Bradley |
| St. Baldrick's Foundation | | Alex Kentsis |
| Hyundai Hope On Wheels | | Alex Kentsis |
| Burroughs Wellcome Fund | | Alex Kentsis |
| Damon Runyon Cancer Research Foundation | | Alex Kentsis |
| Rita Allen Foundation | | Alex Kentsis |
| Leukemia and Lymphoma Research | | Alex Kentsis |
| Starr Center | | Alex Kentsis<br>Christopher R Vakoc |
| Mr. William H. and Mrs. Alice Goodwin | | Alex Kentsis |
| Memorial Sloan-Kettering Cancer Center | | Michael G Kharas<br>Richard P Koche<br>Alex Kentsis |
| Edward P. Evans Foundation | | Robert K Bradley |
| Mark Foundation for Cancer Research | | Robert K Bradley |
| Paul G. Allen Family Foundation | | Robert K Bradley |

The funders had no role in study design, data collection and interpretation, or the decision to submit the work for publication.

## Author contributions
Sumiko Takao, Conceptualization, Formal analysis, Validation, Investigation, Methodology, Writing - original draft, Writing - review and editing; Lauren Forbes, Conceptualization, Formal analysis, Investigation, Methodology, Writing - original draft, Writing - review and editing; Masahiro Uni, Conceptualization, Formal analysis, Investigation, Methodology, Writing - review and editing; Shuyuan Cheng, Formal analysis, Investigation, Methodology, Writing - review and editing; Jose Mario Bello Pineda, Paolo Cifani, Formal analysis; Yusuke Tarumoto, Gerard Minuesa, Celine Chen, Formal analysis, Investigation; Michael G Kharas, Funding acquisition; Robert K Bradley, Christopher R Vakoc, Formal analysis, Funding acquisition; Richard P Koche, Formal analysis, Validation, Methodology, Writing - review and editing; Alex Kentsis, Conceptualization, Resources, Formal analysis, Supervision, Funding acquisition, Validation, Investigation, Methodology, Writing - original draft, Project administration, Writing - review and editing

## Author ORCIDs
Sumiko Takao ![ORCID] https://orcid.org/0000-0001-5573-1677
Shuyuan Cheng ![ORCID] https://orcid.org/0000-0003-0746-509X
Jose Mario Bello Pineda ![ORCID] http://orcid.org/0000-0003-1417-9200
Yusuke Tarumoto ![ORCID] https://orcid.org/0000-0001-6652-9618
Robert K Bradley ![ORCID] http://orcid.org/0000-0002-8046-1063
Richard P Koche ![ORCID] https://orcid.org/0000-0002-6820-5083
Alex Kentsis ![ORCID] https://orcid.org/0000-0002-8063-9191

## Decision letter and Author response
Decision letter https://doi.org/10.7554/eLife.65905.sa1
Author response https://doi.org/10.7554/eLife.65905.sa2

# Additional files

## Supplementary files

• Supplementary file 1. Key material detailed information. 1a. Peptide list. List of retro-inverso peptides with extinction coefficients. 1b. AML Cell line sequencing. Mutational landscape of human AML cell lines determined by whole genome sequencing 1 c. sgRNA sequence list.

• Supplementary file 2. Genom-wide CRISPR screen and Gene expression analysis. 2a. MYBMIM GeCKO CRISPR screen gene summary. Top 50 genes are shown. The complete tables are listed in *Figure 1—source data 1* and Zenodo (DOI: 10.5281/zenodo.4321824) 2b-e. Differentially expressed genes measured by RNA-seq in CRYBMIM, CREBMIM vs PBS-treated MV411 cells (1 hr and 4 hr). Top 50 differentially expressed genes in response to CRYBMIM are shown. The complete tables are listed in *Figure 5—source data 1* and Zenodo (DOI: 10.5281/zenodo.4321824). 2 f. Gene set enrichment analysis (GSEA) for differentially expressed genes measured by RNA-seq in CRYBMIM, CREBMIM vs PBS-treated MV411 cells. Summary of GSEA on CRYBMIM-induced differentially expressed genes are shown. The complete tables are listed in Zenodo (DOI: 10.5281/zenodo.4321824).

• Supplementary file 3. IP-MS and ChIP-seq analysis. 3a. Protein lists identified by IP-MS of MYB and CBP complex purification from MV411 cell nuclei. Summary of identified proteins is shown. The complete tables are listed in *Figure 7—source data 1* and Zenodo (DOI: 10.5281/zenodo.4321824). 3b. MYB complex functional groups. 3 c. Pathway analysis for 9 clusters of transcription factor-remodeled genes in response to CRYBMIM measured by ChIP-seq. Top 20 pathways in each cluster are shown. The complete tables are listed in Zenodo (DOI: 10.5281/zenodo.4321824).

• Transparent reporting form

## Data availability

All supplemental data are available openly via Zenodo (https://doi.org/10.5281/zenodo.4321824). Mass spectrometry proteomics data are available via PRIDE (PXD019708). Gene expression and chromatin dynamics data are available via Gene Expression Omnibus (GSE163470).

The following datasets were generated:

| Author(s) | Year | Dataset title | Dataset URL | Database and Identifier |
|---|---|---|---|---|
| Forbes L, Koche RP, Cifani P, Kentsis A, Takao S, Uni M | 2020 | Supplementary Data for "Convergent organization of aberrant MYB complexes controls oncogenic gene expression in acute myeloid leukemia" | https://doi.org/10.5281/zenodo.4321824 | Zenodo, 10.5281/zenodo.4321824 |
| Kentsis A, Forbes L | 2020 | Definition of aberrant MYB CBP/P300 complex in AML | https://www.ebi.ac.uk/pride/archive/projects/PXD019708 | PRIDE, PXD019708 |
| Koche RP, Forbes L, Takao S, Pineda JMB, Bradley RK, Kentsis A | 2021 | Convergent organization of aberrant MYB complexes controls oncogenic gene expression in acute myeloid leukemia | https://www.ncbi.nlm.nih.gov/geo/query/acc.cgi?acc=GSE163470 | NCBI Gene Expression Omnibus, GSE163470 |

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

# Appendix 1

**Appendix 1—key resources table**

| Reagent type (species) or resource | Designation | Source or reference | Identifiers | Additional information |
| --- | --- | --- | --- | --- |
| Cell line (*Homo-sapiens*) | MV411 | ATCC | Cat# CRL-9591, RRID:CVCL_0064 | |
| Cell line (*Homo-sapiens*) | MOLM13 | DSMZ | Cat# ACC-554, RRID:CVCL_2119 | |
| Cell line (*Homo-sapiens*) | OCIAML2 | DSMZ | Cat# ACC-99, RRID:CVCL_1619 | |
| Cell line (*Homo-sapiens*) | THP1 | ATCC | Cat# TIB-202, RRID:CVCL_0006 | |
| Cell line (*Homo-sapiens*) | NB4 | DSMZ | Cat# ACC-207, RRID:CVCL_0005 | |
| Cell line (*Homo-sapiens*) | Kasumi1 | ATCC | ATCC Cat# CRL-2724, RRID:CVCL_0589 | |
| Cell line (*Homo-sapiens*) | HEL | ATCC | Cat# TIB-180, RRID:CVCL_2481 | |
| Cell line (*Homo-sapiens*) | OCIAML3 | DSMZ | Cat# ACC-582, RRID:CVCL_1844 | |
| Cell line (*Homo-sapiens*) | SKM1 | DSMZ | Cat# ACC-547, RRID:CVCL_0098 | |
| Cell line (*Homo-sapiens*) | U937 | ATCC | Cat# CRL-1593.2, RRID:CVCL_0007 | |
| Cell line (*Homo-sapiens*) | HL60 | ATCC | Cat# CCL-240, RRID:CVCL_0002 | |
| Cell line (*Homo-sapiens*) | K562 | ATCC | Cat# CCL-243, RRID:CVCL_0004 | |
| Antibody | anti-v-Myb + c-Myb (phospho S11) (Rabbit monoclonal, EP769Y) | Abcam | Cat# ab45150, RRID: AB_778878 | IP (7.5 µg for 100 million cells) |
| Antibody | anti-c-Myb (Rabbit monoclonal, D1B9E) | Cell Signaling Technology | Cat# 59995, RRID: AB_2799836 | WB (1:1000) |
| Antibody | anti-CBP (Rabbit monoclonal, D9B6) | Cell Signaling Technology | Cat# 7425, RRID: AB_10949975 | WB (1:500–1000), IP (15 µg for 100 million cells) |
| Antibody | anti-CBP (Rabbit polyclonal, PA1-847) | Invitrogen | Cat# PA1-847, RRID: AB_2083939 | WB (1:500–1000) |

*Continued on next page*

*Appendix 1—key resources table continued*

| Reagent type (species) or resource | Designation | Source or reference | Identifiers | Additional information |
|---|---|---|---|---|
| Antibody | anti-p300 (Rabbit monoclonal, D8Z4E) | Cell Signaling Technology | Cat# 86377, RRID: AB_2800077 | WB (1:500) |
| Antibody | anti-RUNX1 (Rabbit monoclonal, D33G6) | Cell Signaling Technology | Cat# 4336, RRID: AB_10859035 | WB (1:1000) |
| Antibody | anti-RUNX1 (Mouse monoclonal, 1C5B16) | BioLegend | Cat# 659302, RRID: AB_2563194 | WB (1:1000) |
| Antibody | anti-LYL1 (Mouse monoclonal, C-4) | Santa Cruz Biotechnology | Cat# sc-374164, RRID: AB_10986408 | WB (1:500) |
| Antibody | anti-SATB1 (Rabbit monoclonal, P472) | Cell Signaling Technology | Cat# 3643, RRID: AB_2184328 | WB (1:1000) |
| Antibody | anti-E2A (Rabbit monoclonal, D2B1) | Cell Signaling Technology | Cat# 12258, RRID: AB_2797860 | WB (1:1000) |
| Antibody | anti-CEBPA (Rabbit monoclonal, D56F10) | Cell Signaling Technology | Cat# 8178, RRID: AB_11178517 | WB (1:1000) |
| Antibody | anti-CEBPA (Mouse monoclonal, G-10) | Santa Cruz Biotechnology | Cat# sc-166258, RRID: AB_2078042 | WB (1:500) |
| Antibody | anti-SPI1 (Rabbit monoclonal) | Cell Signaling Technology | Cat# 2266, RRID: AB_10692379 | WB (1:1000) |
| Antibody | anti-LDB1 (Rabbit polyclonal) | Cell Signaling Technology | Cat# 55476, RRID: AB_2799486 | WB (1:1000) |
| Antibody | anti-LMO2 (Rabbit monoclonal, E8K6I) | Cell Signaling Technology | Cat# 87182, RRID: NA | WB (1:1000) |
| Antibody | anti-c-Jun (Rabbit monoclonal, 60A8) | Cell Signaling Technology | Cat# 9165, RRID: AB_2130165 | WB (1:1000) |
| Antibody | anti-β-Actin (Mouse monoclonal, 8H10D10) | Cell Signaling Technology | Cat# 3700, RRID: AB_2242334 | WB (1:5000) |
| Antibody | anti-β-Actin (Rabbit monoclonal, 13E5) | Cell Signaling Technology | Cat# 4970, RRID: AB_2223172 | WB (1:5000) |
| Antibody | anti-Lamin B1 (Mouse monoclonal, A-11) | Santa Cruz Biotechnology | Cat# sc-377000, RRID: AB_2861346 | WB (1:500) |
| Antibody | anti-Lamin B1 (Rabbit monoclonal, HRP Conjugate, D9V6H) | Cell Signaling Technology | Cat# 15068, RRID: AB_2798695 | WB (1:1000) |
| Antibody | anti-MED15 (Rabbit polyclonal) | Bethyl | Cat# A302-422A, RRID: AB_1907305 | WB (1:2000) |

*Continued on next page*

*Appendix 1—key resources table continued*

| Reagent type (species) or resource | Designation | Source or reference | Identifiers | Additional information |
|---|---|---|---|---|
| Recombinant DNA reagent | Human CRISPR Knockout Pooled Library GeCKO v2 | *Sanjana et al., 2014* | Addgene Pooled Library #1000000049 | |
| Recombinant DNA reagent | LRG2.1 (U6-sgRNA-GFP) (Plasmid) | *Tarumoto et al., 2018*; *Lu et al., 2018* | Addgene Plasmid #108098, RRID:Addgene_108098 | |
| Recombinant DNA reagent | LRC2.1 (U6-sgRNA-mCherry) (Plasmid) | *Tarumoto et al., 2018*; *Lu et al., 2018* | Addgene Plasmid #108099, RRID:Addgene_108099 | |
| Recombinant DNA reagent | LentiV_Cas9_Blast (Plasmid) | *Tarumoto et al., 2020* | Addgene Plasmid #125592, RRID:Addgene_125592 | |
| Recombinant DNA reagent | LentiV_Cas9_puro (EFS-Cas9-P2A-Puro) (Plasmid) | *Tarumoto et al., 2018*; *Lu et al., 2018* | Addgene Plasmid #108100, RRID:Addgene_108100 | |
| Recombinant DNA reagent | MSCV-BCL2-IRES-GFP (Plasmid) | *Sanda et al., 2013* | | |
| Sequence-based reagent | sgCBP-1; sgCBP-2; sgEP300-1; sgEP300-2; sgCDK1; sgNEG1; sNEG2 | *Tarumoto et al., 2018*; *Lu et al., 2018* | | sgRNAs See *Supplementary file 1c* |
| Peptide, recombinant protein | MYBMIM | *Ramaswamy et al., 2018* | | See *Supplementary file 1a* |
| Peptide, recombinant protein | CRYBMIM; CREBMIM; CG3; TG3; MLLMIM | This study | | See *Supplementary file 1a* |
| Peptide, recombinant protein | Bio-CRYBMIM; Bio-CREBMIM; FITC-CRYBMIM; FITC-CREBMIM | This study | | See *Supplementary file 1a* |
| Peptide, recombinant protein | RI-TAT | This study | | See *Supplementary file 1a* |
| Commercial assay or kit | CellTiter-Glo Luminescent Cell Viability Assay | Promega | G7571 | |
| Commercial assay or kit | CD34 MicroBead Kit UltraPure, human | Miltenyi Biotech | 130-100-453 | |
| Commercial assay or kit | Dynabeads Antibody Coupling Kit | Thermo Fisher | 14311D | |
| Chemical compound, drug | Doxorubicin | Sigma Aldrich | D1515 | |
| Software, algorithm | DepMap 19Q4 | Cancer Dependency Map Portal | RRID:SCR_017655 | |
| Software, algorithm | EMBOSS Needle | European Bioinformatics Institute | RRID:SCR_004727 | Alignment of protein sequences |

*Continued on next page*

*Appendix 1—key resources table continued*

| Reagent type (species) or resource | Designation | Source or reference | Identifiers | Additional information |
|---|---|---|---|---|
| Software, algorithm | MacroModel | Schrodinger | RRID:SCR_014879 | Force Field-based Molecular Modeling |
| Software, algorithm | MAGeCK | *Li et al., 2014* | RRID:N.A. | Genome-wide CRISPR screen |
| Software, algorithm | MaxQuant (version 1.6.0.16) | Max planck institute of biochemistry | RRID:SCR_014485 | mass-spectrometric data analysis |
| Software, algorithm | trim_galore | Babraham Institute | RRID:SCR_011847 | ChIP-seq data analysis |
| Software, algorithm | bowtie2 | Johns Hopkins University | RRID:SCR_016368 | ChIP-seq data analysis |
| Software, algorithm | BEDTools suite | *Quinlan and Hall, 2010* | RRID:SCR_006646 | ChIP-seq data analysis |
| Software, algorithm | MACS 2.0 | *Zhang et al., 2008*, Genomebiol | RRID:SCR_013291 | ChIP-seq data analysis |
| Software, algorithm | CollectRNASeqMetrics | Picard tool | RRID:SCR_006525 | RNA-seq data analysis |
| Software, algorithm | HTSeq v0.6.1. | *Anders et al., 2014* Zanini et al., 2021 (in preparation) | RRID:SCR_005514 | RNA-seq data analysis |
| Software, algorithm | DESeq2 | Bioconductor | RRID:SCR_015687 | RNA-seq and ChIP-seq data analysis |
| Software, algorithm | UCSC knownGene | *Meyer et al., 2013* | RRID:SCR_005780 | RNA-seq data analysis |
| Software, algorithm | Ensembl 71 | *Flicek et al., 2013* | RRID:SCR_002344 | RNA-seq data analysis |
| Software, algorithm | MISO v.2.0 | *Katz et al., 2010* | RRID:SCR_003124 | RNA-seq data analysis |
| Software, algorithm | RSEM v.1.2.4 | *Li and Dewey, 2011* | RRID:SCR_013027 | RNA-seq data analysis |
| Software, algorithm | Bowtie v.1.0.0 | *Langmead et al., 2009* | RRID:SCR_005476 | RNA-seq data analysis |
| Software, algorithm | TopHat v.2.1.1 | *Trapnell et al., 2009* | RRID:SCR_013035 | RNA-seq data analysis |
| Software, algorithm | TMM method | *Robinson and Oshlack, 2010* | RRID:SCR_012802 (edgeR) | RNA-seq data analysis |
| Software, algorithm | pheatmap package | *Kolde, 2020* | RRID:SCR_016418 | RNA-seq data analysis |
| Software, algorithm | R programming environment with Bioconductor | *Huber et al., 2015* | RRID:SCR_001905 | RNA-seq data analysis |
| Software, algorithm | dplyr | *Wickham et al., 2020* | RRID:SCR_016708 | RNA-seq data analysis |
| Software, algorithm | ggplot2 | *Wickham, 2016* | RRID:SCR_014601 | RNA-seq data analysis |
| Software, algorithm | RStudio | RStudio | RRID:SCR_000432 | Statistical analysis |
| Software , algorithm | ImageJ | Fiji | RRID:SCR_002285 | Image processing |

*Continued on next page*

*Appendix 1—key resources table continued*

| Reagent type (species) or resource | Designation | Source or reference | Identifiers | Additional information |
|---|---|---|---|---|
| Software, algorithm | FCS express 7 | De Novo | RRID:SCR_016431 | Flow cytometry data analysis |
| Software, algorithm | OriginPro 2018 | OriginLab | RRID:SCR_014212 | Statistical analysis |

## Supplemental information

All supplemental data are available openly via Zenodo (https://doi.org/10.5281/zenodo.4321824). Mass spectrometry proteomics data are available via PRIDE (PXD019708). Gene expression and chromatin dynamics data are available via Gene Expression Omnibus (GSE163470).

## Lists of supplemental data deposited in zenodo

**MYBMIM Genome-wide CRISPR screen (*Figure 1*)**

| | |
|---|---|
| 01_MYBMIM_GeCKO_gene_summary.xlsx | Genome-wide GeCKO CRISPR screen in MYBMIM treated MOLM13 cells |
| 02_Control_GeCKO_gene_summary.xlsx | Genome-wide GeCKO CRISPR screen in PBS treated MOLM13 cells |

**RNA-seq and GSEA with CRYBMIM and CREBMIB (*Figure 5*)**

| | |
|---|---|
| 03_RNAseq_DESeq2_CRYBMIM_1hr-vs-PBS_1 hr_SignificantAll.xlsx | Excel file of differentially expressed genes measured by RNA-seq in 1 hr CRYBMIM vs PBS treated MV411 |
| 04_RNAseq_DESeq2_CRYBMIM_4hr-vs-PBS_4 hr_SignificantAll.xlsx | Excel file of differentially expressed genes measured by RNA-seq in 4 hr CRYBMIM vs PBS treated MV411 |
| 05_RNAseq_DESeq2_CREBMIM_1hr-vs-PBS_1 hr_SignificantAll.xlsx | Excel file of differentially expressed genes measured by RNA-seq in 1 hr CREBMIM vs PBS treated MV411 |
| 06_RNAseq_DESeq2_CREBMIM_4hr-vs-PBS_4 r_SignificantAll.xlsx | Excel file of differentially expressed genes measured by RNA-seq in 4 hr CREBMIM vs PBS treated MV411 |
| 07_gsea_report_for_na_pos_1552361301359_CRYBMIM-vs-PBS_1 hr.xlsx | Gene set enrichment analysis for differentially expressed genes measured by RNA-seq in 1 hr CRYBMIM vs PBS (positive enrichment) |
| 08_gsea_report_for_na_neg_1552361301359_CRYBMIM-vs-PBS_1 hr.xlsx | Gene set enrichment analysis for differentially expressed genes measured by RNA-seq in 1 hr CRYBMIM vs PBS (negative enrichment) |
| 09_gsea_report_for_na_pos_1552361302266_CRYBMIM-vs-PBS_4 hr.xlsx | Gene set enrichment analysis for differentially expressed genes measured by RNA-seq in 4 hr CRYBMIM vs PBS (positive enrichment) |
| 10_gsea_report_for_na_neg_1552361302266_CRYBMIM-vs-PBS_4 hr.xlsx | Gene set enrichment analysis for differentially expressed genes measured by RNA-seq in 4 hr CRYBMIM vs PBS (negative enrichment) |
| 11_gsea_report_for_na_pos_1552361302042_CREBMIM-vs-PBS_1 hr.xlsx | Gene set enrichment analysis for differentially expressed genes measured by RNA-seq in 1 hr CREBMIM vs PBS (positive enrichment) |
| 12_gsea_report_for_na_neg_1552361302042_CREBMIM-vs-PBS_1 hr.xlsx | Gene set enrichment analysis for differentially expressed genes measured by RNA-seq in 1 hr CREBMIM vs PBS (negative enrichment) |
| 13_gsea_report_for_na_pos_1552361302044_CREBMIM-vs-PBS_4 hr.xlsx | Gene set enrichment analysis for differentially expressed genes measured by RNA-seq in 4 hr CREBMIM vs PBS (positive enrichment) |
| 14_gsea_report_for_na_neg_1552361302044_CREBMIM-vs-PBS_4 hr.xlsx | Gene set enrichment analysis for differentially expressed genes measured by RNA-seq in 4 hr CREBMIM vs PBS (negative enrichment) |

*continued*

**MYBMIM Genome-wide CRISPR screen (*Figure 1*)**

**ChIP-seq analysis for MYB and CBP
(*Figures 6* and *11*)**

| | |
|---|---|
| 15_ChIPseq_DESeq2_Named_Significant Increase_All_CRYBMIM-vs-PBS_1 hr_MYB.xlsx | Differentially increased peaks measured by MYB ChIP-seq in 1 hr CRYBMIM vs PBS treated MV411 |
| 16_ChIPseq_DESeq2_Named_Significant Decrease_All_CRYBMIM-vs-PBS_1 hr_MYB.xlsx | Differentially decreased peaks measured by MYB ChIP-seq in 1 hr CRYBMIM vs PBS treated MV411 |
| 17_ChIPseq_DESeq2_Named_Significant Increase_All_CRYBMIM-vs-PBS_1 hr_CBP.xlsx | Differentially increased peaks measured by CBP ChIP-seq in 1 hr CRYBMIM vs PBS treated MV411 |
| 18_ChIPseq_DESeq2_Named_Significant Decrease_All_CRYBMIM-vs-PBS_1 hr_CBP.xlsx | Differentially decreased peaks measured by CBP ChIP-seq in 1 hr CRYBMIM vs PBS treated MV411 |
| 19_ChIPseq_DESeq2_Named_Significant Increase_All_CRYBMIM-vs-PBS_4 hr_MYB.xlsx | Differentially increased peaks measured by MYB ChIP-seq in 4 hr CRYBMIM vs PBS treated MV411 |
| 20_ChIPseq_DESeq2_Named_Significant Decrease_All_CRYBMIM-vs-PBS_4 hr_MYB.xlsx | Differentially decreased peaks measured by MYB ChIP-seq in 4 hr CRYBMIM vs PBS treated MV411 |
| 21_ChIPseq_DESeq2_Named_Significant Increase_All_CRYBMIM-vs-PBS_4 hr_CBP.xlsx | Differentially increased peaks measured by CBP ChIP-seq in 4 hr CRYBMIM vs PBS treated MV411 |
| 22_ChIPseq_DESeq2_Named_Significant Decrease_All_CRYBMIM-vs-PBS_4 hr_CBP.xlsx | Differentially decreased peaks measured by CBP ChIP-seq in 4 hr CRYBMIM vs PBS treated MV411 |

**MYB and CBP IP-MS data (*Figure 7*)**

| | |
|---|---|
| 23_IP-MS_MYB_IgG_protein list.xlsx | List of proteins identified by IP-MS of IgG control and MYB complex purifications from MV411 cell nuclei |
| 24_IP-MS_CBP_IgG_protein_list.xlsx | List of proteins identified by IP-MS of IgG control and CBP complex purifications from MV411 cell nuclei |

**RNA-seq analysis with CRYBMIM in 5 AML
cell lines (*Figure 9—figure supplement 1*)**

| | |
|---|---|
| 25_RNAseq_expression_all_coding_5AML_celllines.xlsx | All coding gene expression changes measured by RNA-seq in 1 hr CRYBMIM vs PBS treated MV411, HL-60, Kasumi-1, OCI-AML3 and U937 |

**ChIP-seq anaysis for multiple TFs (*Figure 11*)**

| | |
|---|---|
| 26_ChIPseq_DESeq2_FilteredNormalized Counts_PeakNorm_CRYBMIM-vs-PBS_1 hr_allTFs.xlsx | Differentially increased peaks measured by multiple TF ChIP-seq in 1 hr CRYBMIM vs PBS treated MV411 |
| 27_ChIPseq_pathway_analysis_9 clusters.xlsx | Pathway analysis for 9 clusters of transcription factor-remodeled genes measured by ChIP-seq |

