## [Decision Letter]

[Editors' note: this paper was reviewed by Review Commons.]

**Acceptance summary:**

This study describes the characterization of a novel peptidomimetic that disrupts the interaction between MYB and its CBP cofactor and formation of aberrant transcription factor complexes in leukemia cells. As a result, CBP and transcription factors are re-localized over the genome leading to a switch from the oncogenic transcriptional program to a differentiation program and diminished viability of the cancer cells. The data provide evidence for how specific peptide inhibitors can be used to target cancer cells.

**Decision letter after peer review:**

Thank you for submitting your article "Convergent organization of aberrant MYB complex controls oncogenic gene expression in acute myeloid leukemia" for consideration by *eLife*. Your article has been reviewed by a Reviewing Editor and the evaluation has been overseen by Kevin Struhl as the Senior Editor.

The Reviewing Editor has assessed the previous reviews from Review Commons and the revised manuscript and has drafted this decision to help you prepare a revised submission.

We would like to draw your attention to changes in our policy on revisions we have made in response to COVID-19 (https://elifesciences.org/articles/57162). Specifically, when editors judge that a submitted work as a whole belongs in *eLife* but that some conclusions require a modest amount of additional new data, as they do with your paper, we are asking that the manuscript be revised to either limit claims to those supported by data in hand, or to explicitly state that the relevant conclusions require additional supporting data.

Summary:

This study describes the characterization of a novel peptidomimetic that disrupt the interaction between MYB and its CBP cofactor and formation of aberrant transcription factor complexes in leukemia cells. As a result, CBP and transcription factors are re-localized over the genome leading to a switch from the oncogenic transcriptional program to a differentiation program and diminished viability of the cancer cells. The data provide evidence for how specific peptide inhibitors can be used to target cancer cells. The manuscript is pertinent and should be of general interest to readers of e*Life*.

Revisions:

While the authors have satisfactorily responded to the majority of the comments raised by the reviewers of Review Commons, some additional issues have to be addressed before publication can be recommended.

1) Throughout the manuscript, the authors include CEBPA as a MYB cofactor, but it does not co-precipitate with MYB in any of the immunoblots that are shown with the exception of a very weak band in OCAIML2 cells in Figure 8A. The evidence for a MYB-CEBPA interaction is thus very tenuous. The authors should address this issue either by tempering their claims or providing more convincing evidence.

2) It is not clear why CRISPR inactivation of CBP shown in Figure 4 can have such minor effects, considering that the main message of the paper is that MYB-CBP interactions are critical for activating the genes required for growth of the leukemic cells. The authors have to provide a clear explanation of this apparent discrepancy. Related to this is an important comment of reviewer 3 from Review Commons that the peptide inhibitor induced MYB degradation that is clearly seen after 4 hours of treatment in Figure 11C. Given this observation, how can the authors explain the gained MYB peaks in the ChIP-seq. The authors have not correctly addressed this issue. It is of course reasonable to imagine that MYB stability depends upon its interactions with CBP and that when this is blocked by the peptide MYB is degraded. But if this is the case why does CRIPSR inactivation of CPB not induce the same effects. In fact, the stability of MYB in the CBP KO cells should be investigated.

---

## [Author Response]

Dear Dr. Davidson and Dr. Struhl,

Thank you for your help with this manuscript. I want to emphasize just how refreshing and constructive the Review Commons with eLife process has been. I also want to point out that your suggestions are very helpful. Based on this, we have revised the manuscript, incorporating new data for MYB stability and modifying the conclusions to more accurately reflect the findings.

We thank the reviewers for their constructive suggestions, which have substantially improved this work. We have comprehensively revised the manuscript, and detail individual responses below:

Reviewer #1 (Evidence, reproducibility and clarity (Required)):The study by Forbes et al. describes and characterizes a 2nd generation peptide-based inhibitor of the MYB:CBP interaction, termed CRYBMIM, which they use to study MYB:cofactor interactions in leukemia cells. The CRYBMIM has improved properties relative to the MYBMIM peptide, and display more potency in biochemical and cell-based assays. Using a combination of epigenomics and biochemical screens, the authors define a list of candidate MYB cofactors whose functional significance as AML dependencies is supported by analysis of the DepMap database. Using genomewide profiling of TF and CBP occupancy, the authors provide evidence that CRYBMIM treatment reprograms the interactome of MYB in a manner that disproportionately changes specific cis-elements over others. Stated differently, the overall occupancy pattern of many TFs/cofactors shows gains and losses at specific cis elements, resulting in a complex modulation of MYB function and changes in transcription in leukemia cells.Overall, this is a strong, well-written study, with clear experimental results and relatively straightforward conclusions. The therapeutic potential of modulating MYB in cancer is enormous, and hence I believe this study will attract a broad interest in the cancer field and will likely be highly cited. I list below a few control experiments that would clarify the specificity of CRYBMIM.1) Does CRYBMIM bind to other KIX domains, such as of MED15. It would be important to evaluate the specificity of this peptide for whether it binds to other KIX domains.

We analyzed all known human KIX domain sequences, and found that the most similar one to CBP/P300 is MED15 (38% identity), as shown in revised Figure 2—figure supplement 1D. The sequence similarity of the remaining human KIX domains is substantially lower. To determine the specificity of CRYBMIM in binding the CBP/P300 versus MED15, we exposed human AML cell extracts to biotinylated CRYBMIM immobilized on streptavidin beads versus beads alone. Whereas CRYBMIM binds efficiently to CBP/P300, it does not exhibit any measurable binding to MED15 (even though MED15 is highly expressed), as shown in revised Figure 2—figure supplement 1E. While this does not exclude the possibility that CRYBMIM binds to other proteins, the biochemical specificity observed here, combined with the genetic requirement of CBP for cellular effects of CRYBMIM as shown by a genome-wide CRISPR screen (Figure 1B), indicate that CRYBMIM is a specific ligand of CBP/P300. The manuscript has been revised accordingly.

2) Similarly, it would be useful to perform a mass spec analysis to all nuclear factors that associate with streptavidin-immobilized CRYBMIM. This again would be help the reader to understand the specificity of this peptide.

We agree with the reviewer that macromolecular ligands like CRYBMIM may interact with cellular proteins in complex ways. To define specific effects, we utilized four orthogonal strategies, explained below.

First, we purified the CBP-containing nuclear complex using immunoprecipitation and determined its composition by mass spectrometry proteomics. This analysis revealed 833 proteins that are specifically associated with CBP (revised Supplementary file 3). Although technically feasible, the fact that CBP is associated with hundreds of proteins would make the experiment suggested by the reviewer difficult to interpret, because it would be a major challenge to distinguish proteins bound directly by the peptide versus proteins purified indirectly by virtue of the fact that CRYBMIM binds to CBP/P300, which in turn binds to many other proteins. While we recently developed improved methods for cross-linking mass spectrometry proteomics that permit the identification of direct protein-protein interactions (Ser, Cifani, Kentsis 2019, https://doi.org/10.1021/acs.jproteome.9b00085), we believe that these experiments are beyond the scope of the current manuscript, which already includes 40 new figure panels as part of this revision.

In lieu of this experiment, we purified the CBP-containing nuclear complex after treatment with CRYBMIM or control using immunoprecipitation and determined its composition by targeting Western blotting. This analysis revealed RUNX1, LYL1 and SATB1 are specifically associated with CBP (revised Figure 12B), among which RUNX1 is specifically remodeled in the MYB:CBP/P300 complex upon CRYBMIM binding. This transcriptional factor recruitment and remodeling support the idea of CRYBMIM’s specificity for the MYB:CBP/P300 complex.

Second, to define the specificity of CRYBMIM, we used glycine mutants of CRYBMIM and its parent MYBMIM, CG3 and TG3, respectively, in which residues that form key salt bridge and hydrophobic interactions with KIX are replaced with glycines, but otherwise retain all other features of the active probes. Both CG3 and TG3 exhibit significantly reduced effects on the viability of AML cell lines, consistent with the specific effects of CRYBMIM (Figure 3D).

To confirm that this is due to CBP binding, we purified cellular CBP/P300 by binding to biotinylated CRYBMIM, and observed that it can be efficiently competed by excess of free CRYBMIM, but not TAT (Figure 2E).

Finally, to establish definitively that cellular CBP is responsible for CRYBMIM effects, we generated isogenic cell lines that are either deficient or proficient for CBP using CRISPR genome editing. This experiment demonstrated that CBP deficiency confers significant resistance to CRYBMIM, indicating that CBP is required for CRYBMIM mediated effects (revised Figure 4). We revised the manuscript accordingly.

3) The major limitation of this study which modestly lessens my enthusiasm of this work is that the mechanistic model of MYB-sequestered TFs proposed here is based on a face-value interpretation of IP-MS data coupled with ChIP-seq data. Normally, I would expect such a mechanism to be supported with some additional focused biochemical experiments of specific interactions, to complement all of the omics approaches. For example, can the authors evaluate and/or validate further how MYB physically interacts with LYL1, CEBPA, SPI1, or RUNX1. Are these interactions direct or indirect? Which domains of these proteins are involved? Does CRYBMIM treatment modulate the ability of these proteins to associate with one another in a coIP? Do these interactions occur in normal hematopoietic cells? A claim is made throughout this study that these are aberrant TF complexes, but I believe more evidence is required to support this claim.

We appreciate the reviewer’s comment and totally agree with this point. To examine how MYB aberrantly assembles transcription factors in AML, we performed MYB co-immunoprecipitation (co-IP) in a panel of seven genetically diverse AML cell lines with varying susceptibility to CRYBMIM, chosen to represent the common and refractory forms of human AML. Here, we confirmed co-assembly of CBP/P300, LYL1, E2A, LMO2 in all AML cell lines tested, and cell type-specific co-assembly of SATB1 and CEBPA, as shown in revised Figure 8A, which are in agreement with the IP-MS and ChIP-seq results. We further corroborated these findings by co-IP studies of CBP/P300, as shown in the revised Figure 8B. We performed similar co-IP experiments in normal hematopoietic progenitor cells, and found most of the co-assembled factors in AML cells were not observed in normal cells except for CBP/P300 and LYL1, as shown in the revised Figure 9E. Combined with the apparently aberrant expression of E2A and SATB1 in AML cells but not normal blood cells, this leads us to conclude that MYB assembles aberrant transcription factor complexes in AML cells. These complexes can be remodeled by peptidomimetic inhibitors, leading to their redistribution on chromatin, suppression of oncogenic gene expression and induction of cellular differentiation. We confirmed this mechanism by direct biochemical experiments in AML cells, demonstrating disassembly and remodeling of CBP/P300 complexes, as shown in the revised Figure 12. At least some of these interactions are direct, given the known direct binding between MYB and CEBPA (Oelgeschläger, Nuchprayoon, Lüscher, Friedman 1996, https://doi.org/10.1128/mcb.16.9.4717). We revised the manuscript text accordingly.

Reviewer #1 (Significance (Required)):Overall, this is a strong, well-written study, with clear experimental results and relatively straightforward conclusions. The therapeutic potential of modulating MYB in cancer is enormous, and hence I believe this study will attract a broad interest in the cancer field and will likely be highly cited.

We appreciate this sentiment and completely agree with the reviewer. The phenomenon reported in this work represents the first of its kind demonstration of the aberrant organization of transcription factor control complexes in cancer, and its pharmacologic modulation. We believe that this concept will serve as a transformative paradigm for understanding oncogenic gene control and the development of effective therapies for its definitive treatment.

Reviewer #2 (Evidence, reproducibility and clarity (Required)):This manuscript reports the generation of a new and improved peptide mimetic inhibitor of the interaction between MYB and CBP/P300. The original MYBMIM inhibitor of this interaction, reported recently by the same laboratory, was modified by addition and substitution of peptide sequences from CREB, thus improving the affinity of the resulting CRYBMIM peptide to CBP/P300. The improved inhibitor profile results in increased anti-AML efficacy of CRYBMIM over MYBMIM. The authors go on to examine the mechanism underlying the anti-AML activity of CRYBMIM by integrating gene expression analysis, chromatin immunoprecipitation sequencing and mass spectrometric protein complex identification in human AML cells.I have some minor questions the authors may wish to comment on:1) The relocation of MYB, along with CBP/P300, to genes controlling myeloid differentiation (clusters 4 and 9) upon CRYBMIM treatment is reminiscent of the increased binding of MYB to myeloid pro-differentiation genes in AML cells following RUVBL2 silencing, recently reported in Armenteros-Monterroso et al., 2019. Do the authors know if there is any overlap between genes in either of the clusters and the list reported in the latter study?

We thank the reviewer for making this suggestion. We also observe both RUVBL2 and RUVBL1 in the protein complex specifically associated with MYB (Figure 7A and B). We compared the gene expression changes induced by CRYBMIM with those reported by Armenteros-Monterroso et al. in 2019 (https://doi.org/10.1038/s41375-0190495-8), and found that 37% of upregulated genes by RUVBL2 silencing were shared with genes induced by CRYBMIM treatment. In addition, upregulated genes in cluster 4 and 9 included myeloid differentiation-related genes, such as *JUN*, *FOS* and *FOSB,* which were also induced RUVBL2 silencing. We revised the manuscript to reflect this association.

2) Could the authors comment on a possible mechanism to explain the co-localization of MYB and CBP/P300 to the loci in clusters 4 and 9 following CRYBMIM treatment? Is it possible that CBP/P300 is recruited by other transcription factors to these loci, independently of binding to MYB? Or is the binding of CBP/P300 to MYB at these loci somehow more resistant to disruption by CRYBMIM?

The reviewer has focused on an interesting point. At least for cluster 9, these genes exhibit gain of CBP/P300 in association with RUNX1 (Figure 11A), which we confirm by direct biochemical studies of MYB and CBP/P300 complexes immunoprecipitated from AML cells (revised Figure 12B-C). These experiments show that CRYBMIM treatment disrupts the MYB:CBP/P300 complexes, leading to the increased assembly of CBP/P300 with RUNX1. These findings are consistent with a dynamic competition mechanism that governs availability of CBP/P300 to transcriptional coactivation, in which distinct transcription factors compete for limiting amounts of CBP/P300. This possible mechanism is discussed in the revised manuscript.

3) The text states: "Previously, we found that MYBMIM can suppress MYB:CBP/P300-dependent gene expression, leading to AML cell apoptosis that required MYB-mediated suppression of BCL2 (Ramaswamy et al., 2018)." I think this is a typo, since in this study, MYBMIM treatment results in loss of MYB binding to the BCL2 gene and consequent reduction in BCL2 expression. Do the authors mean “MYBMIM-mediated suppression of BCl2” or “loss of MYB-mediated activation of BCL2”?

We thank the reviewer and have corrected this typographic error in the text.

4) The authors explain the failure of excess CREBMIM to displace CBP/P300 from immobilised CREBMIM (Figure 1E-F) by the nature of the CREB:CBP/P300 interaction. Does this imply that CREBMIM is unable to disrupt the interaction between CREB and CBP/P300 in living cells and that the CBP/P300 purified from native MV4;11 lysates by immobilised CREBMIM was from a pool not associated with CREB?

We thank the reviewer for making this point. Indeed, we reproducibly observe that CRYBMIM binding to CBP can be competed with excess free CRYBMIM, but CREBMIM binding cannot be completed by excess CREBMIM. This may be due to the different stabilities of the CBP complexes that are available for binding in cells. Alternatively, it is also possible that CREB binding to CBP, as reflected by CREBMIM, has a relatively slow dissociation rate, as compared to MYB, as reflected by CRYBMIM. We have begun to purify cellular CBP complexes (revised Figure 8 and response to comment 2 for reviewer 1), and aim to define their determinants in future studies, as enabled by the introduction of CRYBMIM, CREBMIM and MLLMIM probes in the current work.

Reviewer #2 (Significance (Required)):Based on this integrative analysis, the authors propose a convincing hypothesis, involving the assembly of aberrant transcription factor complexes and sequestration of P300/CBP from genes involved in normal myeloid development, for the oncogenic activity of MYB in AML. As well as the obvious therapeutic potential of the CRYBMIM inhibitor itself, the data reported here reveal multiple avenues for future investigation into novel anti-AML therapeutic strategies. This is an innovative and important study.This study will be of interest to scientists and clinicians involved in leukaemia research as well as cancer biology in general.My field of expertise: leukaemia biology, leukaemia models, aberrant transcription factor activity in leukaemia

We appreciate and agree with this assessment.

Reviewer #3 (Evidence, reproducibility and clarity (Required)):This manuscript describes an improved MYB-mimetic peptide (cf the group's earlier work published in Nature Communications, 2018) and its effects on AML cell lines. It also describes – and this constitutes the majority of the paper – the dynamics of chromatin occupancy by MYB and other associated transcription factors upon disruption of the MYB-CBP/P300 interaction. The authors suggest this represents a shift from an oncogenic program to a myeloid differentiation program.Major comments:Regarding the improved affinity, and biological activity, of CRYBMIM:1) Improved affinity of CRYBMIM cf MYBMIM: clearly, it is improved, but not by a lot. By MST the increased affinity is about 3x. In terms of effects on AML cell viability: there is no direct comparison, and this should be included. In the group's previous paper there is no direct estimate for MYBMIM but it looks like the IC50 is between 10 and 20 μM so the effect is again around 2.5 fold. Also, the effects of the amino acid substitutions in CG3 are also very small (2.4x) given that 3 critical residues are altered. This is quite concerning.

As pointed out by the reviewer, CRYBMIM exhibits several fold increase in binding affinity, as measured using purified proteins in vitro. Similar increase in cellular potency is observed after short-term treatment of AML cells, as shown in revised Figure 3C. However, increasing the duration of treatment to several days leads to substantial improvement in apparent cellular potency (Figure 3G). For example, while MYBMIM induces approximately 100-fold reduction in cell viability of MV411 cells, CRYBMIM induces more than 1,000-fold reduction. Similarly, whereas MYBMIM exhibited relatively modest effects on OCIAML3 and SKM1 cells, CRYBMIM induces more than 1,000-fold reduction in cell viability. As we show in the revised manuscript, this appears to be due to the combination of increased biochemical affinity and specific proteolysis of MYB, which cooperate to induce extensive remodeling of MYB transcriptional complexes and gene expression (revised Figure 10). In all, this exemplifies how pharmacologic modulators of protein interactions can achieve significantly improved biological potency from relatively modest affinity effects, a concept that recently has been successfully used to develop a variety of PROTACs that leverage this “event-driven” as opposed to occupancy-driven pharmacology. The manuscript has been revised to clarify this point.

2) Does CRYBMIM really "spare" normal hematopoietic cells? Not according to Figure 2E, where there is only a 2-fold difference in IC50.

To better define the relative toxicity of CRYBMIM and MYBMIM, we examined their effects on the growth and survival of normal hematopoietic progenitor cells as compared to AML cells using colony forming assays in methylcellulose under more physiologic conditions in the presence of human hematopoietic cytokines (revised Figure 3E). While CRYBMIM significantly reduced the clonogenic capacity, growth and survival of MV411 AML cells, there were no significant effects on the total clonogenic activity of normal CD34+ human umbilical cord blood progenitor cells under these conditions. At the highest dose, CRYBMIM induced modest reduction in CFUMG colony formation, and modest increase in BFU-E colony formation of normal hematopoietic progenitor cells. We revised the manuscript to indicate that CRYBMIM “relatively spares” normal blood progenitor cells.

3) Figure 2E and Supp Figure S2 appear to be contradictory. The latter shows no effect of 20 μM CRYBMIM on colony formation by normal CD34+ cells, in complete contrast to killing with IC50 of 12.8 μM in Figure 2E. There is no +ve control for Figure S2 ie does the peptide work under colony assay conditions? This MUST be addressed.

We appreciate the attention to this issue. In the original manuscript, we showed dose-response curves of cord blood progenitor cells cultured in suspension supplemented with fetal bovine serum, a system that is known to induce in appropriate hematopoietic cell differentiation (https://doi.org/10.1016/j.molmed.2017.07.003). In the revised manuscript, we show results of colony formation assays of hematopoietic progenitor cells cultured in serum-free, semi-solid conditions supplemented with human hematopoietic cytokines (revised Figure 3E and 3F). This is a more physiologic system which more faithfully maintains normal hematopoietic cell differentiation, as compared to the cellular differentiation induced by fetal bovine serum-containing media lacking hematopoietic growth factors, as used in the experiments in our original manuscript. To establish a positive control, in addition to treating AML cells under the same condition, we used doxorubicin, which is part of current treatment of patients with AML, and which in our experiments, exhibits significant and pronounced reduction in the clonogenic capacity, growth and survival of normal blood progenitor cells (revised Figure 3—figure supplement 1B). The manuscript has been revised accordingly.

4) Figure 2F doesn't include any lines that express very low or undetectable levels of MYB. Some of these should be included to further examine specificity.

We have now tested CRYBMIM against a large panel of non-hematopoietic tumor and non-tumor cell lines, with varying degrees of MYB expression. Some of those cells exhibit high level of MYB gene expression and MYB genetic dependency, which is at least in part correlated with susceptibility to CRYBMIM. (revised Figure 3—figure supplement 2A). The manuscript has been revised accordingly.

Effects on gene expression and MYB binding:Data on MYB target gene expression and apoptosis/differentiation, and the conclusions drawn per se are sound, but:5) Figure S3 seems to show that MYB protein is lost on treatment with CRYBMIM. This isn't even mentioned in the text but raises a whole range of major questions eg why is this the case? Is this what is responsible for the loss of MYB-p300 interaction and/or biological effects on AML cells? Is this what is responsible for the effects on MYB target gene expression in Figure 3 and MYB binding to chromatin in Figure 4? This must be addressed.

We have revised the manuscript to include this discussion, and performed additional experiments to define this phenomenon. We confirmed rapid reduction in MYB protein levels upon CRYBMIM treatment on the time-scale of one to four hours in diverse AML cell lines (revised Figure 10), with the rate of MYB protein loss correlating to the cellular susceptibility to CRYBMIM (revised Figure 10). The manuscript has been revised accordingly.

This is consistent with the specific proteolysis of MYB induced by the peptidomimetic remodeling of the MYB:CBP/P300 complex. We confirmed this by combined treatment with the proteosomal/protease inhibitor MG132 (revised Figure 10C). This effect was specific because overexpression of *BCL2*, which blocks MYBMIM-induced apoptosis (Ramaswamy et al., Kentsis, https://doi.org/10.1038/s41467-017-02618-6), was unable to rescue CRYBMIM-induced proteolysis of MYB, arguing that MYB proteolysis is a specific effect of CRYBMIM rather than a non-specific consequence of apoptosis. The manuscript has been revised accordingly.

6) Figure 4 and the accompanying text are a bit hard to follow, but if I understood them correctly, I am surprised that the "gained MYB peaks" don't include the MYB binding motif itself? This at least deserves some comment. Also, there doesn't seem to have been any attempt to integrate the ChIP-seq data with the expression data of Figure 3. This would provide clearer insights into the identities and types of MYB-regulated genes that are directly affected by suppression of CBP/p300 binding to MYB.

We thank the reviewer for this suggestion. The revised manuscript now includes a comprehensive and integrated analysis of chromatin and gene expression dynamics. In contrast to the model in which blockade of MYB:CBP/P300 induces loss of gene expression and loss of transcription factor and CBP/P300 chromatin occupancy, we also observed a large number of genes with increased expression and gain of CBP/P300 occupancy (revised Figure 11—figure supplement 1A-B). This includes numerous genes that control hematopoietic differentiation, such as *FOS*, *JUN*, and *ATF3*. As a representative example, in the case of *FOS*, we observed that CRYBMIM-induced accumulation of CBP/P300 was associated with increased binding of RUNX1, and eviction of CEBPA and LYL1.

Thus, the absence of “gained MYB peaks” is due to the redistribution of CBP/P300 with alternative transcription factors, such as RUNX1. In all, these results support the model in which the core regulatory circuitry of AML cells is organized aberrantly by MYB and its associated co-factors including LYL1, CEBPA, E2A, SATB1 and LMO2, which co-operate in the induction and maintenance of oncogenic gene expression, as co-opted by distinct oncogenes in biologically diverse subtypes of AML (revised Figure 12). This involves apparent sequestration of CBP/P300 from genes controlling myeloid cell differentiation. Thus, oncogenic gene expression is associated with the assembly of aberrantly organized MYB transcriptional co-activator complexes, and their dynamic remodeling by selective blockade of protein interactions can induce AML cell differentiation. The manuscript has been revised accordingly.

7) The MS studies on MYB-interacting proteins seem very interesting and novel. I am not an expert on MS, though, so I'd suggest this section be reviewed by someone who is. Moreover, I was unable to see the actual data from this study because the material I was provided with didn't include Table S4 and S5.

We appreciate this point. For this reason, we have deposited all of our mass spectrometry data to be openly available via PRIDE (accession number PXD019708), and also openly provide all of the analyzed data via Zenodo (https://doi.org/10.5281/zenodo. 4321824), as additionally provided in the Supplementary Material for this manuscript.

Should the authors qualify some of their claims as preliminary or speculative, or remove them altogether?8) Claims regarding biological activity, specificity and improvements cf MYBMIM should be moderated given the small size of these effects as mentioned above (points 1 and 3).

As explained in detail in response to comments 1-3 above, we have substantially revised the manuscript to incorporate both new experimental results and additional explanations.

9) I found the description of the studies related to Figures 5 and 6 somewhat difficult to follow and convoluted. While changes in MYB and CBP/p300 chromatin occupancy clearly occur on M CRYBMIM treatment, it is not clear that the complexes seen on genes prior to treatment represent "aberrant" complexes. These may just be characteristic of undifferentiated (myeloid) cells. The authors appear to argue that because some of the candidate co-factors show "apparently aberrant expression in AML cells" based on comparison of (presumably mRNA) expression data with normal cells, the presence of these factors in the complexes make them "aberrant" (moreover, the "aberrancy score" of Figure 5 C is not defined anywhere, as far as I can see). This inference is drawing a rather long bow, given that the AML-specific factors may not actually be absent from the complexes in normal cells. So this conclusion should be moderated if a more direct MS comparison cannot be provided (for which I understand the technical difficulties).

We have now measured protein abundance levels of key transcription factors assembled with MYB in AML cells in various normal human hematopoietic cells (revised Figure 9). We found that most transcription factors that are assembled with MYB in diverse AML cell lines could be detected in one or more normal human blood cells, albeit with variable abundance, with the exception of CEBPA and SATB1 that were measurably expressed exclusively in AML cells (revised Figure 9A). Using unsupervised clustering and principal component analysis, we defined the combinations of transcription factors that are associated with aberrant functions of MYB:CBP/P300, as defined by their susceptibility to peptidomimetic remodeling (revised Figure 9B-D). In addition, we directly examined the physical assembly of MYB with key transcription factors in normal hematopoietic cells using co-immunoprecipitation studies (revised Figure 9E). In agreement with the physical association of MYB seen in AML cell lines, we observed association with CBP/P300 and LYL1 in normal hematopoietic cells. However, we did not observe physical association with E2A and SATB1 in normal cells, which indicates aberrant association of these in AML cell lines. This leads us to propose that these complexes are aberrantly assembled, at least in part due to the inappropriate transcription factor co-expression. The manuscript has been revised accordingly.

Would additional experiments be essential to support the claims of the paper?10) Address the issue of the apparent loss of MYB protein upon CRYBMIM treatment. If this is occurring, the whole premise of the subsequent work is undermined.

As explained in detail in response to comment 5 above, we have carried out extensive studies of the specific proteolysis of MYB. We conclude that MYB transcription complexes are regulated both by MYB:CBP/P300 binding and by specific factor proteolysis, and can be induced by its peptidomimetic blockade in AML cells. Such “event-driven” pharmacology is emerging as a powerful tool to modulate protein function in cells, and studies reported in our work should enable its translation into improved therapies for patients, and improved probes for basic science.

11) Provision of a positive control for the experiment of Figure S2.

As explained in detail in response to comment 2 above, we precisely defined the effects of CRYBMIM and MYBMIM on the clonogenic capacity, growth and survival of normal hematopoietic progenitor cells in serum-free, methylcellulose media supplemented with human hematopoietic cytokines. These experiments showed relatively modest effects (9.3 ± 3.8% reduction) of CRYBMIM on normal cells (Figure 3E), as compared to substantial inhibition (54 ± 2.4 % reduction) of the growth and survival of AML cells (Figures 3E). For comparison, doxorubicin led to more than 98 % reduction in clonogenic capacity (revised Figure 3—figure supplement 1B).

12) Are the data and the methods presented in such a way that they can be reproduced?

The revised manuscript includes a complete description of all methods, including a detailed supplement, listing technical details, with all analyzed data available via Zenodo (https://doi.org/10.5281/zenodo.4321824).

13) Are the experiments adequately replicated and statistical analysis adequate?

All experiments were performed in at least three replicates, with all quantitative comparisons performed using appropriate statistical tests, as explained in the manuscript.

Minor comments:Specific experimental issues that are easily addressable.These are mostly indicated above.In addition:14) Why is BCL2 expression down-regulated by MYBMIM but not CRYMYB?

We made the same observation, and attribute this difference to the fact that *BCL2* expression is regulated by several transcription factors, including CEBPA, which is affected by CRYBMIM but not MYBMIM. Similar to MYBMIM treatment, MYB occupancy at the *BCL2* enhancer was reduced upon CRYBMIM treatment. However, new binding sites of other factors, such as CBP/P300 and RUNX1, appeared simultaneously, suggesting that redistribution of transcription factors following CRYBMIM treatment can affect transcriptional regulation of *BCL2* expression.

Are prior studies referenced appropriately?

Yes

Are the text and figures clear and accurate?15) Generally, although some details are missing eg what aberrancy score in Figure 5C means.

Thank you for pointing this out. We have revised this figure to clarify this score, which is defined as the ratio of gene expression in AML cells relative to normal hematopoietic progenitor cells (revised Figure 7C).

16) Do you have suggestions that would help the authors improve the presentation of their data and conclusions?The title of this manuscript could and I think should be changed. The term "therapeutic", is not appropriate because no therapeutic agents are described in the manuscript nor is any form of AML, even experimentally, treated. Also "CBP" should be replaced with CBP/P300, especially since most evidence suggests that P300 is the likely more important partner of MYB (eg Zhao et al., 2011).

We agree and have revised the title to clarify the significance of this work: “Convergent organization of aberrant MYB complexes controls oncogenic gene expression in acute myeloid leukemia.” We have revised the manuscript to indicate CBP/P300.

17) It would be worth discussing the core observation that disruption of the MYB-CBP/P300 interaction actually results in changes in MYB DNA binding. That this would occur is not at all obvious, because CBP/p300 doesn't interact with MYB's DNA binding domain nor does it have intrinsic DNA binding activity.

We thank the reviewer for this comment, and agree that remodeling of the MYB complex must affect the binding of MYB and other cofactors to DNA, at least in part mediated by potential acetylation by CBP/P300.

Reviewer #3 (Significance (Required)):The Nature and Significance of the Advance1) The major significance of this work lies in the chromatin occupancy and MYB complex studies. There are a number of very interesting findings including the apparent redistribution of MYB and/or CBP/P300 upon treatment with CRYBMIM. These suggest a series of changes in factors associated with particular gene sets involved in myeloid differentiation, although as mentioned above particular target genes are not specifically identified. However the pathways corresponding to these are listed in Table S6.

We have revised the manuscript to include the target genes in revised Supplementary file 2 as well as DESeq2 tables (deposited in Zenodo, https://doi.org/10.5281/zenodo.4321824).

2) The new peptide design (CRYBMIM) is interesting but its differences in binding and biological effects of MYBMIM are mostly incremental. See above.

We respectfully disagree and would like to explain how this work is significant both for conceptual and technical reasons. First, while the biochemical affinity of CRYBMIM is quantitatively increased compared with MYBMIM, this quantitatively increased affinity translates into qualitatively improved biological potency, as a result of “event-driven” pharmacology that characterizes pharmacologic protein interaction modulators (please also see response to reviewer 3, comment 1). MYBMIM suppresses the growth and survival mostly of MLL-rearranged leukemias, whereas CRYBMIM does so for the vast majority (10 out of 11) of studied subtypes of AML. This now enables its therapeutic translation, as we are currently pursuing in collaboration with Novartis. Second, its improved biological activity led to the discovery of the previously unknown and unanticipated CBP/P300 sequestration mechanism of oncogenic gene control. We use this discovery to develop a precise model of aberrant gene control in AML that for the first time unifies previously disparate observations into a general mechanism. This is highly significant because it provides shared molecular dependencies for most subtypes of AML, a long-standing conundrum in cancer biology.

Place the work in the context of the existing literature (provide references, where appropriate).– This manuscript builds on and extends the report from the same group in Nature Communications (2018), which described the earlier peptide MYBMIM, some effects on MYB target genes and on AML cells. It and the previous paper also draw on the findings regarding the role of the MYBCBP/P300 interaction in myeloid leukemogenesis (Pattabirman et al., 2014) and on previous genome-wide studies of MYB target genes (Zhoa et al., 2011; Zuber et al., 2011).State what audience might be interested in and influenced by the reported findings.– This manuscript will likely be of interest to scientists interested in MYB per se, in AML, in cancer genomics and transcriptional regulation.Define your field of expertise with a few keywords to help the authors contextualize your point of view. Indicate if there are any parts of the paper that you do not have sufficient expertise to evaluate.– My expertise: AML, experimental hematology, transcription, MYB, cancer genomics3) As mentioned above, I feel that additional expertise is required to review the MS studies.

We have deposited all raw data in PRIDE (accession number PXD019708) and all processed data in Zenodo (https://doi.org/10.5281/zenodo.4321824), making it available for the community for further analysis.

[Editors' note: further revisions were suggested prior to acceptance, as described below.]

Revisions:While the authors have satisfactorily responded to the majority of the comments raised by the reviewers of Review Commons, some additional issues have to be addressed before publication can be recommended.1) Throughout the manuscript, the authors include CEBPA as a MYB cofactor, but it does not co-precipitate with MYB in any of the immunoblots that are shown with the exception of a very weak band in OCAIML2 cells in Figure 8A. The evidence for a MYB-CEBPA interaction is thus very tenuous. The authors should address this issue either by tempering their claims or providing more convincing evidence.

Thank you for this point. The mass spectrometry data shows high abundance and high confidence identification of several C/EBP family members, including CEBPA itself. However, as pointed out, CEBPA associates with the MYB complex as measured by Western blots largely in OCIAML2 cells. Thus, we have revised the manuscript throughout to replace CEBPA with “variable C/EBP family member(s),” and tempered all relevant claims accordingly.

2) It is not clear why CRISPR inactivation of CBP shown in Figure 4 can have such minor effects, considering that the main message of the paper is that MYB-CBP interactions are critical for activating the genes required for growth of the leukemic cells. The authors have to provide a clear explanation of this apparent discrepancy. Related to this is an important comment of reviewer 3 from Review Commons that the peptide inhibitor induced MYB degradation that is clearly seen after 4 hours of treatment in Figure 11C. Given this observation, how can the authors explain the gained MYB peaks in the ChIP-seq. The authors have not correctly addressed this issue. It is of course reasonable to imagine that MYB stability depends upon its interactions with CBP and that when this is blocked by the peptide MYB is degraded. But if this is the case why does CRIPSR inactivation of CPB not induce the same effects. In fact, the stability of MYB in the CBP KO cells should be investigated.

This is a simple and effective experiment, and we thank you for suggesting it. Steadystate levels of MYB protein do not appear to change significantly upon depletion of CBP in MV411 AML cells, as compared to wild‐type cells or control cells targeted with CRISPR to *AAVS1*.

This suggests that either P300 can compensate for CBP‐mediated functions, and/or MYB proteolysis is not regulated solely by its interaction with CBP/P300, but requires specific activity of CBP/P300 induced by CRYBMIM. We have included this result in the revised manuscript.

In the case of chromatin association of MYB upon CRYBMIM treatment as measured by ChIP‐seq, they can be explained by residual MYB protein upon CRYBMIM‐induced proteolysis. This may be due to protection of chromatin‐bound MYB from proteolysis, or alternatively by signal normalization, required for ChIP‐seq analysis. MYB‐binding loci lost upon CRYBMIM treatment showed significant enrichment for known MYB binding motifs, while gained MYB peaks did not (p = 1e‐149 vs 1e‐3, 56 % vs 5.7 % of target sites, respectively). This raises the possibility that DNA binding affinity of MYB could be regulated by CBP/P300, either by direct effects such as MYB deacetylation upon CRYBMIM treatment, or indirectly via binding with other transcription factors. Absolute quantitative measurements, using spike‐in controls, will be needed to define this precisely in future studies. We have revised the manuscript to clarify these points (Figure 6‐figure supplement 1).